# Sleep pressure modulates single-neuron synapse number in zebrafish

Anya Suppermpool[1,2], Declan G. Lyons[1], Elizabeth Broom[1] & Jason Rihel[1✉]

Sleep is a nearly universal behaviour with unclear functions[1]. The synaptic homeostasis hypothesis proposes that sleep is required to renormalize the increases in synaptic number and strength that occur during wakefulness[2]. Some studies examining either large neuronal populations[3] or small patches of dendrites[4] have found evidence consistent with the synaptic homeostasis hypothesis, but whether sleep merely functions as a permissive state or actively promotes synaptic downregulation at the scale of whole neurons is unclear. Here, by repeatedly imaging all excitatory synapses on single neurons across sleep–wake states of zebrafish larvae, we show that synapses are gained during periods of wake (either spontaneous or forced) and lost during sleep in a neuron-subtype-dependent manner. However, synapse loss is greatest during sleep associated with high sleep pressure after prolonged wakefulness, and lowest in the latter half of an undisrupted night. Conversely, sleep induced pharmacologically during periods of low sleep pressure is insufficient to trigger synapse loss unless adenosine levels are boosted while noradrenergic tone is inhibited. We conclude that sleep-dependent synapse loss is regulated by sleep pressure at the level of the single neuron and that not all sleep periods are equally capable of fulfilling the functions of synaptic homeostasis.

Although sleep is conserved across the animal kingdom[1], the precise functions of sleep remain unclear. As sleep deprivation leads to acute impairment of cognitive performance[5], many theories posit that synaptic plasticity associated with learning and memory preferentially occurs during sleep[6]. For example, the synaptic homeostasis hypothesis (SHY) proposes that synaptic potentiation during wakefulness results in an ultimately unsustainable increase in synaptic strength and number that must be renormalized during sleep through synaptic weakening and pruning[2,7,8]. Such sleep-dependent renormalization has been postulated to broadly affect most excitatory synapses throughout the brain[2].

Many, but not all, experimental observations of brain-wide changes in synapses have been consistent with the SHY. Globally, synaptic genes, proteins and post-translational modifications are upregulated during waking and renormalized during sleep[9–12]. In both flies and mice, the number and size of excitatory synapses also increase after prolonged waking and decline during sleep[3,10,13]. Long-term imaging of small segments of dendrites in young and adult mice has also been used to observe sleep–wake-linked synapse dynamics[4,14,15] and, in zebrafish, axon terminals of wake-promoting hypocretin neurons are regulated by the circadian clock[16]. However, other studies have observed no impact of sleep–wake states on synaptic strength and neuronal firing rates[17,18], and some have observed synaptic strengthening during sleep[19–22]. Furthermore, distinct classes of synapse within the same neuronal population can be differentially regulated by sleep–wake states[23], consistent with observations that synaptic plasticity can be regulated in a dendritic-branch-specific manner[24]. Together, these observations paint a complex picture of how sleep sculpts synapse number and strength, raising fundamental questions about whether sleep-dependent synaptic homeostasis operates uniformly across neuronal types and at which scale (for example, dendrite, neuron, circuit or population) sleep acts to modulate synapses.

To examine the scope and selectivity of sleep-linked synaptic plasticity, it is vital to comprehensively track the synaptic changes of individual neurons through sleep–wake states. To that end, we used in vivo synaptic labelling tools in larval zebrafish to image the same neurons and their synapses repeatedly over long timescales, enabling us to map single-neuron synapse changes across sleep and wake states.

## Synapse counts change across 24 h

To visualize excitatory synapses in single zebrafish neurons, we adapted an established fibronectin intrabodies generated with mRNA display (FingR)-based transgenic system that selectively binds to and labels postsynaptic density protein 95 (PSD95)[25–27], a major postsynaptic scaffold of excitatory synapses[28,29] and a readout of synaptic strength[30,31], to enable simultaneous imaging of synapses and neuronal morphology (Fig. 1a). Consistent with previous reports[25,27,32], we confirmed that this modified FingR(PSD95) system labels synapses with high fidelity by driving expression of *Tg(UAS:FingR(PSD95)-GFP-P2A-mKate2f)* in the spinal cord with a *Tg(mnx1:Gal4)* driver line and co-labelling with anti-MAGUK antibodies that recognize the PSD95 protein family. Greater than 90% of FingR(PSD95)+ puncta associated with MAGUK, while 100% of neuronal MAGUK puncta were co-labelled with FingR(PSD95) (Extended Data Fig. 1a–e,h–i). The signal intensities of co-labelled MAGUK and FingR(PSD95) synapses were positively correlated, indicating that

[1]Department of Cell and Developmental Biology, University College London, London, UK. [2]Present address: UCL Ear Institute, University College London, London, UK. ✉e-mail: j.rihel@ucl.ac.uk

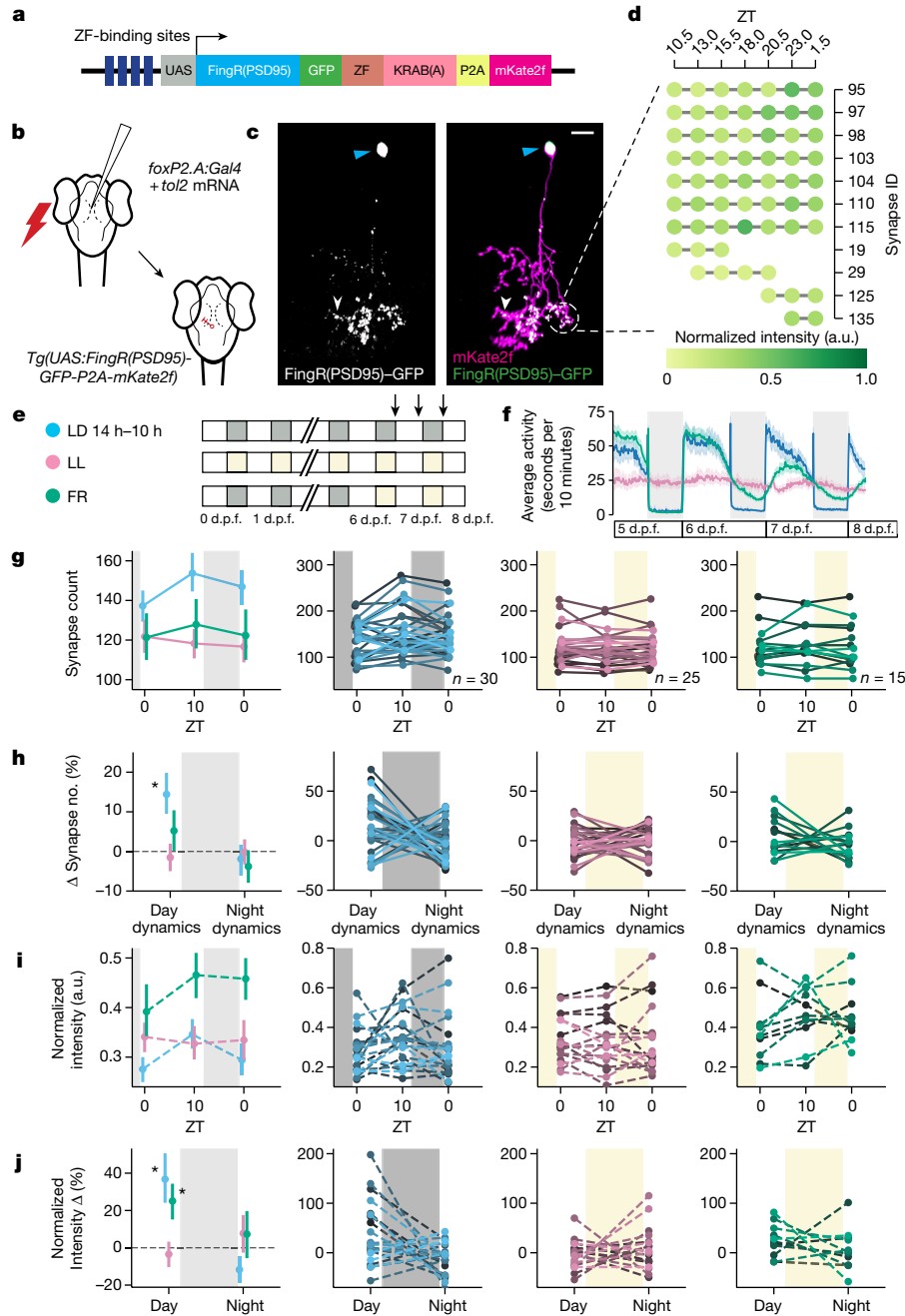

**Fig. 1 | Single-neuron synapse tracking across day–night cycles reveals diverse dynamics. a**, The synapse labelling construct. Zinc finger (ZF) and KRAB(A) domains limit overexpression[25]. **b**, The strategy to sparsely label synapses of FoxP2.A[+] tectal neurons (Methods). **c**, Example FoxP2.A:FingR(PSD95)[+] neuron at 7 d.p.f., with the synapses (white arrowheads, left), nucleus (blue arrowheads, left) and membrane (magenta, right) co-labelled. **d**, Overnight time-lapse tracking of select synapses from the neuron in **c**. The normalized GFP intensity (shading) is shown for each synapse (rows). The complete neuron map is shown in Extended Data Fig. 2a. **e**, Larvae were raised on 14 h–10 h light–dark (LD) cycles (blue), constant light (LL, pink) or switched from LD to LL at 6 d.p.f. (free running (FR), green), and then imaged (arrows) (Methods). **f**, The average locomotor activity and 95% confidence intervals (CIs) of larvae reared under LD (blue, n = 75), clock-break LL (pink, n = 84) or FR (green, n = 98) conditions. **g–j**, The mean and 68% CI (column 1) and individual neuron (columns 2–4) synapse counts (**g**), percentage change in synapse number calculated within each neuron (**h**), normalized synapse intensity (**i**) and percentage change in synapse intensity (**j**) under the LD (blue),

LL (pink) or FR (green) conditions. For columns 2–4, a line is shown for each neuron, collected across 8 LD, 4 LL and 4 FR independent experiments. For **h**, synapse number change (Δ synapse number) dynamics are different during the day from those during the night under LD conditions (*P = 0.043, repeated-measures analysis of variance (ANOVA)). Synapse number change dynamics under LD cycling are significantly different from those under LL conditions (*P = 0.015, main effect of condition, two-tailed mixed ANOVA, post hoc Benjamini–Hochberg correction; Hedge's g = 0.761). For **j**, day–night dynamics are significantly different under LD from those under the other conditions (P < 0.01, repeated-measures ANOVA). Both daytime FR and LD day–night dynamics are significantly different from those under the LL condition (mixed ANOVA interaction (condition × time), P = 0.029; FR versus LL, P = 0.038, g = 0.937; LD versus LL, P = 0.027, g = 0.792; post hoc Benjamini–Hochberg correction, two-tailed). At night, LD versus FR, g = −0.538; LD versus LL, g = −0.527. The diagram in **a** is adapted from ref. 27, CC BY 4.0, and the diagram in **b** is adapted from ref. 33, CC BY 4.0. The colour key in **e** applies also to **f–i**.

the signal intensity is a reliable readout of synaptic PSD95 content, as reported previously[26] (Extended Data Fig. 1f,g).

To test whether behavioural state modulates synapse strength and number at the single-neuron level, we focused on larval tectal neurons, which are accessible to imaging, have well-defined morphological and functional identities[33] and have a stable window of synapse maturation from 7 to 9 days post-fertilization (d.p.f.)[34]. Tectal neurons also undergo spike-timing-dependent plasticity[35] and receive a mixture of inputs that foster 'competition' among synapses[36,37], a criterion envisaged by the SHY[2]. To sparsely label tectal neurons, we co-electroporated a plasmid driving Gal4 off the *foxp2.A* promoter with *tol2* mRNA into *Tg(UAS:FingR(PSD95)-GFP-P2A-mKate2f)* larvae at 3 d.p.f.[38] (Fig. 1b,c and Methods). This method resulted in approximately 10% of larvae containing a single FoxP2.A:FingR(PSD95)⁺ neuron, allowing for repeated, long-term imaging of the synapse counts and intensities in the same neuron in a continuously mounted preparation (Fig. 1c,d and Extended Data Fig. 2a). After confirming the relative stability of tectal neuron synapse counts in the 6–9 d.p.f. developmental window (Extended Data Fig. 2b–d), we imaged each labelled neuron across a 14 h–10 h light–dark cycle at 7 d.p.f., collecting images just after lights on (zeitgeber time 0 (ZT0), 7 d.p.f.), near the end of the day (ZT10) and after a night of sleep (ZT0, 8 d.p.f.) (Fig. 1e; an example neuron with synapse changes tracked across two timepoints is shown in Extended Data Fig. 3), leaving larvae to behave freely between imaging sessions. On average, the tectal neuron synapse number increased significantly during the day from 137 to 153 synapses (+14.4%) but decreased at night by −1.90% to 146 synapses (Fig. 1g,h (blue)). Similar day–night changes in the net synapse counts were observed in separate experiments that imaged neurons over multiple days and nights (Extended Data Fig. 4a–e), with no evidence of artefacts from repeated imaging (Extended Data Fig. 4f–h). Moreover, the average synapse FingR(PSD95)–GFP signal intensity increased significantly during the waking day phase (+36.8%) and decreased in the night sleep phase (−11.7%) (Fig. 1i,j).

To test whether these synaptic dynamics are influenced by the direct action of lighting conditions or are instead controlled by an internal circadian clock, we also tracked neurons under conditions of either constant light from fertilization, which prevents the formation of functional circadian clocks and leads to arrhythmic behaviour in zebrafish (clock-break)[39–41], or constant light after light–dark entrainment, which maintains damped circadian behaviour (free running)[42] (Fig. 1e,f). Under clock-break conditions, changes in synapse number and intensity were abolished and remained smaller compared with in larvae raised on light–dark cycles (Fig. 1g–j (pink)). Under free-running conditions, synapse numbers continued to increase during the subjective day and decrease during the subjective night, albeit strongly damped (Fig. 1g,h (green)). The average synapse intensity was significantly elevated across all timepoints and showed a further significant increase in strength only during the subjective day, with no loss of intensity during the subjective night (Fig. 1i,j (green)). Collectively, these data show that, while light influences the baseline levels of synaptic strength (Fig. 1i), changes in synapse counts are independent of lighting conditions but do require an intact circadian clock (to drive rhythmic sleep–wake behaviour; see below) (Fig. 1g).

Moreover, although rhythmic day–night changes in synapses were detected in the average of all of the single neurons, the tracking of individual neurons revealed that many cells have different, even opposing, synaptic dynamics (Fig. 1g–j (right)). We therefore sought to test whether these diverse patterns mapped onto distinct neuronal subtypes (that is, cellular diversity) or whether they are due to variations in animal behaviour (that is, individual sleep–wake histories).

## Synapse cycling across neuronal subtypes

To test whether distinct synapse day–night dynamics are associated with morphological subtypes of tectal neurons, we measured position, branching, length and other parameters of FoxP2.A:FingR(PSD95)–GFP⁺ neurons, many of which project only within the tectum at 7 d.p.f. Clustering analysis found four subtypes, consistent with previous studies[33,43] (Fig. 2a–c and Extended Data Fig. 5a–c). Tracking synapses across three light–dark cycles revealed that each neuronal subtype has, on average, different patterns of net synapse counts (excluding the rarely observed type 1 neurons). Specifically, dynamics consistent with the SHY were robustly observed only in the densely bistratified type 2 neurons, with an average increase of 15.3 synapses during the day and a reduction of 17.7 synapses at night, and weakly observed in type 4 neurons (+8.5 during the day and −8.2 overnight; Fig. 2d–g and Extended Data Fig. 5d–f). By contrast, many type 3 neurons consistently exhibited the opposite pattern, with an average increase in synapse number at night and a slight decrease during the day (Fig. 2d–g). However, compared with under clock-break conditions, in which no subjective day–night-linked changes occur (Extended Data Figs. 5g–j and 6a,b), the FingR(PSD95)–GFP signal intensity of type 3 and 4 neurons, but not type 2 neurons, increased during the day and decreased at night (Extended Data Fig. 6a–c), suggesting that synapse number and PSD95 content are differentially regulated in tectal subtypes. These subtype-specific alterations in synapse number cannot be explained by differences in larval sleep–wake behaviour, as the sleep amount was the same regardless of which neuron subtype was labelled in the larva (Extended Data Fig. 7a–c).

As type 2 neurons have two prominent arbourization fields, we examined whether changes in day–night synapse number are heterogenous across different dendritic segments of individual neurons. Analysing the synapse number changes in four distinct classes of dendritic segment in type 2 neurons revealed that only the proximal arbour, which receives local inputs from the tectum and long-range inputs from brain areas such as the hypothalamus[44], displayed significantly robust average increases in synapse number during the day and reductions at night (Extended Data Fig. 7d–f). By contrast, synapse number dynamics within the distal arbour, which receives the majority of its inputs from the retina[43], were more diverse. No correlations could be detected among the different dendritic compartments within the same neuron (Extended Data Fig. 7f), suggesting that the time of day and sleep–wake states do not have uniform effects on synapse number even within the same neuron.

## Sleep pressure facilitates synapse loss

If the synapses of individual neurons are regulated by sleep–wake states independently of the circadian clock, these dynamics should be altered by sleep deprivation (SD). We developed a gentle handling SD protocol in which zebrafish larvae are manually kept awake with a paintbrush for 4 h at the beginning of the night (ZT14–ZT18) and subsequently allowed to sleep (Supplementary Video 1). Sleep in larval zebrafish is defined as a period of inactivity lasting longer than 1 min, as this is associated with an increased arousal threshold, homeostatic rebound and other criteria of sleep[40,45]. After SD, the phase of the circadian clock machinery was unaffected, but larvae slept significantly more, with individual sleep bouts lasting longer, compared with non-sleep-deprived larvae (Extended Data Fig. 8a,b), consistent with SD leading to increased sleep pressure[46–48]. Next, we visualized synapses of individual tectal neurons at 7 d.p.f. immediately before (ZT13–ZT14) and after (ZT18–ZT20) SD, and again the next morning (ZT0–ZT1) (Fig. 3a and Extended Data Fig. 9a). Between the imaging sessions, we used video tracking to monitor sleep–wake behaviour (Methods). In control larvae, tectal neurons lost synapses overnight; however, this synapse loss was confined to the first part of the night (ZT14–ZT18), with an average loss of 1.7 synapses per hour, in contrast to the last part of the night (ZT18–ZT24), during which synapse loss was undetectable (+0.2 synapses per hour) (Fig. 3b (blue)). By contrast, neurons gained an average of 2.8 synapses per hour during SD (Fig. 3b (orange)). During the recovery period after SD,

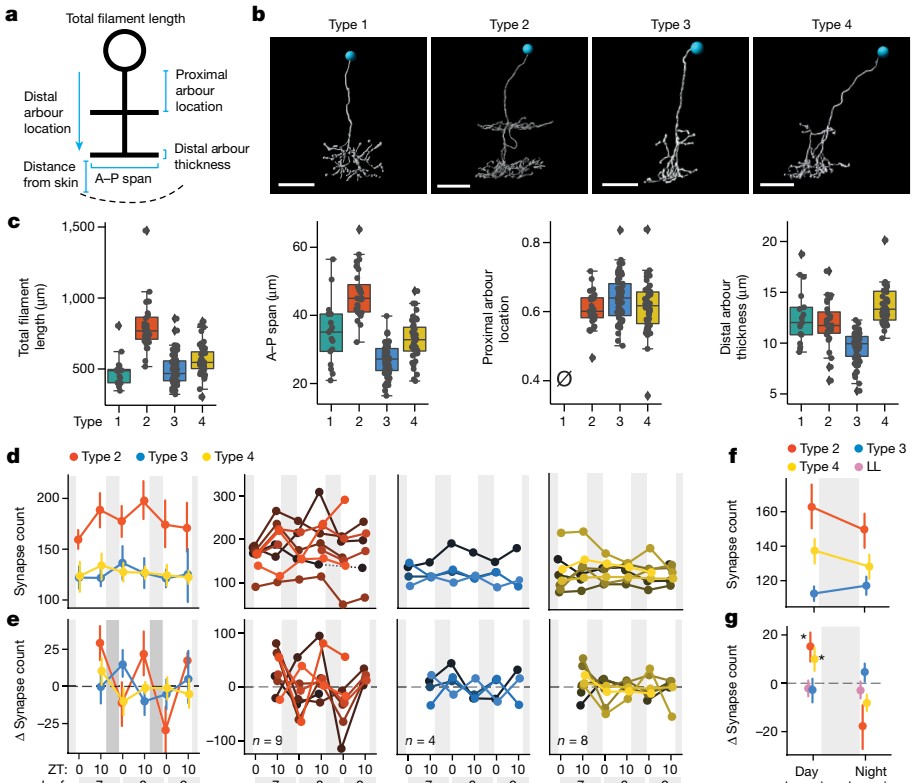

**Fig. 2 | Subtype-specific synapse changes in FoxP2.A tectal neurons over 3 days. a**, The morphological parameters used to characterize FoxP2.A tectal neurons. A–P, anterior–posterior. **b**, Examples of each morphological subtype, chosen from $n = 17$ (type 1), $n = 28$ (type 2), $n = 61$ (type 3) and $n = 42$ (type 4) neurons collected over 26 independent experiments. The blue circles label nuclei. **c**, Example of the parameters used to distinguish the four subtypes. For the box plots, the centre lines show the median, the box limits show the interquartile range and the whiskers represent the distribution for each parameter. The slashed zero indicates that the feature is absent. See also Extended Data Fig. 5. **d**–**g**, Synapse counts across multiple LD cycles for FoxP2.A tectal neurons of different subtypes. **d**,**e**, Average (68% CI) synapse counts (**d**) and average (68% CI) synapse number change (**e**) of subtypes (column 1) and for each neuron (columns 2–4), collected over 8 independent experiments. **f**,**g**, Average (68% CI) synapse counts (**f**) and net change (**g**), averaged across all days and nights for each subtype and larvae, including additional neurons tracked over a single day (Extended Data Fig. 5). Tectal subtype influences synapse changes (mixed ANOVA, interaction $P = 0.012$, subtype × time). Type 2 ($n = 16$) and type 4 ($n = 15$) neurons gain more synapses during the day under LD conditions compared with under LL clock-break conditions ($P = 0.018$, $g = 0.952$; $P = 0.021$, $g = 0.812$, respectively). At night, both type 2 and type 4 neurons lose synapses relative to type 3 (type 2 versus type 3, $P = 0.038$, $g = -0.714$; type 4 versus type 3, $P = 0.038$, $g = -0.781$, post hoc Benjamini–Hochberg correction, one-tailed). For **b**, scale bars, 10 μm.

tectal neurons lost synapses at a rate of 2.2 synapses per hour (Fig. 3b and Extended Data Fig. 8c). As during normal sleep, FoxP2.A tectal neuron subtypes responded differently to SD, with type 2 and even type 3 neurons (which did not have SHY-concordant changes under baseline conditions) gaining synapses during SD and losing them during recovery sleep, whereas type 4 neurons did not show any change (Extended Data Fig. 8d). This suggests that SD biases synapses towards loss during subsequent sleep, even in neurons with different synapse dynamics under baseline conditions.

As both SD and control larvae were at the same circadian phase, we conclude that sleep–wake states are the main driver of net changes in synapses in tectal neurons, and the effects of circadian clock disruption on synapses were primarily due to the loss of sleep rhythms (Fig. 1). Consistent with this interpretation, the total time that each larva spent asleep was significantly correlated with the rate of synapse change (Fig. 3c and Extended Data Fig. 8g). Only after SD, when sleep and synapse loss were high across most larva–neuron pairs, was this correlation lost, which may indicate that either the machinery that supports sleep-dependent synapse loss can saturate or SD-induced rebound sleep is not fully equivalent to baseline sleep. The converse relationship was not observed, as the rate of synapse gain during SD was not correlated with either the subsequent total sleep or the average sleep bout lengths of single larvae (Extended Data Fig. 8f). Consistent

with the effects of SD, natural individual variation in sleep timing was predictive of the time period in which synapses were lost. 'Early sleepers' slept more during the first half of the night and lost synapses only during this period, whereas 'late sleepers' preferentially slept in the second half of the night and had a net loss of synapses only during the late night (Fig. 3d,e and Extended Data Fig. 8e). Finally, to test whether sleep-dependent synapse loss is generalizable to neurons that do not receive direct retinal input, we confirmed that synapses of both presumptive vestibulospinal neurons that stabilize posture[49] and MiD2cm reticulospinal neurons involved in fast escapes[50,51] showed synapse gains during SD and synapse loss during sleep (Fig. 3f–h).

Two explanations are consistent with the observed relationships between sleep and synapse change: either sleep is a permissive state for synapse loss, or sleep pressure, which builds as a function of waking, drives synapse loss during subsequent sleep. As sleep pressure and subsequent sleep amount at night are tightly linked under both baseline and SD conditions, we sought to disentangle their relative influences on synaptic change using sleep-inducing drugs to force larvae to sleep during the day, when sleep pressure remains low (Fig. 4a,b and Extended Data Fig. 9c). Exposing larvae for 5 h during the day (ZT5–ZT10) to either 30 μM melatonin, which in zebrafish is a natural hypnotic that acts downstream of the circadian clock to promote sleep[52], or 30 μM clonidine, an α2-adrenergic receptor agonist

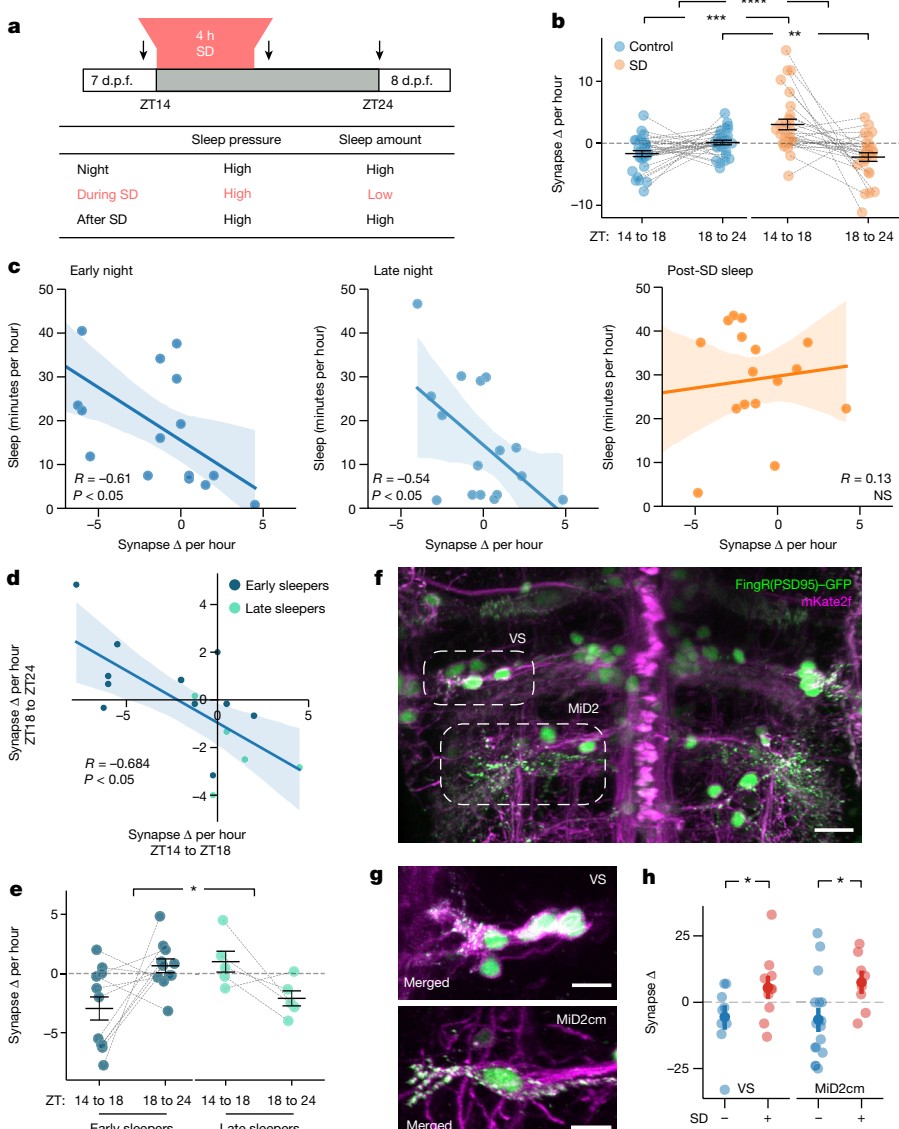

**Fig. 3 | Synapse counts of neurons are modulated by sleep and SD. a**, The 4 h gentle handling SD paradigm (ZT14–ZT18). Larvae were video-tracked and neurons were periodically imaged (arrows). **b**, The mean ± s.e.m. change in synapse counts per hour for the SD (orange, n = 31 neurons) and control (blue, n = 28) groups. **c**, Sleep time versus the change in synapse counts per hour for each larva during either the early (ZT14–ZT18, left) or late (ZT18–ZT24, middle) night for controls and after SD (ZT18–ZT24, right). The rate of synapse change is negatively correlated with sleep time during both early and late night but not after SD. **d**, In control larvae, the change in early night synapse counts is negatively correlated with late night synapse change. Early and late sleepers are defined as larvae that either sleep more in the first or second phase of the night, respectively. **e**, Synapse counts per hour for early- and late-night sleeping control larvae in the early (ZT14–ZT18) and late (ZT18–ZT24) phases of the night. Data are mean ± s.e.m. **f–h**, The reticulospinal neuron synapse number is

modulated by sleep and wake states. **f**, Example reticulospinal neurons from the *Tg(pvalb6:KALTA4)$^{u508}$* line co-labelled by FingR(PSD95)–GFP (green, nuclei and synapses) and mKate2f (magenta, membrane). Vestibulospinal (VS) and MiD2cm neurons are indicated by the dashed ovals. **g**, Vestibulospinal (top) and MiD2cm (bottom) neurons from different larvae showing FingR(PSD95)$^+$ synapses (green) co-localized to the cell membrane (magenta). **h**, Changes in synapse number (mean and 68% CI) from ZT14 to ZT18 for vestibulospinal and MiD2cm neurons. Each dot represents the average across multiple neurons per larva. For **b** and **e**, statistical analysis was performed using two-tailed mixed ANOVA interaction (condition × time) with post hoc Benjamini–Hochberg correction; ****P = 0.00007, ***P = 0.0002 and **P = 0.006 (**b**) and *P = 0.01 (**e**). For **h**, statistical analysis was performed using one-tailed Student's t-tests; *P < 0.03. Scale bars, 15 µm (**f**) and 10 µm (**g**). The lines in **c** and **d** depict the linear regression with the 95% CI.

that inhibits noradrenaline release and increases sleep in zebrafish[45,53], significantly and strongly increased total sleep and the average length of sleep bouts mid-day (Fig. 4c and Extended Data Fig. 10a,b), with this drug-induced sleep remaining reversible by strong stimuli (Extended Data Figs. 9d,e and 10d–g). Forced daytime sleep altered the build-up of sleep pressure, leading to reduced and delayed sleep in the subsequent night (Extended Data Fig. 9e). However, drug-induced sleep at a time of low sleep pressure was not sufficient to trigger synapse loss, with tectal neurons still gaining an average of 1.0–1.7 synapses per

hour, which was not significantly different from the synapse gains in the controls (Fig. 4d). Similarly, artificially boosting adenosine signalling—one of the postulated molecular substrates of sleep pressure[54]—by administering 45 µM 2-choloroadenosine increased sleep during the day but also led to net gains in tectal neuron synapses (+0.9 synapse per hour) (Fig. 4c and Extended Data Fig. 10c). Tectal neurons also gained synapses (+0.4 synapse per hour) in larvae that were co-administered 2-chloroadenosine and melatonin, despite sleeping more than 35 minutes per hour (Fig. 4c,d). By contrast, simultaneously

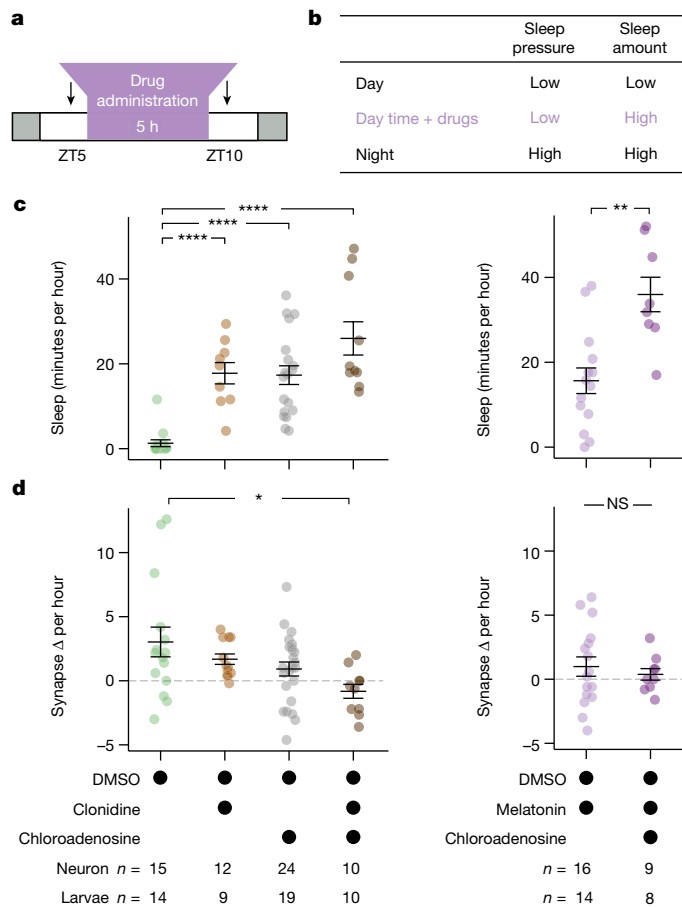

basis, more diverse patterns of synapse change are revealed. These observations may explain some discrepancies among previous studies of the SHY, as these single-neuron synaptic dynamics would not be captured by population-level, single-time-point snapshots of synapse number or function. We also show that sleep is necessary but not sufficient for synaptic loss, as synapse loss occurred only when sleep was accompanied by high sleep pressure associated with adenosine signalling and low noradrenergic tone. Adenosine signalling has been shown to promote Homer1a-dependent downscaling and destabilization of synapses, whereas noradrenergic signalling has been found to prevent this process[55]. Our data link these mechanisms to sleep pressure and sleep behaviour in vivo. Whether single-neuron or subcellular variation in the expression or sensitivity to these synapse-regulating signals could account for the diversity of synapse alterations remains an interesting possibility for future work. Sleep pressure, as reflected by the density of slow-wave activity in mammalian sleep, has also been linked to changes in synapses associated with learning and memory[11,56]. We find that sleep-linked synapse loss depends on molecular signals linked to high sleep pressure and, notably, also mirrors slow-wave activity by occurring predominantly in the early part of the sleep period[6]. This finding raises the question of whether epochs of sleep associated with low sleep pressure, such as in the latter half of the night, have additional, non-synaptic remodelling roles. If so, the evolution, persistence and ubiquity of these different sleep epochs could be under specific regulatory and selective pressures.

**Fig. 4 | Single-neuron synapse loss during sleep is driven by boosting adenosine and blocking noradrenaline. a**, Larvae were temporarily treated with sleep-promoting drugs during the day (ZT5–ZT10). The black arrows indicate the imaging periods before and after drug treatment. **b**, Drug-induced sleep during the day disentangles sleep pressure (that is, low) from sleep amount (that is, high), which are otherwise tightly correlated. **c**, Drug-treated larvae sleep significantly more during the day compared with the dimethyl sulfoxide (DMSO)-treated controls. **d**, During the day (from ZT5–ZT10), synapse counts increase under all control and drug conditions, except during co-administration of clonidine and 2-chloroadenosine, when synapses are significantly lost. Data are mean ± s.e.m. *n* values represent the number of neurons (top row) or fish (bottom row). For **c** and **d**, statistical analysis was performed using Kruskal–Wallis tests with post hoc Dunn's multiple-comparison test (left) and one-way ANOVA (right); not significant (NS), $P > 0.5$; *$P = 0.034$, **$P < 0.01$, ****$P < 0.0001$.

boosting adenosine signalling while inhibiting noradrenaline release with clonidine resulted in synapse loss (−0.8 synapses per hour) in tectal neurons (Fig. 4c,d and Extended Data Fig. 9c), which express both adenosine and adrenergic receptors (Extended Data Fig. 11). These results demonstrate that daytime sleep can support synapse loss under conditions of high sleep pressure and low noradrenergic tone, possibly through direct signalling events.

## Discussion

The SHY proposes that synapse numbers and strength increase during wake and decrease during sleep. By tracking synapses of single tectal neurons through sleep–wake states and circadian time, our data resolve several outstanding questions about the scale, universality and mechanisms of sleep-linked plasticity. We show that SHY-concordant dynamics of the synapse population within single neurons are present on average across many cells but, when examined on a neuron-by-neuron

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

## Methods

### Animals

Zebrafish husbandry and experiments were conducted according to UCL Fish Facility standard protocols and under project licenses PA8D4D0E5 and PP6325955 awarded to J.R., according to the UK Animal Scientific Procedures Act (1986). Embryos were kept in Petri dishes in fish water (5 mM NaCl, 0.17 mM KCl, 0.33 mM $CaCl_2$, 0.33 mM $MgSO_4$ and 0.1% methylene blue) in a 14 h–10 h light–dark cycle incubator at 28 °C. Petri dishes exposed only to fish water were cleaned with 75% ethanol, washed, soaked overnight in distilled water, air-dried and rinsed with fish water before reuse. The sex of AB/TL zebrafish larvae is not biologically determined at the early developmental stages used for these studies.

### Cloning and transgenesis

Transgene constructs that simultaneously encode FingR targeting PSD95 and membrane markers of neuronal morphology were generated using the In-Fusion HD Cloning System (Clontech). First, the GFP in a pCS2-P2A-GFP-CAAX was replaced with mKate2f by combining the linearized pCS2 (through inverse PCR; primers: 5′-GGATCTAGGACCGGGGTTTTC-3′ and 5′-GTGCTCTCCTGACCTC TAGAA-3′) with amplified mKate2f from dUAS-mKate2f (gift from the Tada laboratory, UCL) with 15 bp overhangs complementary to pCS2 site of insertion (primers: 5′-CCCGGTCCTAGATCCATGG TGAGCGAGCTGATTAAG-3′ and 5′- AGGTCAGGAGAGCACTCAGG AGAGCACACAGCAGCT-3′). Next, the template plasmid pTol2-zcUAS:PSD95.FingR-EGFP-CCR5TC-KRAB(A) (from the Bonkowsky laboratory, University of Utah; Addgene, 72638) was linearized by inverse PCR after the KRAB(A) sequence (primers: 5′-AGCCATA GAAGCAAGATTAGA-3′ and 5′- GGAGGTGTGGGAGGTTTTTTC-3′). The P2A-mKate2f sequences were then amplified with 15 bp overhangs complementary to the pTol2-zcUAS:PSD95.FingR-EGFP-CCR5TC-KRAB(A) insertion site (primers: 5′-CTTGCTTCTATGG CTGCCACGAACTTCTCTCTGTTA-3′ and 5′- ACCTCCCACACCTCCTC AGGAGAGCACACAGCAGCT-3′) and combined with the linearized FingR template.

To generate the stable *Tg(UAS:FingR(PSD95)-GFP-CCR5TC-KRAB(A)-P2A-mKate2f)* line, purified pTol2-zcUAS:PSD95.FingR-EGFP-CC R5TC-KRAB(A)-P2A-mKate2f DNA construct was sequenced to confirm gene insertion and co-injected (10 ng $\mu l^{-1}$) with emx3:Gal4FF[57] (10 ng $\mu l^{-1}$) and *tol2* transposase mRNA (100 ng $\mu l^{-1}$) at 1 nl into wild-type TL embryos at the one-cell stage. At 3 d.p.f., injected embryos were screened for mosaic expression of mKate2f, then raised to adulthood. The *tol2* transposase mRNA was in vitro transcribed from the NotI-linearized pCS-TP6287 plasmid (gift from the Wilson laboratory, UCL) using the SP6 mMESSAGE mMACHINE Kit (Ambion). RNA was purified using RNA Clean & Concentrator Kits (Zymo Research). Germline transmission was determined by mating adult fish to *nacre* mutants (*mitfa*$^{w2/w2}$, pigmentation mutants[58]) and subsequently identifying their progeny for mKate2f fluorescence, then raising to adulthood to establish a stable *Tg(UAS:FingR(PSD95)-GFP-CCR5TC-KRAB(A)-P2A-mKate2f)*$^{u541}$;*Tg(emx3:Gal4FF)*$^{u542}$ line. Owing to the negative-feedback mechanism in the system, *Tg(UAS:FingR(PSD95)-GFP-CCR5TC-KRAB(A)-P2A-mKate2f)* expression is extremely low. To increase the number of transgene copies and the level of expression in the background reporter line, the double transgenic *Tg(UAS:FingR(PSD95)-GFP-CCR5TC-KRAB(A)-P2A-mKate2f)*;*Tg(emx3:Gal4)* fish were incrossed for imaging experiments and maintained by alternating incrosses and outcrosses to *nacre* mutants.

### Whole-mount synaptic immunohistochemistry and imaging

Staining for MAGUK expression was performed using whole-mount immunohistochemistry adapted from a previous study[59]. Zebrafish larvae (2 d.p.f.) were dechorionated and fixed with 4% formaldehyde methanol-free (Pierce Thermo Fisher Scientific, 28906) in BT buffer (1.0 g sucrose, 18.75 $\mu l$ 0.2 M $CaCl_2$, topped up to 15 ml with $PO_4$ buffer (8 parts 0.1 M $NaH_2PO_4$ and 2 parts 0.1 M $Na_2HPO_4$)). To increase the signal-to-noise ratio, the fixing time was decreased to 1.5–2 h at 4 °C, although this led to softer samples. The samples were washed with $PO_4$ buffer and distilled $H_2O$ for 5 min at room temperature, then permeabilized with ice-cold 100% acetone for 5 min at −20 °C. After washing with distilled $H_2O$ and $PO_4$ buffer for 5 min each, the samples were blocked with blocking buffer containing 2% goat serum, 1% bovine serum albumin and 1% DMSO in 0.1 M PBS pH 7.4 for at least 2 h. The samples were then incubated with primary antibodies (see below for list) diluted in blocking buffer at 4 °C overnight. The embryos were washed 4–6 times for at least 20 min in blocking buffer at room temperature and incubated in secondary antibodies overnight at 4 °C. To remove unbound secondary antibodies, the embryos were washed again and transferred to glycerol in a stepwise manner up to 80% glycerol in PBS.

The primary antibodies used for staining were anti-pan-MAGUK (mouse monoclonal, K28/86, Millipore) and anti-tRFP (rabbit polyclonal, AB233, Evrogen), both at a dilution of 1:500. To avoid overamplification of signal outside of the synapse, FingR(PSD95)–GFP puncta were visualized using its own fluorescence. The following secondary antibodies were used at a dilution of 1:200: Alexa-Fluor 568 goat anti-rabbit IgG and Alexa-Fluor 633 goat anti-mouse IgG monoclonal (Life Technologies).

Confocal images were obtained using the Leica TCS SP8 system with HC PL APO ×20/0.75 IMM CS2 multi-immersion objective set to glycerol (Leica Systems). *z* stacks were obtained at 1.0 $\mu m$ depth intervals with sequential acquisition settings of 1,024 × 1,024 px. The raw images were compiled using NIH Image J (http://imagej.nih.gov/ij/). To analyse the colocalization of the puncta, maximum projections of 5–10 $\mu m$ were taken for each cell. Grey values were taken from the cross-section of the puncta using the plot-profile tool from ImageJ. Puncta grey values were normalized against the whole-stack grey value of their respective channels.

The colocalization and relationships between FingR(PSD95)–GFP and antibody staining were analysed using custom Python scripts (available at GitHub (https://github.com/anyasupp/single-neuron-synapse)). For colocalization of FingR and antibody puncta (and vice versa), the presence of puncta with maximum normalized grey value of at least 50% higher than the baseline was used. To estimate the size of the puncta, the normalized grey values were interpolated with a cubic polynomial implemented by the SciPy (v.1.11.4) function scipy.interpolate.interp1d before finding the full width at half maximum.

### Single-cell FingR(PSD95) expression using electroporation

To sparsely label single tectal cells, a FoxP2.A:Gal4FF activator plasmid (gift from M. Meyer) was electroporated into the *Tg(UAS:FingR(PSD95)-GFP-ZFC(CCR5TC)-KRAB(A)-P2A-mKate2f)*-positive larvae at 3 d.p.f according to a previously described method[33]. Anaesthetized 3 d.p.f. zebrafish larvae were mounted in 1% low-melting-point agarose (Sigma-Aldrich), perpendicular to a glass slide in a Petri dish filled with electroporation buffer (180 mM NaCl, 5 mM KCl, 1.8 mM $CaCl_2$, 5 mM HEPES, pH 7.2) with 0.02% tricaine (MS-222, Sigma-Aldrich). Excess agarose along the larval body was then removed to allow access for the electroporation electrodes. A FoxP2.A:Gal4FF construct (500 ng $\mu l^{-1}$) was injected into the midbrain ventricle together with *tol2* mRNA (20 ng $\mu l^{-1}$) and Phenol Red (~0.025%) at 5–8 nl using a micro glass needle (0.58 mm inside diameter, Sutter Instrument, BF100-58-15) pulled using a micropipette puller (Model P-87 Sutter Instrument). After injection, the positive electroporation electrode was placed lateral and slightly dorsal to the hemisphere of the target optic tectum, and the negative electrode was placed lateral and ventral to the contralateral eye. Five 5 ms trains of 85 V voltage pulses at 200 Hz were delivered through the electrodes using an SD9 stimulator (Grass Instruments). Electroporated larvae were screened for sparse, single-cell expression

of FoxP2:FingR(PSD95)$^+$ neurons using a ×20/1.0 NA water-dipping objective and an LSM 980 confocal microscope with Airyscan 2 (Zeiss) at 5–6 d.p.f.

### Repeated Imaging of FingR-labelled synapses

For synapse-tracking experiments, *Tg(UAS:FingR(PSD95)-GFP-CCR 5TC-KRAB(A)-P2A-mKate2f)* larvae that were electroporated with FoxP2.A:Gal4FF were reared at 28 °C under various light schedules. At 5–6 d.p.f., larvae were visually screened for the expression of single or sparsely labelled FoxP2.A:FingR(PSD95)$^+$ neurons in the tectum using a ×20/1.0 NA water-dipping objective and the LSM 980 confocal microscope with Airyscan 2 (Zeiss) and placed into individual wells of six-well plates (Thermo Fisher Scientific) to keep track of individual larvae and the corresponding labelled neurons, each well containing approximately 10 ml of fish water. For repeated live imaging of reticulospinal neurons, *Tg(UAS:FingR(PSD95)-GFP-CCR5TC-KRAB(A)-P2A-mKate2f)* were crossed to a *Tg(pvalb6:KALTA4)*$^{u508}$ driver line[50] (gift from the Bianco laboratory at UCL) and visually screened for larvae with a labelled reticulospinal population. For imaging FingR(PSD95)-GFP puncta, the larvae were anaesthetized with 0.02% tricaine for 5–10 min and immobilized in 1.5–2% low-melting-point agarose (Sigma-Aldrich) in fish water. The larvae were head-immobilized with the tail free and allowed to recover from anaesthesia during imaging. Imaging was performed at the appropriate zeitgeber/circadian time (ZT, where ZT0 is lights on) according to the experimental paradigm. For day–night synapse tracking, larvae were repeatedly imaged at approximately ZT0–ZT2 and ZT10–ZT12 at 7 d.p.f., 8 d.p.f. and 9 d.p.f. at 28.5 °C with the chamber lights on. For imaging performed during the dark phase (ZT14–ZT24), the temperature was kept at 28.5 °C with the chamber lights off. When immobilizing the larvae for night imaging, the handling was performed under low red light (Blackburn Local Bike Rear Light 15 Lumen; 5.2–30.5 lux, measured at the plate level). After imaging, larvae were unmounted from agarose by releasing agarose around their heads and allowing the larvae to independently swim out of the agarose. Unmounted larvae were then placed back into individual wells of six-well plates.

FingR(PSD95)$^+$ neuron image stacks were acquired using a ×20/1.0 NA water-dipping objective and the LSM 980 confocal microscope with Airyscan 2 (Zeiss). GFP and mKate2f were excited at 488 nm and 594 nm, respectively. *z* stacks were obtained at a 0.34 µm voxel depth with sequential acquisition settings of 2,024 × 2,024 px, giving a physical resolution of 0.0595376 µm in *x*, 0.0595376 µm in *y* and 0.3399999 µm in *z* and 16-bit using SR4 mode (imaging 4 pixels simultaneously). Pixel alignment and processing of the raw Airyscan stack were performed using ZEN Blue software (Zeiss).

### Locomotor activity assay

Tracking of larval zebrafish behaviour was performed as previously described[45], with slight modifications. Zebrafish larvae were raised at 28.5 °C under a 14 h–10 h light–dark (LD) cycle or constant light (LL) or switching from 14 h–10 h light–dark to constant light (free-running (FR) conditions). At 5–6 d.p.f., each FoxP2.A:FingR(PSD95)$^+$ larva was placed into individual wells of a six-well plate (Thermo Fisher Scientific) containing approximately 10 ml of fish water. The locomotor activity of some larvae was monitored using an automated video tracking system (Zebrabox, Viewpoint LifeSciences) in a temperature-regulated room (26.5 °C) and illuminated with white lights on either 14 h–10 h light–dark cycles or constant light conditions at 480–550 lux with constant infrared illumination. The larval movement was recorded using the Videotrack 'quantization' mode with the following detection parameters: detection threshold, 15; burst, 100; freeze, 3; bin size, 60 s. The locomotor assay data were analysed using custom MAT-LAB (MathWorks) scripts available at GitHub (https://github.com/JRihel/Sleep-Analysis). Any 1 min period of inactivity was defined as 1 min of sleep, according to the established convention for larval

zebrafish[40]. For experiments examining the effects of drug treatment on behaviour that did not involve live imaging, such as the clonidine dark pulse experiment (Extended Data Fig. 10d–g), 24-well (Thermo Fisher Scientific) and 96-well plates (Whatman) were used instead of the 6-well plates used for synapse imaging experiments. Sleep latency for Extended Data Fig. 9c–e was calculated using frame-by-frame data (collected at 25 fps), using code available at GitHub (https://github.com/francoiskroll/FramebyFrame).

### Sleep deprivation assay

Zebrafish larvae were raised at 28.5 °C under a 14 h–10 h light–dark cycle to 6 d.p.f., when they were video-tracked (see the 'Locomotor activity assay' section). Randomly selected 7 d.p.f. larvae were then sleep deprived for 4 h immediately after lights off from ZT14 to ZT18. Non-deprived control larvae were left undisturbed. Larvae that were individually housed in six-well plates were manually sleep deprived under dim red light (Blackburn Local Bike Rear Light 15 Lumen) by repeated gentle stimulation using a No. 1-2 paintbrush (Daler-Rowney Graduate Brush) to prevent larvae from being immobile for longer than 1 min. For most stimulations, this required only putting the paintbrush into the water; if the larvae remained immobile, they were gently touched. The 4 h SD protocol was performed by experimenters in 2 h shifts. All sleep deprived and control larvae were imaged at around ZT14 and ZT18 on 7 d.p.f. and again at ZT0 on 8 d.p.f. (see the 'Repeated imaging of FingR-labelled synapses' section).

### Drug exposure for live imaging

*Tg(UAS:FingR(PSD95)-GFP-CCR5TC-KRAB(A)-P2A-mKate2f)* larvae that had been electroporated with FoxP2.A:Gal4FF (see the 'Single-cell FingR(PSD95) expression using electroporation' section) were kept under a 14 h–10 h light–dark cycle until 7 d.p.f., then imaged at ZT4–ZT5 (see the 'Repeated imaging of FingR-labelled synapses' section). Larvae were transferred to individual wells of a six-well plate containing 10 ml of sleep-promoting drugs, alone or in combination, as follows: 30 µM melatonin (M5250, Sigma-Aldrich) in 0.02% DMSO; 30 µM of clonidine hydrochloride (C7897, Sigma-Aldrich) in 0.02% DMSO; 45 µM 2-chloroadenosine (C5134, Sigma-Aldrich) in 0.02% DMSO; and 0.02% DMSO in fish water as controls[45,52,60,61]. Combinations of drugs were applied at the same concentrations as the single-dose conditions, maintaining the final DMSO concentration of 0.02%. Sleep induction was monitored with video-tracking (see the 'Locomotor activity assay' section) for 5 h, after which the drugs were removed by 2–3 careful replacements of the fish water using a transfer pipette followed by transferring the larvae individually to a new six-well plate with fresh water. The larvae were then reimaged using the Airyscan system (see the 'Repeated imaging of FingR-labelled synapses' section).

### Tectal cell segmentation and clustering

The morphology of tectal neurons at 7 d.p.f. was segmented and measured using Imaris v.8.0.2 (Bitplane) and ImageJ (NIH). The total filament length for each neuron was obtained using the Imaris Filament function. The anterior–posterior span of the distal arbour was calculated using the Measurement function at an orthogonal view in 3D. The relative proximal arbour locations were calculated by dividing the proximal arbour distance from the nucleus by the total length of the neuron obtained using Filament function of Imaris. The distance from the skin, distal arbour thickness and distal arbour to skin distance were obtained using the rectangle Plot_Profile tool of ImageJ at an orthogonal view of the neuron to calculate the fluorescence intensity across the tectal depth. The intensity profiles were then analysed using custom Python scripts to obtain the maximum width using area under the curve functions following published methods[33,43].

Additional clustering and statistical analyses were performed using custom scripts written in Python (available at GitHub (https://github.com/anyasupp/single-neuron-synapse)). For segmentation clustering,

six morphological features of FoxP2.A cells were standardized and reduced in dimensionality by projecting into principal component analysis space. The first four components, which explained 89% of the variance, were selected to use for clustering. These components were then clustered using $k$-means clustering with $k$ ranging from 1 to 11. Using the elbow method, Calinski Harabasz coefficient and silhouette coefficient, we found $k = 4$ to be the optimal number of $k$ clusters.

## Puncta quantification and statistics

All image files of synapse tracking experiments were blinded by an independent researcher before segmentation and puncta quantification. To count the number of FingR(PSD95)–GFP puncta, each neuron's morphology was first segmented using the Filament function in Imaris v.8.0.2 (Bitplane). FingR(PSD95)–GFP puncta were labelled using the Spots function, thresholded using the Quality classification function at approximately 130–200 depending on the image file. The number and location of GFP puncta were also manually checked for accuracy. FingR(PSD95)–GFP puncta lying on the FingR+ neuron (mKate2f red channel) were extracted using the Find Spots Close to Filament XTension add-on in IMARIS.

The percentage changes in synapse number and intensity were calculated using the following formula:

$$\Delta(\%) = \left(\frac{x - x_{t-1}}{x_{t-1}}\right) \times 100,$$

Where $x$ represents either synapse number or intensity and $x_{t-1}$ is the respective synapse number or intensity at the previous timepoint. Statistical tests were implemented using Python[62]. Values in the figures represent the average ± 68% CI unless stated otherwise.

Synapse intensity was calculated using the ratio of the normalized average FingR(PSD95)–GFP intensity and mKate2f, to account for depth-dependent signal reduction[63]. First, the average FingR(PSD95)–GFP and mKate2f (cell morphology) intensities at the same location within the neuron were extracted using the Imaris Spots function. Next, these average intensity values were normalized to their respective channel maximum and minimum value to account for larval position inconsistencies between imaging as follows:

$$\text{Normalized mean intensity} = \frac{\text{Average intensity} - \text{Channel}_{min}}{\text{Channel}_{max} - \text{Channel}_{min}}.$$

Depth-dependent signal reduction was corrected by calculating the FingR(PSD95)–GFP:mKate2f ratio as follows:

$$\text{Normalized mean puncta intensity} = \frac{\text{Normalized mean GFP}}{\text{Normalized mean mKate2f}}.$$

Before statistical analysis, all datasets were tested for normality using the Shapiro–Wilk test followed by direct visual inspection of $Q$–$Q$ plots. For repeated-measures design, the data were first tested for sphericity using Mauchly's test; repeated-measures or mixed ANOVAs were then performed, corrected with Greenhouse–Geisser correction when sphericity was violated, followed by post hoc $t$-tests corrected with Benjamini–Hochberg correction for multiple comparisons. For multiple-sample comparisons, equal variances were tested using Levene's tests. If variances were equal, either one-way ANOVA (multiple groups) with post hoc Benjamini–Hochberg correction or Student's $t$-tests (two groups) were performed to test for significant differences. If variances were unequal, Kruskal–Wallis (multiple groups) with Dunn's multiple-comparison correction or Mann–Whitney $U$-tests (two groups) were performed to test for significant differences. All of the statistical analyses performed are provided in Supplementary Data 1.

## *per3* circadian rhythm bioluminescence assay

Larvae (6 d.p.f.) from a *Tg(per3:luc)$^{g1}$;Tg(elavl3:EGFP)$^{knu3}$* incross were individually placed into wells of 24-well plates in water containing 0.5 mM beetle luciferin (Promega). From ZT14 (the light to dark transition) the next day, half of the larvae were subjected to a sleep deprivation paradigm (see the 'Sleep deprivation assay' section) under dim red light, while the others were left undisturbed in similar lighting conditions. At the end of the 4 h sleep deprivation period, the larvae were individually transferred to the wells of a white-walled 96-round-well plate (Greiner Bio-One) and sealed with an oxygen-permeable plate-seal (Applied Biosystems). Bioluminescence photon counts, reflecting luciferase expression driven by the *per3* promoter, were sampled every 10 min for three consecutive days, in constant dark at 28 °C, using the TopCount NXT scintillation counter (Packard).

## HCR fluorescence in situ hybridization

FoxP2.A neurons were sparsely labelled with GFP by co-electroporating wild-type AB larvae with FoxP2.A:Gal4FF and UAS:eGFP[1] at 500 ng µl$^{-1}$ each (see the 'Single-cell FingR(PSD95) expression using electroporation' section). Whole-mount hybridization chain reaction (HCR) was performed on larvae with FoxP2.A neurons positive for GFP at 7 d.p.f. using an adapted protocol from a previous study[64]. In brief, larvae were fixed with 4% PFA and 4% sucrose overnight at 4 °C. The next day, the larvae were washed with PBS to stop fixation and the brains were removed by dissection. The dissected specimens were permeabilized using proteinase K (30 µg ml$^{-1}$) for 20 min at room temperature, then washed twice in PBS with 0.1% Tween-20 (PBST), before being post-fixed in 4% PFA for 20 min at room temperature. The larvae were then washed in 0.1% PBST and prehybridized with prewarmed HCR hybridization buffer (Molecular Instruments) for 30 min at 37 °C.

Probes targeting multiple genes associated with different types of adenosine or adrenergic receptors were combined and labelled to the same hairpins. For example, probes detecting *adora1a-b* (encoding adenosine receptor A1a and A1b) contain initiators that correspond with hairpins (B3) labelled with Alexa 546 fluorophore, whereas *adora2aa*, *adora2ab* and *adora2b* (encoding adenosine receptors A2aa, A2ab and A2b) contain initiators that correspond with hairpins (B5) labelled with Alexa 647 fluorophore (Supplementary Data 2). Probe solutions consisting of cocktails of HCR probes for each transcript (Thermo Fisher Scientific) were prepared with a final concentration of 24 nM per HCR probe in HCR hybridization buffer. The larvae were then incubated in probe solutions overnight at 37 °C. Excess probes were removed by washing larvae four times for 15 min with probe wash buffer (Molecular Instruments) at 37 °C followed by two 5 min washes of 5× SSCT buffer (5× sodium chloride sodium citrate and 0.1% Tween-20) at room temperature. Preamplification was performed by incubating the samples with amplification buffer (Molecular Instruments) for 30 min at room temperature. Hairpin h1 and hairpin h2 were prepared separately by snap-cooling 4 µl of 3 µM stock at 95 °C for 20 min and 20 °C for 20 min. The larvae were then incubated with h1 and h2 hairpins in 200 µL amplification buffer overnight in the dark at room temperature. Excess hairpins were washed thoroughly the next day twice for 5 min and three times for 30 min with 5× SSCT at room temperature. The specimens were then imaged using a ×20 water-immersion objective and the LSM 980 confocal microscope with Airyscan 2 (Zeiss). The endogenous GFP signal from FoxP2.A was visualized without amplification.

## Reporting summary

Further information on research design is available in the Nature Portfolio Reporting Summary linked to this article.

## Data availability

The data are available at GitHub (https://github.com/anyasupp/single-neuron-synapse)[65]. Source data are provided with this paper.

## Code availability

The code used to generate figures in this manuscript can be found at GitHub (https://github.com/anyasupp/single-neuron-synapse)[65]. The sleep analysis code is available at GitHub (https://github.com/JRihel/Sleep-Analysis)[66]. The frame by frame analysis code can be found at GitHub (https://github.com/francoiskroll/FramebyFrame)[67].

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

**Acknowledgements** We thank all of the current and past members of the Rihel laboratory for discussions and feedback on this project and the members of the zebrafish community for sharing protocols and reagents; S. Lim for her assistance in blinding experimental files; A. Gilbert for help with early SD experiments; L. Sheets for her guidance on synapse immunohistochemistry; L. Elias for her guidance on gene expression; N. Nikolaou for sharing his knowledge of FoxP2.A neurons; C. Trivedi for his help with designing HCR probes; and the staff at the UCL Fish Facility for fish husbandry and UCL Imaging Facility for their expertise. This work was supported by UCL Research Scholarship (to A.S.), an EMBO Fellowship awarded to D.G.L. (ALTF 1097-2016), a Medical Research Council studentship (MR/W006774/1 to E.B.), a European Research Council Starting Grant (282027 to J.R.) and a Wellcome Trust Investigator Award (217150/Z/19/Z to J.R.).

**Author contributions** J.R. and A.S. conceived and designed all of the experiments with input from D.G.L. A.S. performed all of the experiments with help from D.G.L. (circadian clock/dark pulse and SD), E.B. (HCR) and J.R. (SD). A.S., D.G.L. and J.R. wrote the manuscript with input from E.B.

**Competing interests** The authors declare no competing interests.

**Additional information**
**Correspondence and requests for materials** should be addressed to Jason Rihel.

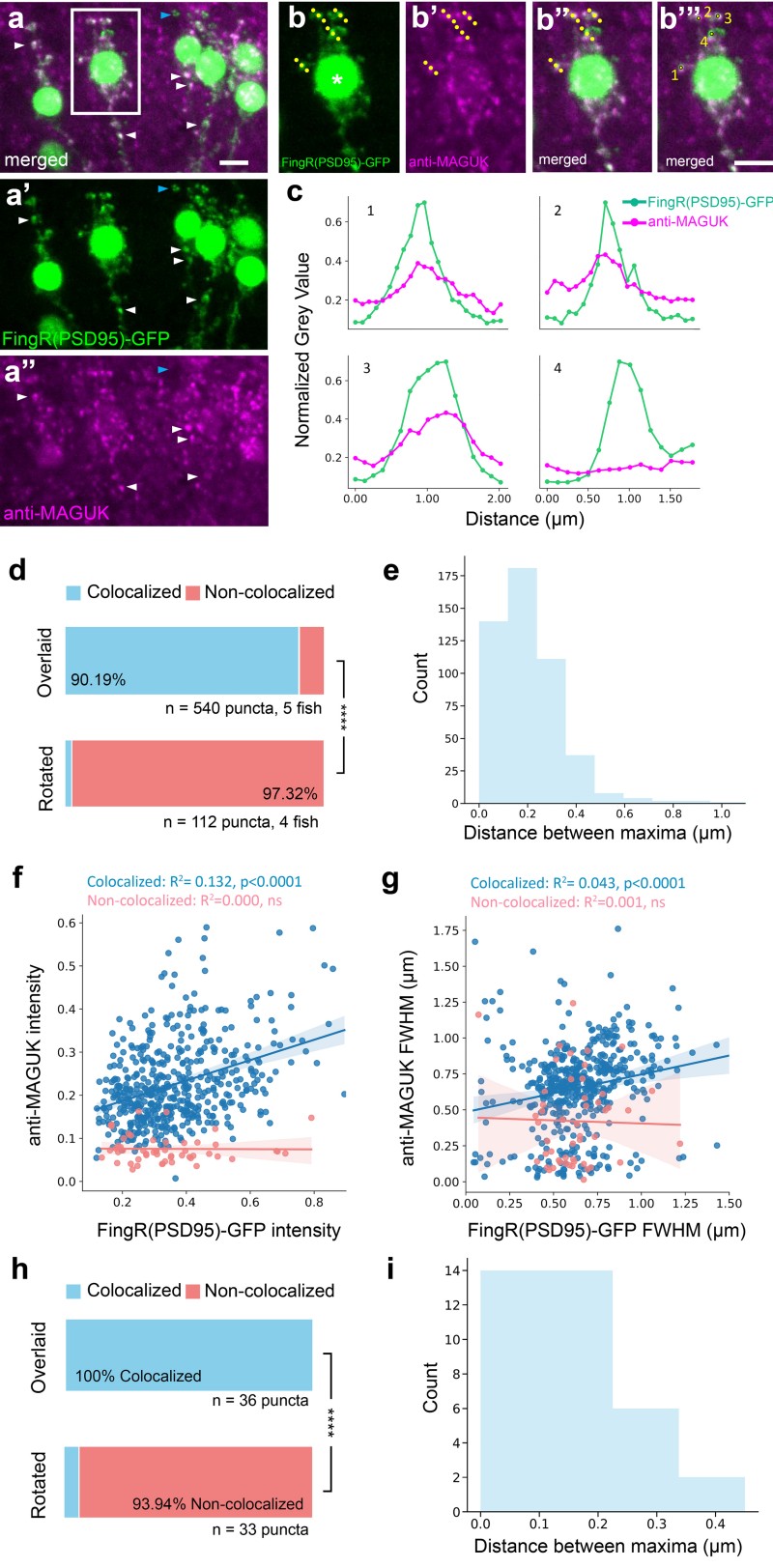

**Extended Data Fig. 1** | See next page for caption.

**Extended Data Fig. 1 | The modified FingR(PSD95)-GFP construct labels synapses in vivo. a-a''**, Maximum projection (Z-stack, ~10 μm) of anti-MAGUK immunohistochemistry and endogenous fluorescence of FingR(PSD95)-GFP in the spinal cord of 2 dpf *Tg(mnx1:Gal4)* larvae. Examples of FingR(PSD95)+ puncta co-labelled by anti-MAGUK are indicated by white arrowheads; an example of a FingR(PSD95)+ not labelled by anti-MAGUK is indicated by the blue arrowhead. **b-b'''**, Higher magnification (white box from **a**) depicting how sectional grey values for each synapse were obtained. **b**, The FingR(PSD95)-GFP channel showing part of a neuron with its nucleus (asterisk) and synaptic puncta (green). Dotted lines indicate example cross-sectional areas obtained for each synapse. **b'**, Anti-MAGUK puncta of the same neuron. **b'',b'''**, FingR(PSD95)-GFP and MAGUK channels merged, with examples of cross-sections 1–4. **c**, Examples of normalized cross-sectional grey values for anti-MAGUK signals and FingR(PSD95)-GFP signal for the same puncta (numbered 1–4 in **b'''**). Three examples in which FingR(PSD-95)-GFP co-localized with anti-MAGUK signals (#1–3) and one example (#4) where a FingR(PSD-95)-GFP punctum did not co-localize with MAGUK. See Methods for details. **d**, Percentage of FingR(PSD-95)-GFP synapses that co-localized with anti-MAGUK+ puncta (blue). As a control for chance co-localization, the calculation was repeated on images in which the anti-MAGUK image was rotated by 90° relative to the FingR(PSD-95)-GFP channel. ****P = 1.1 × 10$^{-83}$ Chi-square. **e**, Histogram of the distance between all co-localized FingR(PSD95)-GFP and anti-MAGUK cross-sectional grey value peaks. **f-g**, The intensity and Full Width Half Max (FWHM) of FingR(PSD95)-GFP and anti-MAGUK puncta are weakly, but significantly, positively correlated. Blue and red lines depict the linear regression curve and 95% CI for the colocalized and non-colocalized populations, respectively. n = 540 puncta, 5 fish (data as in **d**). **h**, Percentage of anti-MAGUK+ puncta that co-localized with FingR(PSD-95)-GFP synapses (blue). As a control for chance co-localization, the calculation was repeated on images in which the FingR(PSD-95)-GFP image was rotated by 90° relative to the anti-MAGUK channel. ****P = 3.1 × 10$^{-14}$ Chi-square. **i**, Histogram of the distance between co-localized anti-MAGUK and FingR(PSD95)-GFP cross-sectional grey value peaks. Scale bar: 5 μm (**a-b'''**).

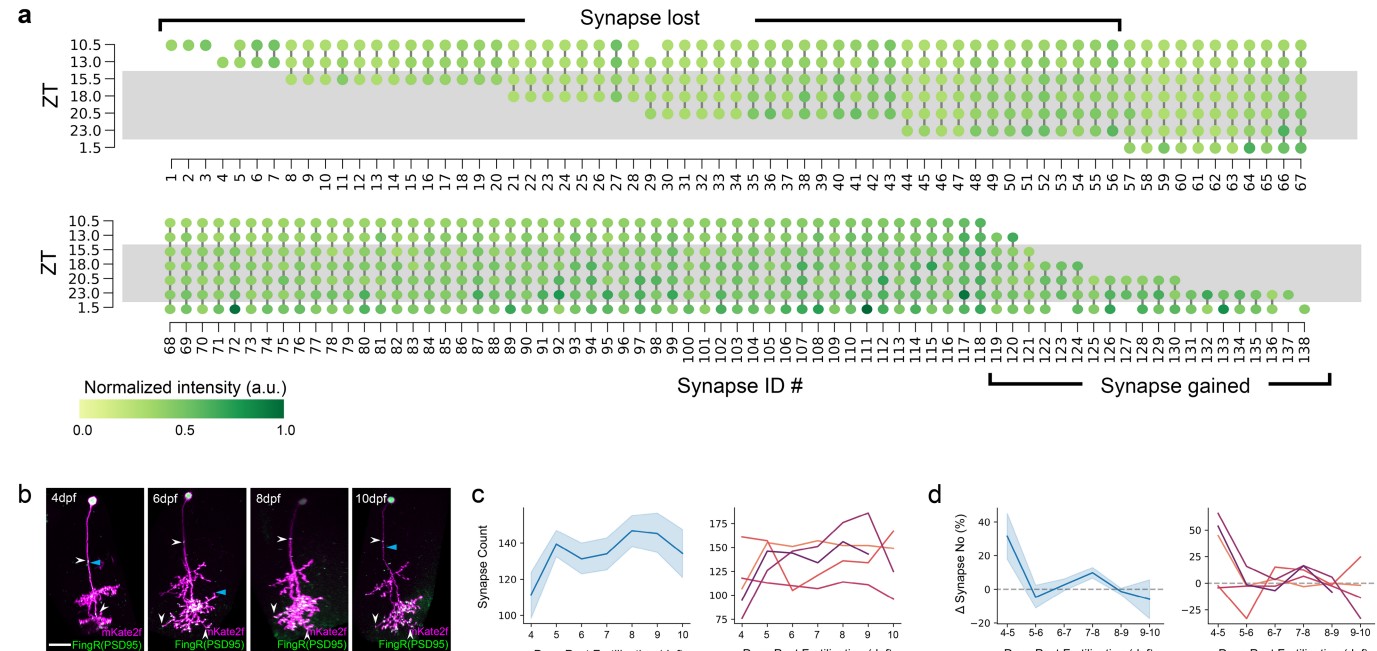

**Extended Data Fig. 2 | The synapse number of single tectal neurons is developmentally stable at 6–9 dpf. a**, The full map of synapse tracking from the neuron in Fig. 1c. Each column depicts a synapse, and the colour indicates the normalized GFP intensity of each synapse. In this example, 56 synapses disappeared and 20 synapses appeared during the imaging, resulting in a net change of −36 synapses. Grey bars depict night (ZT14-24). **b**, Example of a single FoxP2.A:FingR(PSD95)+ neuron imaged through development from 4–10 dpf. Nuclei and synapses are FingR(PSD95)-GFP+ (green), and cellular morphology

is labelled by mKate2f (magenta). White arrowheads indicate examples of puncta that persisted through time. Blue arrowheads indicate examples of synapses gained/lost through time. **c**, Synapse counts across all neurons (average and 68% CI) (**left**) and for single neurons through 4–10 dpf (**right**). **d**, Average percentage change in synapse number and 68% CI calculated from the previous time point (**left**) and for each neuron (**right**). The percentage change in synapse number across time is close to zero between 6–9 dpf. n = 5 cells, 5 larvae. Scale bar: 15 μm (**b**).

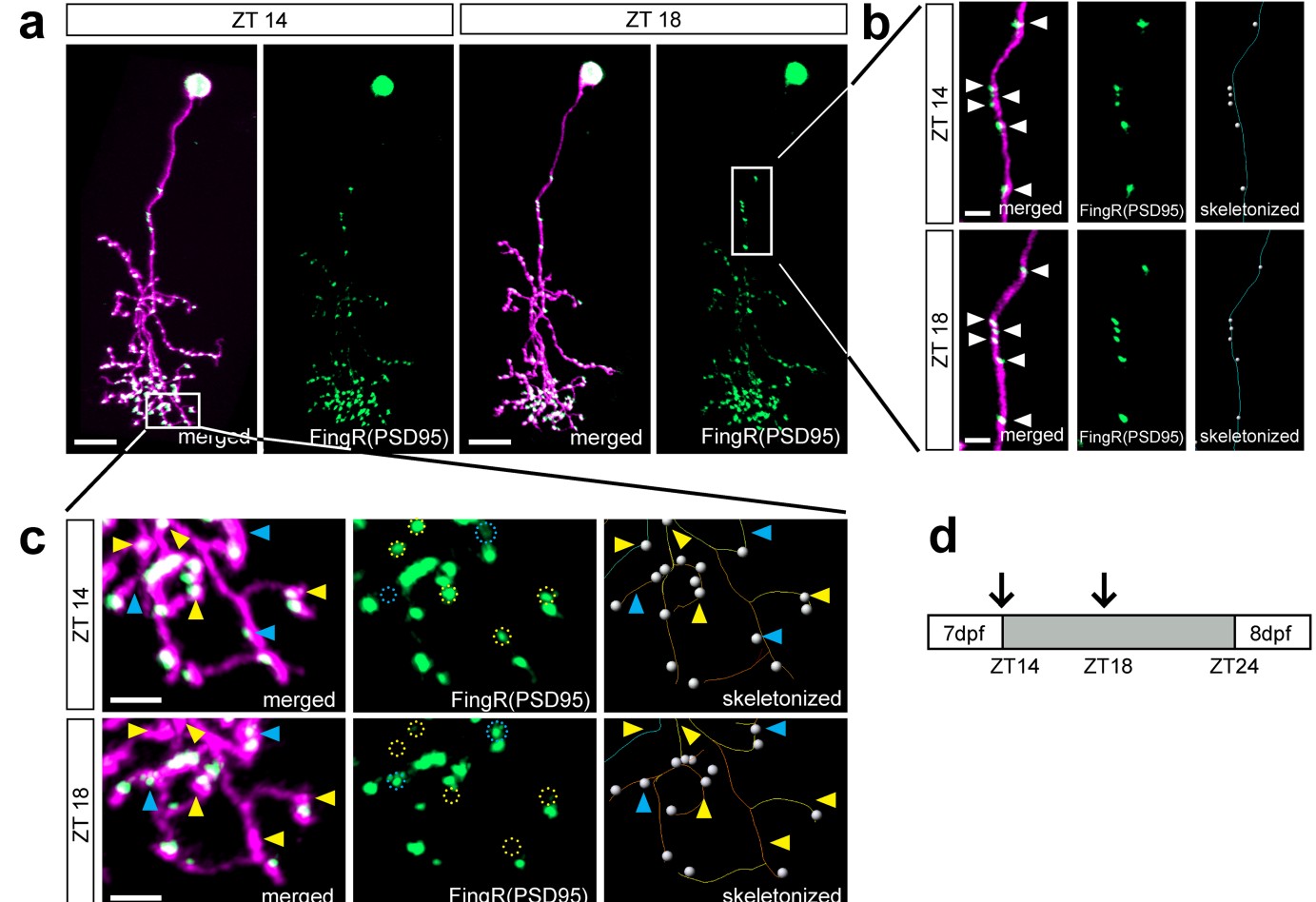

**Extended Data Fig. 3 | Example of a single FoxP2.A:FingR(PSD95)+ neuron at ZT14 and ZT18. a**, A single FoxP2.A:FingR(PSD95)+ tectal neuron imaged at ZT14 and ZT18. Nuclei and synapses are FingR(PSD95)-GFP+ (green), and cellular morphology is labelled by mKate2f (magenta). **b**, Higher magnification of the primary dendrite segment (white box in **a**). Right panels show semi-automatic skeletonization (lines) of neurites and detection of FingR(PSD95)-GFP puncta (grey spheres, **Methods**). **c**, Higher magnification of a section of the distal arbour (white box in **a**). FingR(PSD95)-GFP+ puncta that appeared (blue circles and arrowheads) and disappeared (yellow circles and arrowheads) between ZT14 and ZT18 can be observed. **d**, Schematic showing imaging times (black arrows) at ZT14 and ZT18 on the night of 7 dpf. Scale bars: 10 μm (**a**) and 2.5 μm (**b,c**).

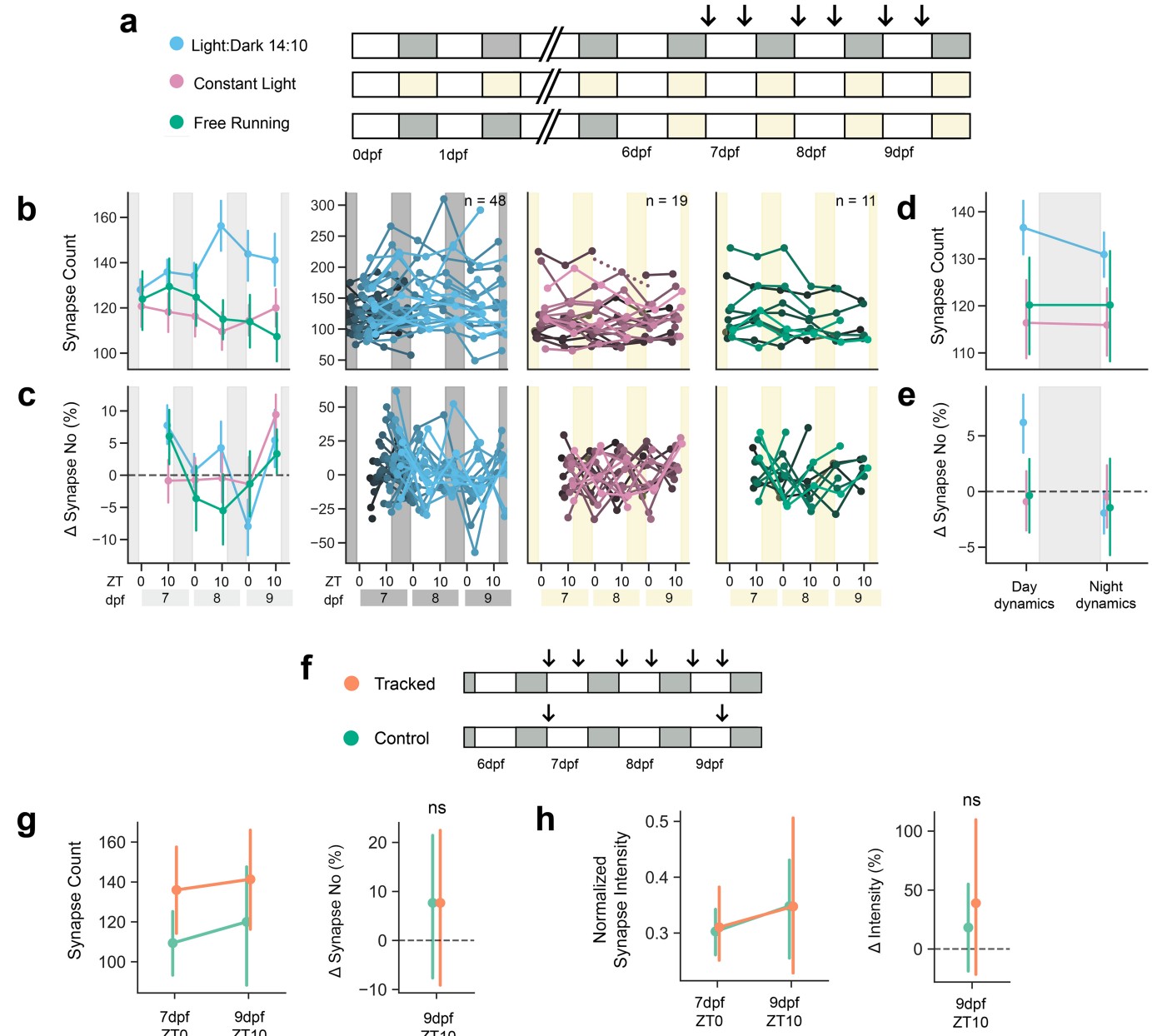

**Extended Data Fig. 4 | Extended tracking of single neurons over multiple days. a**, Larvae were raised on 14h–10h LD cycles (blue), on constant light (pink), or switched from LD to LL at 6 dpf ('free running', FR, green) and repeatedly imaged (arrows) at ZT0 and ZT10 for each day from 7–9 dpf. **b-c**, The average (68%CI) (**b**) and percentage change (**c**) for synapse counts at each timepoint in LD (blue), LL (pink), or FR (green) conditions from 7–9 dpf (**left**). Each n = neuron is plotted as a single line (**right**). **d-e**, Average synapse counts and percentage change (68%CI) for ZT0 and ZT10 combined across all tracked days for each lighting condition (LD, 13 independent experiments; LL, 4 experiments, and FR, 4 experiments). The ZT10 timepoint from 9 dpf was excluded to avoid interference from a new developmental round of synaptogenesis. **f**, Schematic

of experiment to test whether repeated imaging affected synapse number and strength measurements. Larvae raised in LD (indicated by white and grey boxes) were either imaged six times between 7–9 dpf at ZT0 and ZT10 (Tracked, orange) or imaged at ZT0 on 7 dpf and ZT10 on 9 dpf (Control, green). **g-h**, Average (with 68%CI) synapse counts (**g**) and normalized average synapse intensity (**h**) at the first and last time point (7 dpf ZT0 and 9 dpf ZT10) for tracked and control larvae (**left**). The percentage changes in synapse number (**g**, **right**) and average synapse intensity (**h**, **right**) were not statistically different between tracked and control larvae. Controls: n = 6 neurons, 4 larvae; Tracked: n = 14 neurons, 14 larvae collected over 8 independent experiments. ns, P > 0.05 Student's t-test, two tailed.

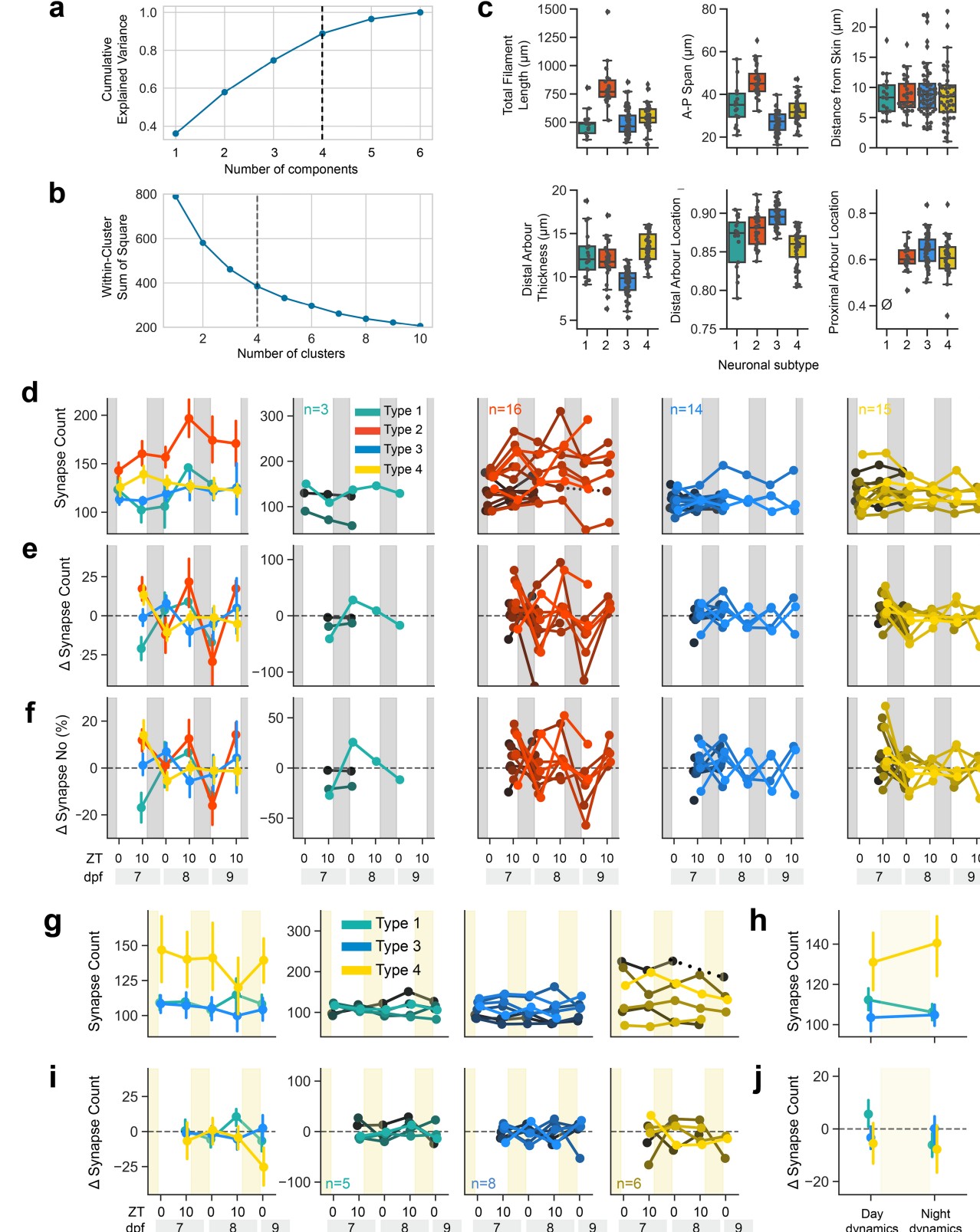

**Extended Data Fig. 5** | See next page for caption.

**Extended Data Fig. 5 | FoxP2.A tectal neurons have four morphological subtypes. a**, Principal component analysis using the subtype morphological features depicted in Fig. 2a. Four principal components (dotted line) account for >85% of the variance. **b**, The optimal number of clusters for k-means clustering was determined using the elbow method by plotting the within-cluster sum of squares. Four clusters were chosen (dotted line). **c**, The six features used to cluster FoxP2.A neurons (collected over 26 experiments) by morphological subtype. Boxes depict the median and interquartile range and the whiskers represent the distribution for each parameter. The slashed zero means the feature is absent. **d-f** (**left**), Synapse counts with 68%CI (**d**), average change (68%CI) in synapse counts (**e**), and percentage change (68%CI) in synapse counts (**f**) in different FoxP2.A tectal neuron subtypes of larvae raised in normal LD conditions. **d-f** (**right**), Each neuron is plotted, grouped by subtype. **g**, Average (68%CI) synapse counts of tectal subtypes (**left**) and for each n= neuron (**right**) across multiple days under clock-break (LL) conditions. Note the lack of Type 2 neurons in LL. **h**, Average (68%CI) synapse counts during the subjective day or night under clock-break conditions. **i**, Average change (68%CI) in synapse counts (**left**) and single neurons (**right**) across multiple days under clock-break conditions, sorted by tectal subtype. **j**, the average change (68%CI) in synapse counts for the subjective day and night under clock-break conditions. Data in **g-j** are from 4 independent experiments.

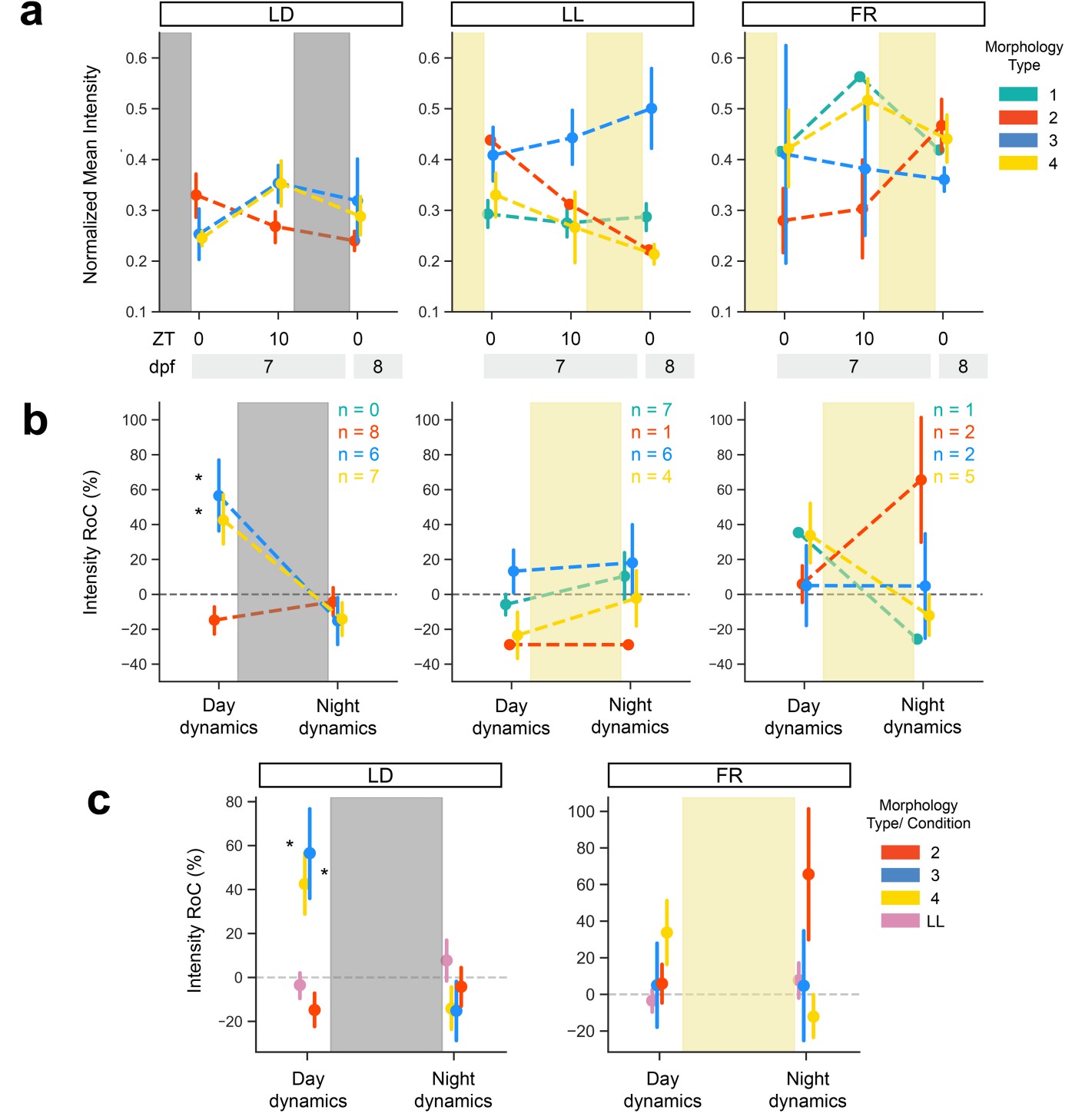

**Extended Data Fig. 6 | FingR(PSD95):GFP signal intensity increases during the day and decreases at night in some, but not all tectal subtypes.** **a**, Average and 68% CI of normalized synapse intensity on LD, LL, and FR conditions across one day and night for a subset of tectal neurons from Fig. 2 imaged under identical microscopy settings to enable intensity measurements. Note that the loss of the circadian clock alters the relative abundance of Type 1 and Type 2 neurons. **b**, Percentage change (mean and 68% CI) in normalized synapse intensity calculated as in Fig. 1. Compared to Type 2 neurons, Type 3

(p = 0.026; g = 1.777) and Type 4 (p = 0.026; g = 1.651) neurons have increased synapse intensities during the day (mixed ANOVA, interaction (subtype*time) p = 0.03, post-hoc Benjamini-Hochberg, one tailed). **c**, Both Type 3 (p = 0.026; g = 1.691) and Type 4 (p = 0.026; g = 1.408) neurons have significantly increased synapse intensities (with 68% CI) during the day relative to clock-break (LL) conditions (mixed ANOVA, interaction (condition*time) p = 0.006, post-hoc Benjamini-Hochberg, one tailed). Data are collected from 8 independent LD, 4 LL, and 4 FR experiments.

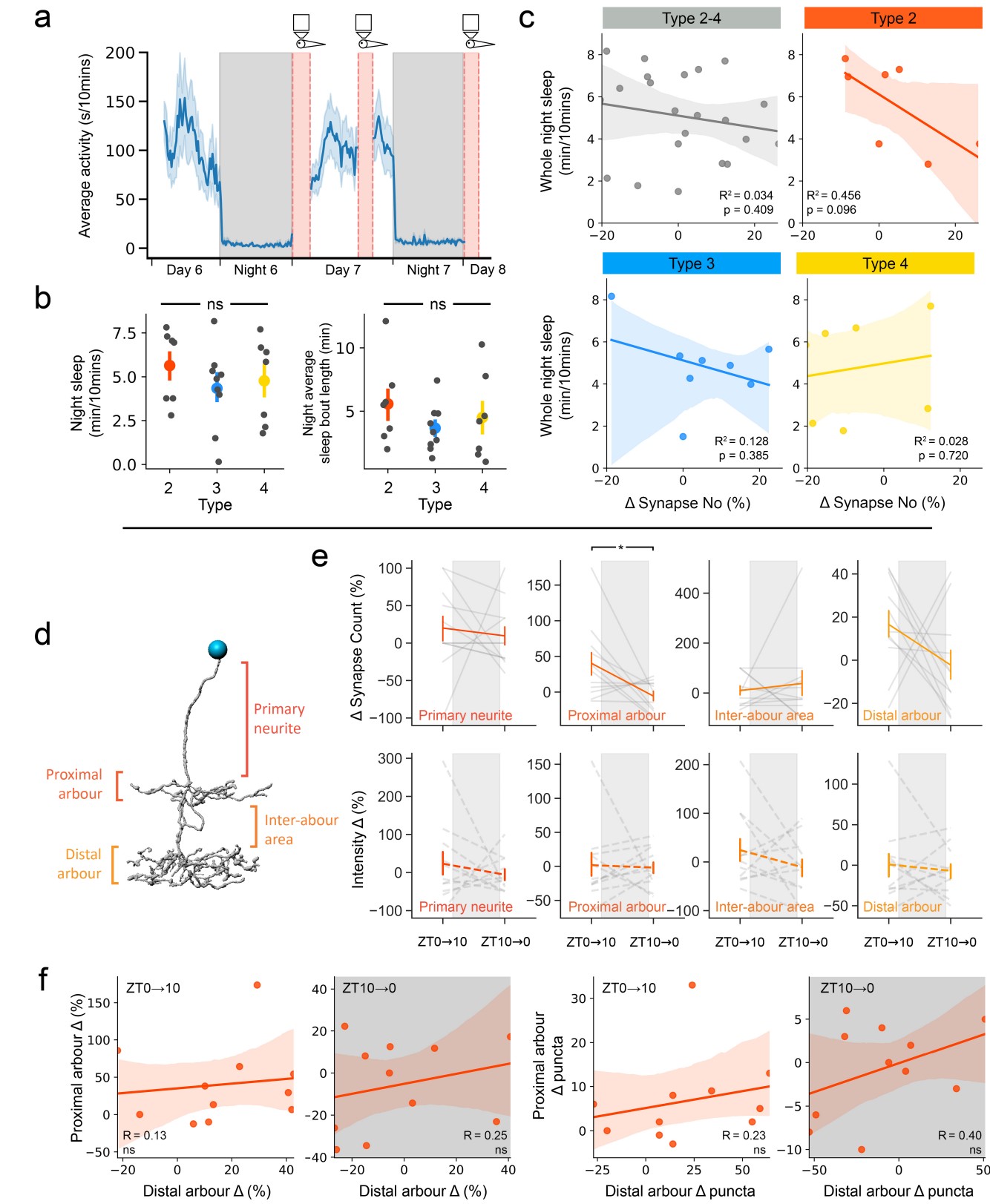

**Extended Data Fig. 7** | See next page for caption.

**Extended Data Fig. 7 | Tectal subtype labelling does not bias larval sleep amount and sleep-wake states have non-uniform effects on synapses within neuronal compartments. a**, Schematic of behavioural and synapse tracking experiment set up. Larval locomotor behaviour was tracked on a 14 h–10 h LD cycle from 6–8 dpf. The average activity ( ± 95% CI) of 10 example larvae are plotted across two days and nights. Larvae were removed from the tracking arena and imaged at lights on (ZT0) and again at ZT10 (dotted red bars). White and grey boxes indicate day and night periods, respectively. **b**, 7 dpf Larvae had similar levels of sleep and sleep bout lengths at night ( ± SEM) regardless of the FoxP2.A tectal neurons subtype labelled in each larva (ns, p > 0.05, Kruskal-Wallis; 5 independent experiments). **c**, For each neuron/larva, the average percentage change of synapse number is plotted versus the average 7 dpf night-time sleep. **d**, Type 2 tectal neurons were divided into four segments: the primary neurite, proximal arbour, inter-arbour area, and distal arbour. **e**, The average and 68% CI of synapse number and intensity dynamics within each of the four segments. Grey lines represent segments from individual neurons. *P = 0.037, repeated-measures ANOVA with Greenhouse-Geisser correction. **f**, Proximal and distal arbours synapse number dynamics are not correlated. The relationship between the absolute and relative (%) synapse number change of the proximal and distal arbours of individual Type 2 neurons during the day and night phase. Linear regressions in **c** and **f** are fitted with 95% CI.

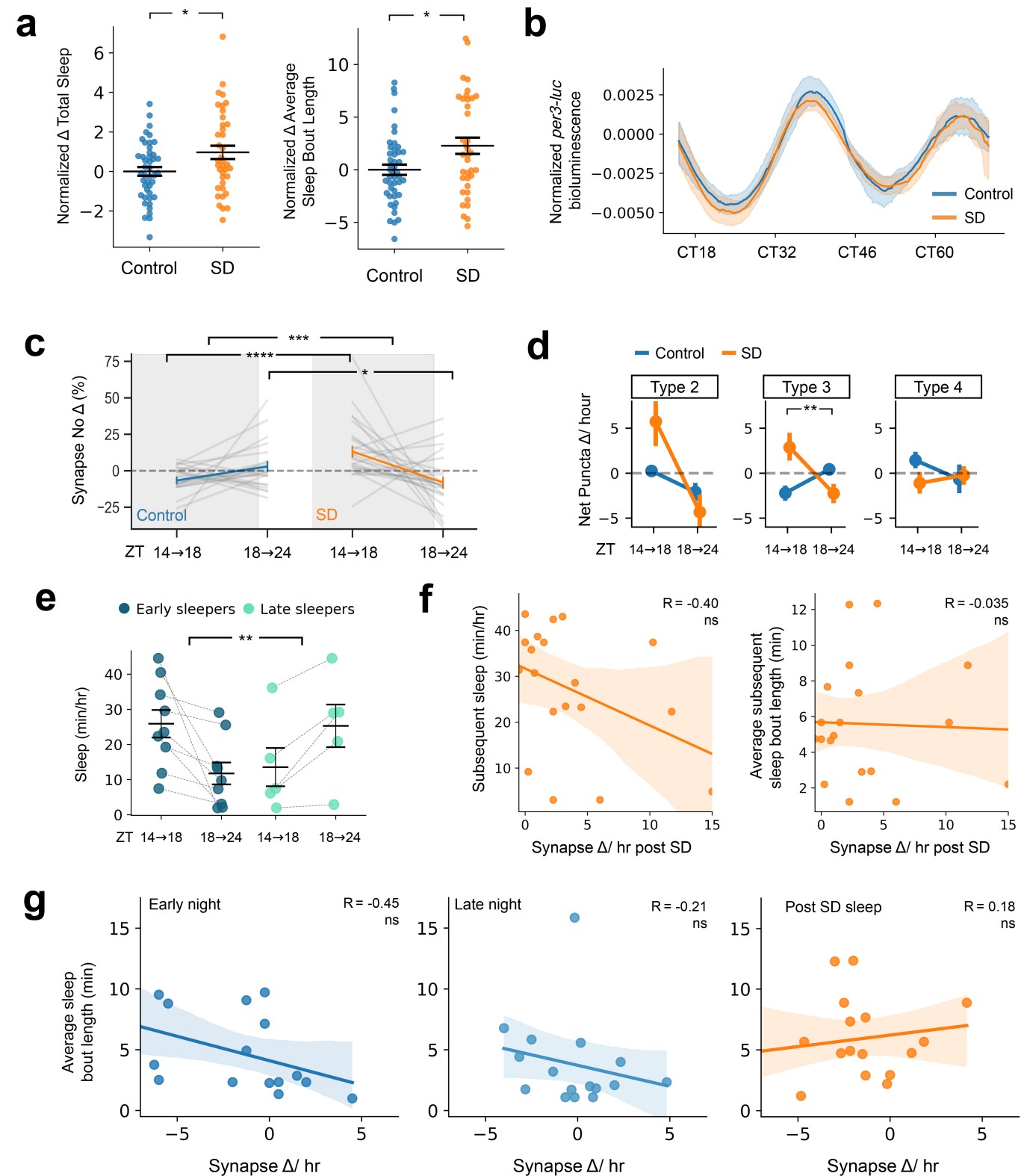

**Extended Data Fig. 8** | See next page for caption.

**Extended Data Fig. 8 | Sleep deprivation affects synapse number in tectal neuron subtypes. a**, Percentage change of total sleep (**left**) and average sleep bout length (**right**) of each larva (dots) in the 6 hr post SD (ZT18-24, 7dpf), normalized to the circadian-matched time at 6 dpf. The black lines depict the average ± SEM. *P < 0.02, one-way ANOVA. **b**, The SD method did not alter circadian clock phase as measured by the bioluminescence driven by a *Tg(per3-luc)* reporter line for the clock gene *per3* expression. The detrended *per3* bioluminescence rhythms ( ± 95%CI) remained in phase for both SD (n = 14 larvae) and control (n = 12) larvae over multiple days of constant dark conditions. Circadian time (CT = 0 last lights ON transition). **c**, The percentage change in synapse number within each neuron between imaging sessions at ZT14 and ZT18, and between imaging at ZT18 and ZT24. **d**, Average (68%CI) for net synapse change per hour for FoxP2.A tectal subtypes in control or sleep deprived larvae. Type 3, but not Type 4 neurons significantly gain synapses after SD (Mixed ANOVA, post-hoc Benjamini-Hochberg, one tailed **p = 0.01, g = 1.266) and subsequently lose them (p = 0.014, g = −1.034) relative to controls. Type 2 lacks enough matched controls to assess. **e**, Sleep amount for early and late sleepers in the early (ZT14-18) and late (ZT18-24) phase of the night (5 independent experiments). The black lines depict the average ± SEM. **f**, For each neuron/larva, changes in synapse number during extended wakefulness did not correlate with either the subsequent total sleep or average sleep bout lengths (mean ± 95% CI). **g**, Changes in synapse numbers for each neuron/larva did not significantly correlate with the average sleep bout lengths during the early and late night of controls, or after SD (mean ± 95% CI). *P ≤ 0.05, **P ≤ 0.01, ***P ≤ 0.001, ****P ≤ 0.0001, Mixed ANOVA interaction (condition*time), post-hoc Benjamini-Hochberg, two tailed.

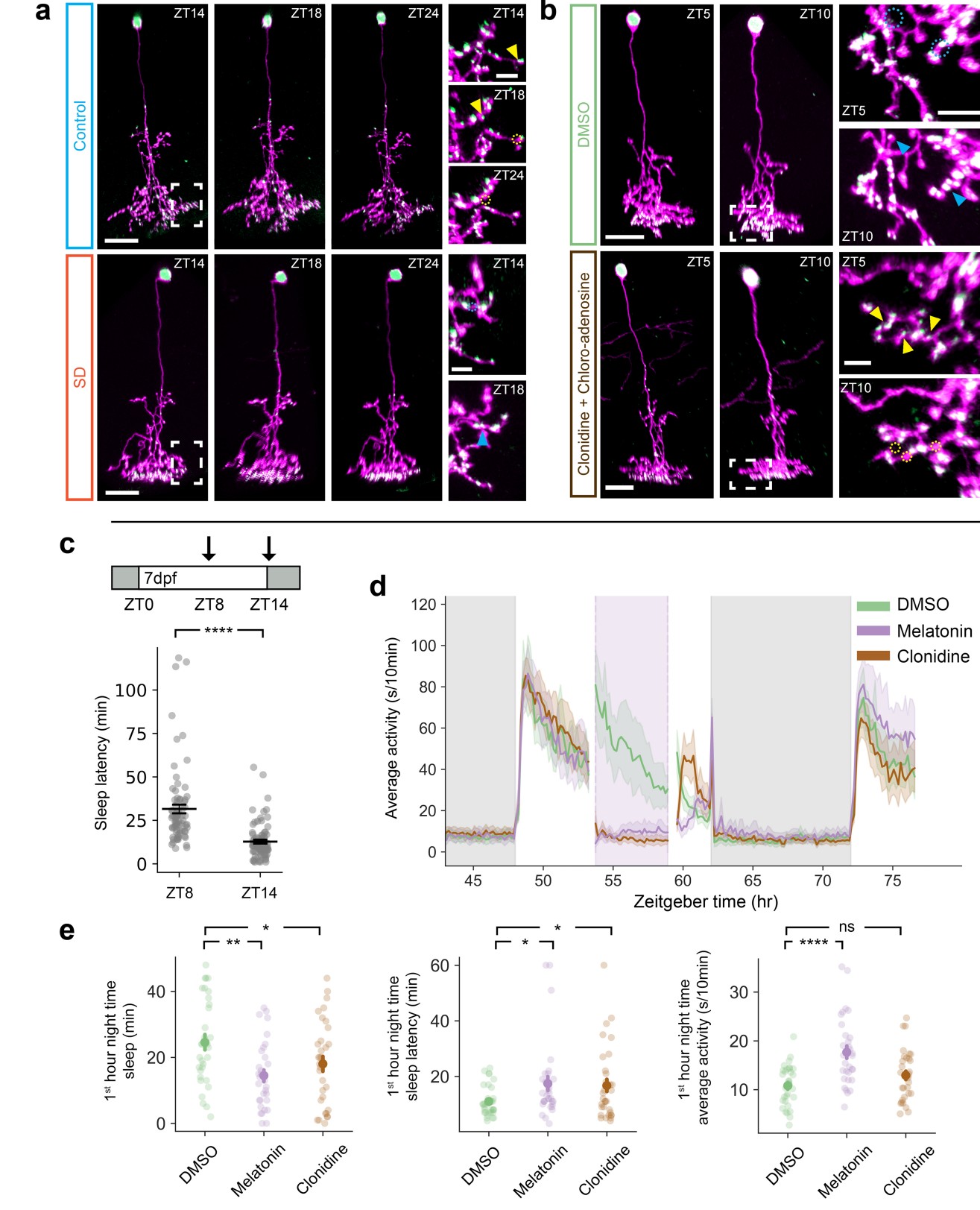

**Extended Data Fig. 9** | See next page for caption.

**Extended Data Fig. 9 | Examples of manipulated single FoxP2.A:FingR(PSD95)+ neurons and clonidine and evidence that daytime drug treatment reduced sleep the following night. a**, **left** Example FoxP2.A:FingR(PSD95)+ tectal neurons imaged before (ZT14), immediately after (ZT18), and 6 h after (ZT24) sleep deprivation and control. Nuclei and synapses are FingR(PSD95)-GFP+ (green), and cellular morphology is labelled by mKate2f (magenta). **Right**, Higher magnification (dotted white box) showing the same dendritic segments at each time point, with examples of synapses lost (yellow arrows and dotted circles) or gained (blue arrows and circles). Note that, for illustrative purposes, the dendrites are depicted at a different angle in these higher magnification images. **b**, An example neuron before (ZT5) or after (ZT10) exposure to clonidine and 2-chloroadenosine. Scale bars: 15 μm (**a**, **b left**) and 5 μm (**a**, **b right**). **c**, Larvae (n = 80) exposed to lights OFF at mid-day (ZT8, first arrow in schematic) took longer to sleep (mean ± SEM) compared to lights OFF at the end of day (ZT14, 2nd arrow). ****$P = 2.27 \times 10^{-15}$, Kruskal-Wallis. **d**, Average locomotor activity ( ± 95%CI) on a 14 hr:10 hr LD cycle before, during, and after a 5 hr midday (ZT5-10, 7 dpf, shaded purple panel) exposure to melatonin (n = 31 larvae), clonidine (n = 32), or DMSO (n = 32). Data from two independent experiments. **e**, Larvae treated with either melatonin or clonidine from ZT5-10 had reduced and delayed sleep ( ± SEM) in first hour of the night (ZT14-15) compared to controls. *$P < 0.05$, **$P < 0.01$, ****$P < 0.0001$ Dunnett's Test.

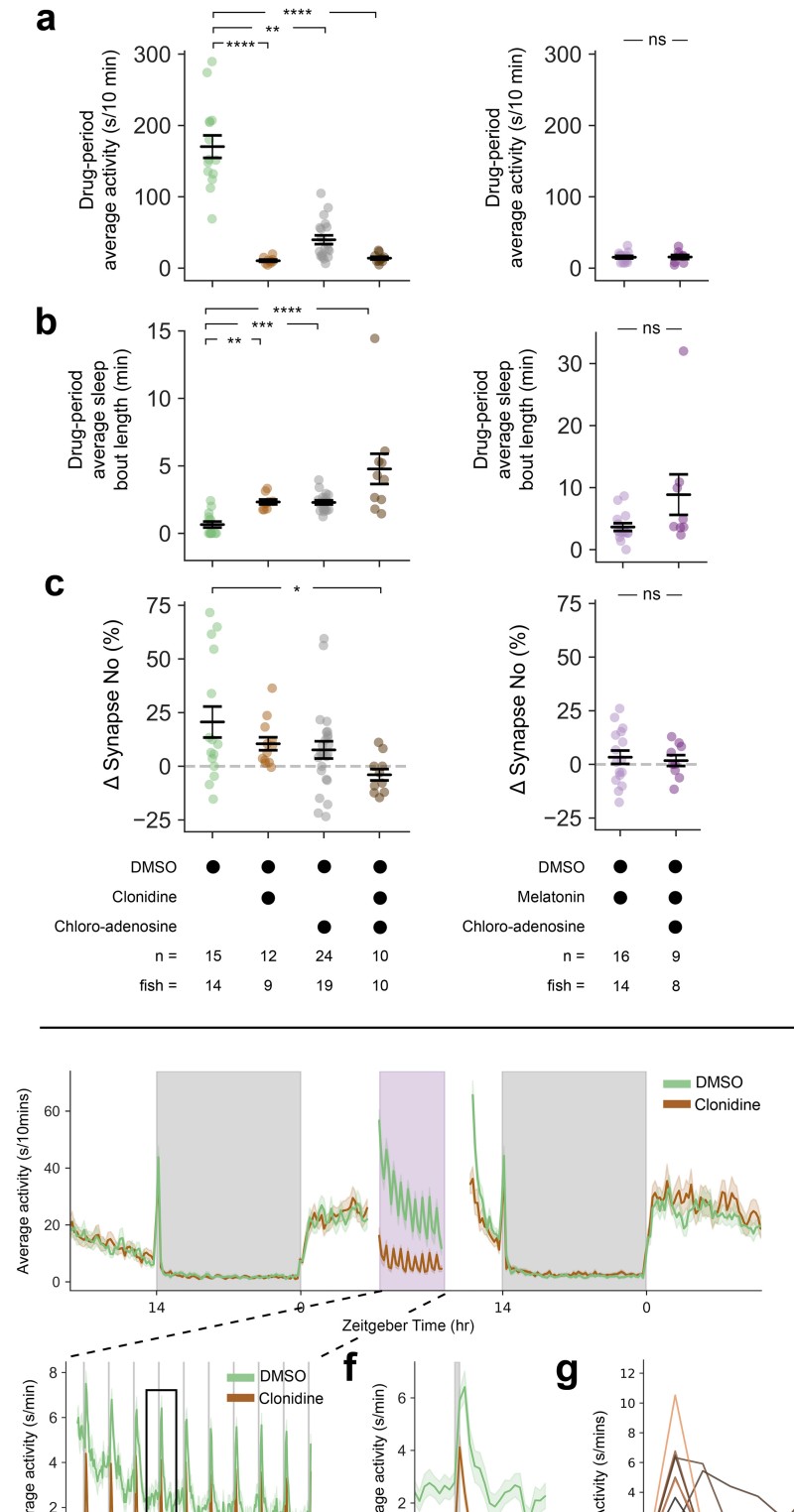

**Extended Data Fig. 10** | See next page for caption.

**Extended Data Fig. 10 | Drug-evoked day time sleep induces synapse loss only when clonidine and 2-chloroadenosine are co-administered.** **a-b**, Clonidine-, 2-chloroadenosine-, and/or melatonin-treated larvae have a lower average activity ( ± SEM) and longer average sleep bout lengths ( ± SEM) during the 5 hr drug period compared to DMSO treated controls. **c**, The average percentage change in synapse number ( ± SEM) within each neuron of DMSO, clonidine-, 2-chloroadenosine-, and/or melatonin-treated larvae. *P < 0.05, **P < 0.01, ****P < 0.0001 Kruskal-Wallis with post-hoc Dunn's test (**b** left and right; and **c**, left) or one-way ANOVA (**a** right, **c** right). **d**, The average activity of larvae before, during and after treatment with either 30 μM clonidine or DMSO from ZT5-10 (purple shaded area) at 7 dpf. 1-minute dark pulses were given every 30 min during the treatment period to test for responsiveness. **e**, Higher resolution time-course of average locomotor activity during the drug treatment and dark-pulse period (ZT5-10). **f**, Both clonidine and DMSO-treated larvae respond to dark pulse with an increase in locomotion, known as the visuomotor response or dark photokinesis. Shown is the average locomotor response to a single 1-minute dark pulse delivered at ZT7. **g**, Locomotor activity for each larva-treated with clonidine (1-minute bin) at the time of dark pulse (ZT7) shown in **d**. Of the 13 larvae that were inactive at the onset of the 1-minute dark pulse, 12 rapidly increased their locomotor activity within 1 min.

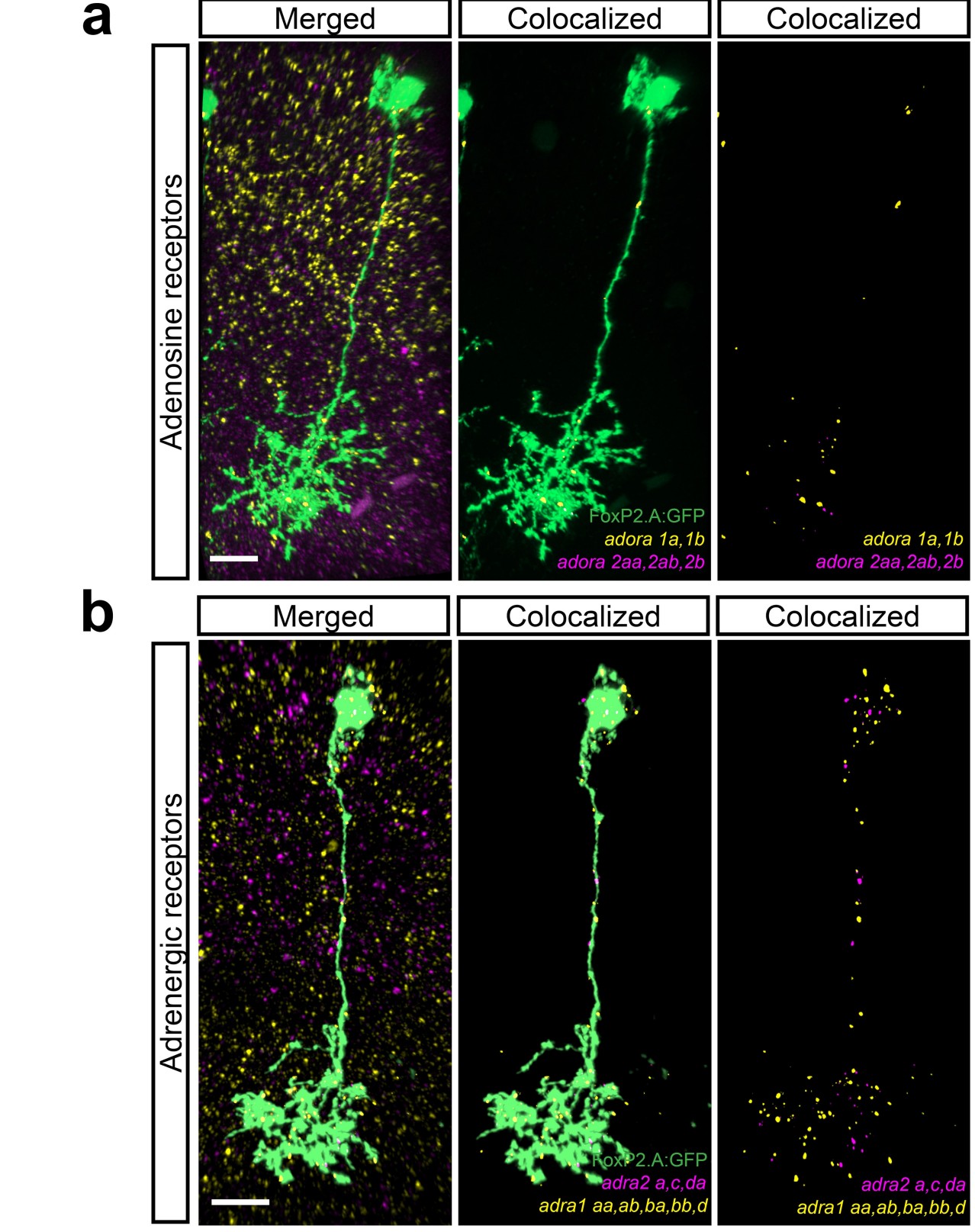

**Extended Data Fig. 11 | FoxP2.A+ neurons express adenosine and adrenergic receptors transcripts.** Examples of adrenergic and adenosine receptor transcripts that colocalize with labelled FoxP2.A+ neurons (middle and right panel) as detected by in situ Hybridization Chain Reaction (HCR, see Methods). **a**, A single labelled tectal neuron (green) colocalizes with a cocktail of HCR probes that detect *adora1a-b* (yellow, encoding for adenosine receptors A1a and A1b) and *adora2aa, -ab, -b* (magenta, encoding for adenosine receptors A2aa, A2ab, and A2b) transcripts. **b**, Single FoxP2.A+ neuron (green) also colocalize with an HCR probe cocktail that detects *adra1 aa,-ab, -ba, -bb, -d* (yellow, encoding zebrafish α1 adrenergic receptor orthologs) and *adra2a, -c, -da* (magenta, encoding zebrafish α2 adrenergic receptor orthologs) transcripts. Scale bar: 10 µm (**a**, **b**). Representative data from 5 larvae. Images of co-localized transcripts chosen from n = 11 neurons (**a**) and n = 10 neurons (**b**).

# Reporting Summary

## Statistics

For all statistical analyses, confirm that the following items are present in the figure legend, table legend, main text, or Methods section.

| n/a | Confirmed | |
|---|---|---|
| ☐ | ☒ | The exact sample size (*n*) for each experimental group/condition, given as a discrete number and unit of measurement |
| ☐ | ☒ | A statement on whether measurements were taken from distinct samples or whether the same sample was measured repeatedly |
| ☐ | ☒ | The statistical test(s) used AND whether they are one- or two-sided<br>*Only common tests should be described solely by name; describe more complex techniques in the Methods section.* |
| ☒ | ☐ | A description of all covariates tested |
| ☐ | ☒ | A description of any assumptions or corrections, such as tests of normality and adjustment for multiple comparisons |
| ☐ | ☒ | A full description of the statistical parameters including central tendency (e.g. means) or other basic estimates (e.g. regression coefficient) AND variation (e.g. standard deviation) or associated estimates of uncertainty (e.g. confidence intervals) |
| ☐ | ☒ | For null hypothesis testing, the test statistic (e.g. *F*, *t*, *r*) with confidence intervals, effect sizes, degrees of freedom and *P* value noted<br>*Give P values as exact values whenever suitable.* |
| ☒ | ☐ | For Bayesian analysis, information on the choice of priors and Markov chain Monte Carlo settings |
| ☒ | ☐ | For hierarchical and complex designs, identification of the appropriate level for tests and full reporting of outcomes |
| ☐ | ☒ | Estimates of effect sizes (e.g. Cohen's *d*, Pearson's *r*), indicating how they were calculated |

*Our web collection on statistics for biologists contains articles on many of the points above.*

## Software and code

Policy information about availability of computer code

| Data collection | zebrafish behavior tracking: Zebrabox/Videotrack (quantization mode), Viewpoint LifeSciences |
|---|---|
| Data analysis | NIH Image J (http://imagej.nih.gov/ij/)<br>Antibody analysis: scripts in Python (available at https://github.com/anyasupp/single-neuron-synapse)<br>Behavior: scripts in Matlab R2023 (Mathworks): https://github.com/JRihel/Sleep-Analysis<br>Behavior, frame-by-frame; R: https://github.com/francoiskroll/FramebyFrame). |

For manuscripts utilizing custom algorithms or software that are central to the research but not yet described in published literature, software must be made available to editors and reviewers. We strongly encourage code deposition in a community repository (e.g. GitHub). See the Nature Portfolio guidelines for submitting code & software for further information.

## Data

Policy information about <u>availability of data</u>

All manuscripts must include a <u>data availability statement</u>. This statement should provide the following information, where applicable:

- Accession codes, unique identifiers, or web links for publicly available datasets
- A description of any restrictions on data availability
- For clinical datasets or third party data, please ensure that the statement adheres to our <u>policy</u>

Data and code can be found https://github.com/anyasupp/single-neuron-synapse.

## Research involving human participants, their data, or biological material

Policy information about studies with <u>human participants or human data</u>. See also policy information about <u>sex, gender (identity/presentation), and sexual orientation</u> and <u>race, ethnicity and racism</u>.

| | |
|---|---|
| Reporting on sex and gender | *Use the terms sex (biological attribute) and gender (shaped by social and cultural circumstances) carefully in order to avoid confusing both terms. Indicate if findings apply to only one sex or gender; describe whether sex and gender were considered in study design; whether sex and/or gender was determined based on self-reporting or assigned and methods used.* <br> *Provide in the source data disaggregated sex and gender data, where this information has been collected, and if consent has been obtained for sharing of individual-level data; provide overall numbers in this Reporting Summary.  Please state if this information has not been collected.* <br> *Report sex- and gender-based analyses where performed, justify reasons for lack of sex- and gender-based analysis.* |
| Reporting on race, ethnicity, or other socially relevant groupings | *Please specify the socially constructed or socially relevant categorization variable(s) used in your manuscript and explain why they were used. Please note that such variables should not be used as proxies for other socially constructed/relevant variables (for example, race or ethnicity should not be used as a proxy for socioeconomic status).* <br> *Provide clear definitions of the relevant terms used, how they were provided (by the participants/respondents, the researchers, or third parties), and the method(s) used to classify people into the different categories (e.g. self-report, census or administrative data, social media data, etc.)* <br> *Please provide details about how you controlled for confounding variables in your analyses.* |
| Population characteristics | *Describe the covariate-relevant population characteristics of the human research participants (e.g. age, genotypic information, past and current diagnosis and treatment categories). If you filled out the behavioural & social sciences study design questions and have nothing to add here, write "See above."* |
| Recruitment | *Describe how participants were recruited. Outline any potential self-selection bias or other biases that may be present and how these are likely to impact results.* |
| Ethics oversight | *Identify the organization(s) that approved the study protocol.* |

Note that full information on the approval of the study protocol must also be provided in the manuscript.

# Field-specific reporting

Please select the one below that is the best fit for your research. If you are not sure, read the appropriate sections before making your selection.

☒ Life sciences ☐ Behavioural & social sciences ☐ Ecological, evolutionary & environmental sciences

For a reference copy of the document with all sections, see <u>nature.com/documents/nr-reporting-summary-flat.pdf</u>

# Life sciences study design

All studies must disclose on these points even when the disclosure is negative.

| | |
|---|---|
| Sample size | We did not have any prior knowledge of the neuron to neuron synapse count variability, nor how this would change over time. Thus, we could not perform power calculation estimates in advance. Post-hoc power calculations based on light:dark synapse data showed that a sample size of n=6 is sufficient to achieve a 95% power at an alpha of 0.01. |
| Data exclusions | Analysis of Type 1 neurons was excluded due to their extreme rarity. |
| Replication | Experiments involving neuron synapse dynamics were independently repeated and replicated. Sleep deprivation and drug analyses are based on at least three independent study sessions. The effects of sleep deprivation on synapse dynamics was retested in two independent non-visual neuronal populations. |
| Randomization | For behavioral tracking experiments, larvae were randomly chosen to be tracked from Petri dishes of ~50 larvae each. |
| Blinding | All image files of synapse tracking experiments were blinded by an independent research prior to segmentation and puncta quantifications. |

# Reporting for specific materials, systems and methods

We require information from authors about some types of materials, experimental systems and methods used in many studies. Here, indicate whether each material, system or method listed is relevant to your study. If you are not sure if a list item applies to your research, read the appropriate section before selecting a response.

## Materials & experimental systems

| n/a | Involved in the study |
|-----|----------------------|
| ☐ | ☒ Antibodies |
| ☒ | ☐ Eukaryotic cell lines |
| ☒ | ☐ Palaeontology and archaeology |
| ☐ | ☒ Animals and other organisms |
| ☒ | ☐ Clinical data |
| ☒ | ☐ Dual use research of concern |
| ☒ | ☐ Plants |

## Methods

| n/a | Involved in the study |
|-----|----------------------|
| ☒ | ☐ ChIP-seq |
| ☒ | ☐ Flow cytometry |
| ☒ | ☐ MRI-based neuroimaging |

## Antibodies

| | |
|---|---|
| Antibodies used | Anti-pan-MAGUK (mouse monoclonal, clone K28/86, Millipore)<br>Anti-tRFP (rabbit polyclonal, AB233, Evrogen) |
| Validation | According to manufacturer: Demonstrated to react with rat. Predicted to react with human based on immunogen design. Predicted to react with mouse based on 100% sequence homology. Demonstrated to work in zebrafish in Sheets et al., 2011; replicated in this study to label synapse puncta. |

## Animals and other research organisms

Policy information about studies involving animals; ARRIVE guidelines recommended for reporting animal research, and Sex and Gender in Research

| | |
|---|---|
| Laboratory animals | zebrafish: AB/TL; nacre |
| Wild animals | *Provide details on animals observed in or captured in the field; report species and age where possible. Describe how animals were caught and transported and what happened to captive animals after the study (if killed, explain why and describe method; if released, say where and when) OR state that the study did not involve wild animals.* |
| Reporting on sex | Sex is not yet biologically determined in the larval zebrafish ages used in this study. |
| Field-collected samples | *For laboratory work with field-collected samples, describe all relevant parameters such as housing, maintenance, temperature, photoperiod and end-of-experiment protocol OR state that the study did not involve samples collected from the field.* |
| Ethics oversight | project licenses PA8D4D0E5 and PP6325955 awarded to JR, according to the UK Animal Scientific Procedures Act (1986) |

Note that full information on the approval of the study protocol must also be provided in the manuscript.

