## [Peer Review File · Nature]

Manuscript Title: Sleep pressure modulates single-neuron synapse number in zebrafish

Reviewer Comments & Author Rebuttals

Reviewer Reports on the Initial Version:

Referees' comments:

Referee #1 (Remarks to the Author):

The study by Suppermpool and colleagues aims at testing, in their own words, “the scale and selectivity of SHY”, according to which overall synaptic strength increases during waking due to ongoing learning and decreases during sleep. This is a very important study that includes many technically challenging experiments. While the evidence supporting SHY so far has been collected mainly in rodents and flies, this study extends it to zebrafish larvae and addresses several specific unresolved questions. The experiments were performed in single tectal neurons of the zebrafish larvae at a developmental time (dpf 7-9), when synapse count is stable, ruling out major confounding issues due to significant synapse formation or pruning. The first main result is that, in the majority of single neurons, the average synaptic strength, measured by counting excitatory synapses and averaging the PSD95 signal intensity of all excitatory synapses in each neuron, is higher at the end of the waking day and decreases after sleep at night, in line with the main tenet of SHY. The authors show convincingly that these changes are driven by sleep and wake and not by the circadian clock. Moreover, through pharmacological experiments the authors provide direct evidence to SHY’s assumption that sleep-dependent synaptic renormalization must be a self-limiting process, and that the need for such a process originates from plastic changes occurring during wake.

Another major finding of the paper is that a minority of neurons show no changes in overall synaptic strength due to sleep and wake, or show changes opposite to those expected by SHY. While this is potentially very interesting, it is worth pointing out that SHY does not claim that all synapses behave in the same way across the sleep/wake cycle, but only that the net effect of wake is an increase in synaptic strength, and the net effect of sleep is a decrease. The molecular and ultrastructural evidence is in line with this claim and shows that most (not all) synapses get stronger/bigger after wake and smaller/weaker after sleep. Moreover, and more crucial here, the conclusion that some neurons behave differently is reached through experiments in which only one neuron was studied in each larva, and no information about the actual sleep/wake pattern for each animal is provided. In other words, it remains possible that the variability is not driven by the single neuron but rather by the behavior of the single animal (i.e., some larvae did not sleep well enough to allow synapse loss). Looking at figure 3, where sleep/wake data were collected for each larva, it is obvious that there is significant variability in the amount of sleep at night across animals, even after sleep deprivation and more so in baseline conditions. Adding the sleep/wake values for the animals that were used for figures 1 and 2 would go a long way to addressing this question, because even in the type 2 neurons there is some variability.

A related point is that the more robust results were found in type 2 neurons, which are also the more numerous; type 3 includes only 4 neurons, making it difficult to draw strong conclusions without knowing about the sleep/wake details; interestingly, type 3 neurons seems to behave more like type 2 neurons after sleep deprivation, which should be discussed; in general, as mentioned above, figure 3 strongly suggests that there is large interanimal variability in the sleep pattern at night, which is not surprising given the definition of sleep used in zebrafish; from my reading of the Methods I could not determine whether the animals used for figures 1 and 2 were videotracked between imaging sessions; if so, the sleep data should be correlated with the synaptic data, to address the fundamental questions whether the variability is across cells or across animals. In sum, the authors should 1) show the sleep amounts; 2) test whether sleep amounts and synaptic changes are correlated at the individual animal level.

The statement that neurons lose synapses only when sleep pressure is high (e.g. line 153) should be clarified; as it stands, it could be interpreted as if synapse loss only occurs during recovery sleep after sleep deprivation, which is clearly not the case because it happens also in controls. Moreover, figure 3 appears to show that the average synapse loss in the first 4 hours of sleep in controls (ZT14-18) is not different from the average synapse loss in the first 4 hours of recovery sleep after sleep deprivation (but please provide the statistics). Figure 3 also shows that in controls, the amount of sleep is still correlated with the loss of synapses even in the second part of the night. The fact that synapse loss in controls occurred, on average, during the first but not the second part of the night, is presumably because on average, larvae slept more early in the night than later; it would be helpful to provide these numbers (average sleep in the first and second part of the night). In short, the current data show that synaptic changes require a minimum amount of sleep to occur, but even sleep late at night in controls can do the job.

Other points

Abstract:

Line 11: a more recent ref for SHY (Neuron 2014; ref 7) should be added here

Lines 12-13: the refs here are not the correct ones for “large neuronal populations....etc”

Lines 15-17: reading this sentence the reader would likely assume (as I did), that more than one neuron was studied in each animal, which is not the case; please clarify

Lines 17-18: I am not sure the data actually support the claim that it is only sleep with high sleep pressure that is linked to synaptic renormalization (see other comment about figure 3), or in any case this sentence is likely to be misinterpreted by sleep scientists

Line 17: please avoid the use of “downscaling”, unless evidence can be provided that the decrease in synaptic size after sleep is actually scaling (proportional to size)

Line 19: it should be clear that forced sleep was pharmacologically induced

Introduction

Line 40 (Ref 16): It is unclear why this study, which shows that downward homeostatic regulation of firing rate is restricted to sleep, would be inconsistent with SHY.

Line 40-41: it should be stated that Refs 4,21 refer to studies in young adolescent mice in which there is still spine formation/elimination: in fact, both studies found that spine formation prevails over spine elimination during wake, and the opposite during sleep

Results

Line 70-71: SHY is about synaptic strength; the number of synapses may or may not change, especially in a mature brain; in other words, SHY does not require that both strength and number must change

Line 81: Figure 1d and Ext Figure 2 prove developmental stability in the number of synapses, but the point is never actually made in the main text; for this reason the message conveyed by panel 1d is a bit obscure. Legend (line 729): each row is a synapse?

Line 84: larvae are freely moving between imaging sessions, but are they video-recorded to quantify their sleep/wake amounts?

Line 106-107: the conclusion that SHY does not apply to all neurons assume that the only difference across animals is in the cell, not in the sleep/wake pattern. As mentioned above, it is crucial to provide evidence to support this assumption.

Line 104-107: this sentence implies that multiple neurons were studied in each animal, which is not the case; please correct; "population level" here refers to different neurons in different animals

Line 113: 4 neuron subtypes are mentioned, but type 1 is never discussed; please state why.

Figure 2g is difficult to reconcile with the statement that there were minimal synaptic changes in type 4 neurons; this seems to apply mainly to their number

Line 136: see previous comment for the abstract; the sentence that neurons lose synapses in sleep only when sleep pressure is high may be misinterpreted

Line 170: in relation to the permissive (or not) role of sleep, it seems relevant to discuss the study by Miyamoto et al (Nature Comm 2021).

Discussion

Line 203; most slow waves are actually local, and local plastic changes lead to local changes in sleep slow waves (Huber et al., Nature 2004).

Methods

Line 539: it is stated that locomotor activity was measured in randomly selected larvae. Are those the ones used for image analysis in figures 1 and 2?

Signed: Chiara Cirelli

Referee #2 (Remarks to the Author):

Suppermpool et al. Sleep pressure modulates single neuron dynamics in zebra fish.

This is a well written and engaging paper, reporting on a stylish and original way of testing the Synaptic Homeostasis Hypothesis (SHY) hypothesis, a major but controversial hypothesis for the reason why we sleep. SHY suggests that synapses are downscaled during sleep, but in the last few years, there have been quite a few papers finding the opposite, at least in mammalian systems. Here, Suppermpool et al., use zebra fish larvae and an elegant genetic way to visualize excitatory synapses to count synapses in the tectum of the fish during wake and natural sleep, following sleep deprivation, and with drug-induced sleep. The authors find that, indeed, many excitatory synapses disappear during sleep, but this depends on the type of neuron, and the subdomain of the neurons. This loss happens mainly during high sleep pressure, and after sleep deprivation, but not with drug-induced sleep. The findings of Suppermpool et al could be important for the sleep field, showing for the first time what actually happens to synapses globally during sleep. It's a win-win situation: the originators of SHY might be pleased, as this new paper confirms the basic SHY idea; but also the investigators who found the opposite findings will be intrigued, as many tectal neurons show no change or even an increase in synapse number during larval sleep (Type 3 neurons). So, the current manuscript can be a satisfying resolution of the previous conflicting literature, although revealing quite a complicated and subtle situation.

Specific points:

1. Development. Are these findings confounded by synaptogenesis in these larval fish? As far as I could understand, synapses increase during the day (e.g. 14%) but only decrease at night by 1.9% (lines 85, 86). Similarly, the average GFP signal intensity increased 37% in wake and only decreased 11.7% at night (lines 90, 91). Similarly, focusing in on type 2 cells, 14.8% increase during the day, 8.7 % reduction at night. So overall, synapses are being added to the brain each day? Thus, a confound to the study is that the brain is still developing? Maybe this type of synapse loss/pruning during sleep is a specific developmental phenomenon and nothing to do with sleep pressure? This confound should be addressed. If it is not a developmental effect, why is there a net increased synapse number? (a further point, the decrease in synapse number is not "renormalization", otherwise values would go back to what they were before, and this word "renormalization" is misused in the Abstract).

2. Circadian influences. It was a good experiment to do this in free running and “clockbreak” conditions (e.g. green and pink traces in Fig. 1g). Nevertheless, as the authors discuss, there is a baseline circadian effect, but damped, and in “clockbreak conditions” (Pink traces in Fig 1g) the synaptic changes are abolished. (line 98 onwards). So, doesn't this mean the opposite of what the authors concluded, that there is a strong effect of the circadian system? Or have I misunderstood something? This needs to be in the Discussion maybe.

In Fig 1g and i, in the free running conditions (green lines), the synapse count is lower, but the relative intensity is surprisingly much higher compared to the control (blue lines). As the authors mentioned (line 100-102), it may indicate that there is an increase in the synaptic strength. At ZT10, although the synapse count did not change, the intensity significantly increased. As for the synaptic function, both the number and the strength of synapses matter. Does it mean that under the free running conditions, the homeostasis is displayed slightly different? (increased total synaptic efficacy during the day due to enhanced strength at individual synapse, although the number of synapse does not change, and then renormalized at night).

4. To test the idea that only the sleep elicited by high sleep pressure causes synapse number to change, the authors want to elicit extra day time sleep by giving the fish either melatonin or clonidine. They call these drug-evoked states “sleep” (in the abstract and elsewhere in the text). The description of this vigilance state as ‘sleep’ is not strictly accurate. In mammals, clonidine (a dirty alpha2 adrenergic agonist) might give a NREM-like sleep that is a sedative state, as well as numerous other changes in vasculature, heart rate etc. Furthermore, It might decrease cAMP which might have the opposite effect on what is happening in real sleep, and so maybe counteract any synaptic changes found in real sleep? And melatonin resets the circadian clock, but at least in mammals is definitely not a hypnotic. So, when the authors write “daytime sleep” for clonidine (line 209), this is misleading. Depending on dose, this is quite likely to be sedation/light anesthesia. So, what they really can conclude is that synapse numbers do not change during “sedation”. Which is fine. But the effect of “daytime sleep”, in my opinion, has not been tested, and this should be addressed if the authors want to show that the last line of the abstract is true.

5. In Fig. 4b, the conceptual diagram, please label the y axis. Is this sleep drive on the left-hand side and amount of sleep on the right hand side? This diagram is quite confusing, and there is also a lot of wishful thinking in it, in my opinion. I think the purple zone for ZT5 to ZT10 is sedation (at least for clonidine, as noted above), not induced daytime “sleep”, and the dotted curves are presumably the ‘Process S’ for sleep drive. First, why does the dotted purple curve for clonidine have lower values? – this suggest clonidine has reduced the sleep drive and has therefore fulfilled some aspect of sleep's function independent of synapse number adjustment? This would be a very interesting result if true, because it would suggest that clonidine could replace natural sleep, the opposite of what the authors are suggesting? Second point is, in general, why should the homeostatic sleep drive have these shaped curves as a universal property of sleep drive? This curve shape (Process S) is experimentally based on delta power in the mammalian EEG with time awake, and also assumes delta power is reflective of the sleep drive. I feel it is at best a hopeful hypothesis here. There is no a priori reason why any sleep drive

curves should have this shaped curve for zebra fish?

6. Abstract. A small point, but the phrase “synapses are collectively lost during sleep in most, but not all neurons” is a rather strong statement for quite a complex and subtle situation found by the authors. Of the 4 types of tectal neurons studied, this loss seems to happen mainly in type 2 neurons, and in fact, only in a subdomain of these neurons (Lines 115 and 117), and Type 3 have the “opposite dynamic”, an effect of synapse gain as strong as synapse loss. So, I feel a rephrasing of the abstract is needed to get the complexity across.

7. Another small point. Several times the authors describe sleep as being more “intense” (lines 146, lines 169). This word is used too loosely and could be removed. I think “intensity” is assumed to be higher if delta power of NREM sleep (in mammals) is higher. But is there evidence that delta power does correspond to intensity? Other than higher delta power, there is no clear evidence that rebound sleep after sleep deprivation is actually more “intense”. (I have been guilty of the same misuse myself, but I now think we should not use this word unless there is evidence; sleep can be of course longer, but this is not necessarily more “intense”).

8. This is just a comment. Certainly, adenosine is a neurotransmitter like many others that can regulate sleep, but adenosine as a candidate somnogen is a rather tired (excuse the pun) suggestion (line 200). At least in mammals, there is contradictory evidence about adenosine’s role as a somnogen, and especially about the measurements of and if adenosine does actually increase with wakefulness. It now seems hard to find support for this idea. So, it is interesting here that the authors want to speculatively resurrect it. How do they envisage the adenosine increases being confined to just one part of the cell (for e.g. type 2 neurons)?

(9. Minor. Journal name (Neuron?) missing from ref. 25 and ref. 8).

Referee #3 (Remarks to the Author):

The synaptic homeostasis hypothesis, formulated by Cirelli and Tononi, has shaped much of the thinking about sleep function in the field of neurobiology. The hypothesis has been supported by a large amount of data from a few lab, demonstrating increased measures of synaptic density, strength, and activity following a period of wakefulness when compared with a similar period of sleep. However, in recent years (i.e., the past 10 or so) a number of publications, from many different labs and model systems, have demonstrated that synaptic "downscaling" is not uniform during sleep, may be neuron type-specific, and is not consistently linked to any functional consequences of sleep/sleep deprivation in the brain.

With this in mind, the present study uses a recently-developed genetic tool to image synapse density and "intensity" (which may or may not be related to synaptic strength) in a different neural system, the

optic tectum of developing zebrafish. The technique used for this quantification is recently-developed and interesting. However the overall experimental design and outcome are similar to much of what has been presented in the literature recently - some synapses on some neurons (but not all) are reduced after a period of sleep and increased after a period of wake. Hence, this adds another example of a phenomenon to an established body of literature, but does not constitute a major advance.

A major confound, however, is that sleep/wake control in the main experiments is primarily a function of ambient light. In the case of a brain region that processes visual information, this raises many questions about how generalizable the findings will be. Even in the case of experiments involving sleep deprivation, the manipulation occurs in red light, and it isn't clear at all that this doesn't constitute a period of heightened visual input to the tectum. This (to my read, more parsimonious) interpretation - that visual input drives synapse formation, and lack thereof drives loss, in the tectum, would reconcile some of the findings in the paper that are incongruous with the hypothesis - i.e., that hypnotic augmented sleep does not cause the same synapse loss.

Another major concern is with data presentation, in that the authors do not present any image of what precisely they are quantifying. There is a zoomed out view of an entire dendritic arbor, and then a cartoon schematic of what is happening to synapses - no data in the middle. It is hard for any reviewer to evaluate the quality of the data when it isn't even shown in a representative image.

Referee #4 (Remarks to the Author):

In this manuscript, Suppermpool et al. explored the permissive or causal role of sleep on synaptic downscaling using the tectal neurons of larval zebrafish. The authors selected three zeitgeber times spanning one day-night cycle to measure the synaptic number change over daytime and nighttime. Then they used sleep deprivation (SD) to test the sleep function on synapse number change. Finally, they argued that the sleep's effects on synapse number change is subordinate to sleep pressure for the induced sleep is not sufficient to trigger synapse loss at daytime with low sleep pressure. The authors leveraged the advantage of zebrafish to measure all synapses on individual tectal neurons and found subtype-specific (type 2 tectal neuron) change and regulation by sleep. In total, this study mainly describes a potentially interesting observation that overlaps with a prior study (Appelbaum et al., 2010) while views more from the sleep perspective. And the subtype-dependent and even sub-arbor dependent phenomenon is beyond comprehension. A more detailed mechanistic investigation would strengthen the manuscript.

Major Concerns:

1. Appelbaum et al, Neuron, (2010) previously showed a homeostatic regulation of synapse number in Hypocretin neurons in larval zebrafish. Although using different cell model, like the present study, they used synaptic marker to assay the change of number of synaptic puncta of individual cells after sleep deprivation. This prior study has striking similarities to the present manuscript in methodology and result: increase of synaptic number after SD. Unfortunately, despite the relevance, this prior study was

not cited appropriately or discussed. This prior study also examined the circadian influence while the present study only investigated and explained from the sleep perspective.

2. Throughout the manuscript, the author only measured the total number and intensity change of FingR(PSD95) puncta, but articulated this is synapse dynamics. I envision "synapse dynamics" as the concurrent synapse formation and elimination events. What do the authors mean by synapse dynamics and, given their definition, how is synapse dynamics affected?

3. How about increase sleep at nighttime with high sleep pressure as implied by the authors. Can the sleep pressure be experimentally measured? At single neuron/synapse level, whether the synapses gained during SD correlate with synapse loss or are more sensitive to subsequent sleep recovery.

Minor Concerns:

1. line 73–'have a stable window of synapse maturation from 5 to 9 days post fertilization (dpf)'
The synapse should not be mature until about 7 dpf as shown in the previous work (Smith, 2004, Nat. Neurosci.).

2. line 161–'Consistent with this, the total time each 160 larva spent asleep was significantly negatively correlated with the rate of synapse loss in both halves of the night.'

This description is controversial. Should not it be positively correlated?

Author Rebuttals to Initial Comments:

Suppermpool et al

Response to Reviewers:

We thank the reviewers for their detailed and generally enthusiastic reviews. We have addressed all of the reviewer comments, including the addition of new data that demonstrates the generalizability of our results to two non-visual neuron types. We also present new experimental evidence for a mechanistic explanation for the relationship between sleep pressure and synapse dynamics—interference with adenosine and noradrenaline signaling breaks the sleep/synapse relationship, allowing for synapse loss during drug-induced daytime naps. We believe these new experiments together with our other changes have made our manuscript even stronger.

Reviewer comments are in *italics*. Our comments are in **red**.

Reviewer 1:

The study by Suppermpool and colleagues aims at testing, in their own words, “the scale and selectivity of SHY”, according to which overall synaptic strength increases during waking due to ongoing learning and decreases during sleep. This is a very important study that includes many technically challenging experiments.

We thank the reviewer for the recognition of the importance and the scale of the technical achievement of our work.

While the evidence supporting SHY so far has been collected mainly in rodents and flies, this study extends it to zebrafish larvae and addresses several specific unresolved questions. The experiments were performed in single tectal neurons of the zebrafish larvae at a developmental time (dpf 7-9), when synapse count is stable, ruling out major confounding issues due to significant synapse formation or pruning. The first main result is that, in the majority of single neurons, the average synaptic strength, measured by counting excitatory synapses and averaging the PSD95 signal intensity of all excitatory synapses in each neuron, is higher at the end of the waking day and decreases after sleep at night, in line with the main tenet of SHY. The authors show convincingly that these changes are driven by sleep and wake and not by the circadian clock. Moreover, through pharmacological experiments the authors provide direct evidence to SHY’s assumption that sleep-dependent synaptic renormalization must be a self-limiting process, and that the need for such a process originates from plastic changes occurring during wake.

We strongly agree that our results, which we believe are the first to test SHY via direct observation of whole neuron synapse dynamics via repeated imaging, are in line with major predictions and assumptions of SHY. We are grateful for the assertion that we have shown the sleep dependence of synapse changes convincingly.

Another major finding of the paper is that a minority of neurons show no changes in overall synaptic strength due to sleep and wake, or show changes opposite to those expected by SHY. While this is potentially very interesting, it is worth pointing out that SHY does not claim that all synapses behave in the same way across the sleep/wake cycle, but only that the net effect of wake is an increase in synaptic strength, and the net effect of sleep is a decrease. The molecular and ultrastructural evidence is in line with this claim and shows that most (not all) synapses get stronger/bigger after wake and smaller/weaker after sleep.

We completely agree that SHY does not insist that **all neurons must have sleep-wake dependent synapse dynamics, and we apologize if we have implied this. We have altered our language to better reflect this view. For example, instead of couching these observations as evidence against SHY, we now simply report the result and frame the data in the context of neuronal subtype or behavioural**

diversity. For example, lines 112-117 now read: “Moreover, although rhythmic day:night changes in synapses were detected in single neurons on average, the tracking of individual neurons revealed that many cells have different, even opposite, synaptic dynamics (Figure 1g-j, right panels). We therefore sought to test whether these diverse patterns mapped onto distinct neuronal subtypes (e.g. cellular diversity) or might be due to variations in animal behaviour (e.g. individual sleep/wake histories).”

However, we believe identifying exceptions to SHY provide critical clues to the relationship between synapse dynamics and sleep and the extent to which SHY governs diverse neuron types. The ultrastructural evidence from rodents does show subsets of synapses behave opposite to SHY (as we also see in Figure 1d), but we would like to point out that these studies are on snapshots of neuronal populations, leaving single neuron dynamics unresolved. For example, in our dataset every neuron gains/strengthens some synapses and loses/weakens others (e.g., **Extended Data Figure 2a**), but the net change within most neurons is best in line with SHY.

Moreover, and more crucial here, the conclusion that some neurons behave differently is reached through experiments in which only one neuron was studied in each larva, and no information about the actual sleep/wake pattern for each animal is provided. In other words, it remains possible that the variability is not driven by the single neuron but rather by the behavior of the single animal (i.e., some larvae did not sleep well enough to allow synapse loss). Looking at figure 3, where sleep/wake data were collected for each larva, it is obvious that there is significant variability in the amount of sleep at night across animals, even after sleep deprivation and more so in baseline conditions. Adding the sleep/wake values for the animals that were used for figures 1 and 2 would go a long way to addressing this question, because even in the type 2 neurons there is some variability.

This is a very important point. Unfortunately, the experimental challenge of collecting multiple timepoints over multiple days in various lighting contexts for so many single neurons precluded simultaneous tracking of sleep/wake in these Fig1-2 datasets. However, to address your concerns, we have now collected simultaneous behaviour and synapse imaging data for an additional 21 neurons, covering 6 Type 2 neurons, 8 Type 3 neurons, and 6 type 4 neurons (Type 1 is exceedingly rare). In these data, we find there is no statistically significant difference in sleep amount or sleep bout lengths, precluding any effect of random or systematic bias in sleep across larva-neuron pairs in these data. These data are presented in new **Extended Data Figure 6**. Moreover, as we reported in the original text and expand to include the new data (current Figure 2d-g), only Type 2 and (weakly) Type 4, but not Type 3 show an increase in synapses during the waking day and loss at night. Finally, we have limited data from single larvae where we could clearly image two neurons. As you can see in **Reviewer Figure 1**, many neuron pairs within a fish have similar changes in synapse numbers during sleep, but some neurons within the same fish show divergent changes. For example, Fish 2 has a Type 2 neuron that has a net synapse loss, while a Type 3 neuron gains synapses. Since these neurons are in the same larva, they have experienced the same sleep history. We have included this

data only as a reviewer figure for now, due to the small n of these observations; however, we would be happy to include as a supplemental figure if the reviewer prefers.

Reviewer Figure 1. Neurons in the same animal can have opposing synapse count changes during sleep.

A related point is that the more robust results were found in type 2 neurons, which are also the more numerous; type 3 includes only 4 neurons, making it difficult to draw strong conclusions without knowing about the sleep/wake details;

We now provide sleep/wake and synapse data for an additional 8 Type 3 neurons (**Extended Data Figure 6**), which we have included in the analysis of **Figure 2f,g**. These data are consistent with our original conclusions drawn from fewer neurons that Type 3 cells have synapse dynamics opposite to SHY's predictions.

interestingly, type 3 neurons seems to behave more like type 2 neurons after sleep deprivation, which should be discussed;

This is an excellent point. We have reorganized the data to now include this point more clearly in **Extended Data Figure 8d**. We have also added keylines to the text (line 166-172): "As during normal sleep, FoxP2.A tectal neuron subtypes responded differently to SD, with Type 2, and even Type 3 neurons (which did not have SHY-concordant dynamics under baseline conditions), gaining synapses during SD and losing them during recovery sleep, whereas Type 4 neurons did not show any change (Extended Data Figure 8d). This suggests that SD biases synapses towards loss during sleep, even in neurons with different synapse dynamics in baseline conditions."

in general, as mentioned above, figure 3 strongly suggests that there is large interanimal variability in the sleep pattern at night, which is not surprising given the definition of sleep used in zebrafish; from my reading of the Methods I could not determine whether the animals used for figures 1 and 2 were videotracked between imaging sessions; if so, the sleep data should be correlated with the synaptic data, to address the fundamental questions whether the variability is across cells or across animals. In sum, the authors should 1) show the sleep amounts; 2) test whether sleep amounts and synaptic changes are correlated at the individual animal level.

As noted, we have added new data to **Extended Data Figure 6** to address these points.

The statement that neurons lose synapses only when sleep pressure is high (e.g. line 153) should be clarified; as it stands, it could be interpreted as if synapse loss only occurs during recovery sleep after sleep deprivation, which is clearly not the case because it happens also in controls.

We have modified this text to be more accurate, as we did not intend to imply that synapses can only be lost during recovery sleep after sleep deprivation. The subheading now reads, "High sleep pressure facilitates sleep-dependent synapse loss". We believe our new analyses and new mechanistic data (see below) support this claim.

Moreover, figure 3 appears to show that the average synapse loss in the first 4 hours of sleep in controls (ZT14-18) is not different from the average synapse loss in the first 4 hours of recovery sleep after sleep deprivation (but please provide the statistics).

The reviewer is correct that the average synapse loss in the 4 hours of recovery sleep after sleep deprivation is larger but not that different than the first 4 hours of baseline sleep. However, paired experimental designs do not allow for a straightforward statistical comparison between average synapse loss between different time points and conditions (i.e. strictly, only time-matched comparisons between conditions are allowed). However, to follow your suggestion, we independently compared early night controls to late night SD and found that indeed these two groups are not significantly different (ns $p > 0.5$, Kruskal-Wallis ; ns $p > 0.5$, one way ANOVA). This could reflect a saturation of the machinery required to remove synapses, capping the rate of loss. Alternatively, this could be influenced by the difference in circadian time of these two conditions. We now note both these points in the text.

Figure 3 also shows that in controls, the amount of sleep is still correlated with the loss of synapses even in the second part of the night. The fact that synapse loss in controls occurred, on average, during the first but not the second part of the night, is presumably because on average, larvae slept more early in the night than later; it would be helpful to provide these numbers (average sleep in the first and second part of the night). In short, the current data show that synaptic changes require a minimum amount of sleep to occur, but even sleep late at night in controls can do the job.

These comments drive to the heart of one of the central tensions in the interpretation of our data. That is, how do we disentangle sleep pressure from sleep amount, given that animals experiencing high sleep pressure will also sleep more? The sleep deprivation data shows that this early versus late night is not due to the circadian clock (since recovery sleep after SD is circadian matched to the baseline late night), but does not fully disentangle sleep pressure from sleep amount.

Inspired by the reviewer comments, we realized that we could probe this question further by examining the relationship between synapses and sleep in control larvae that naturally experienced sleep deprivation in the first half of the night (i.e. larvae that naturally stayed awake versus larvae that slept more). Consistent with the gentle-handling sleep deprivation data, larvae that were "late sleepers" gained synapses during the first half of the night, while "early sleepers" lost synapses in the first part of the night (new **Figure 3d-e** and **Extended Data Figure 8e**). Moreover, we find a strong negative correlation between the rate of synapse loss in the first half of the night versus the second (i.e. if a neuron lost synapses early at night, they would gain later at night, **Figure 3d**). Thus, sleep late at night *can* do the job as the reviewer suggested, but apparently only if this late sleep is associated with higher sleep pressure.

However, none of these data fully separate sleep pressure and sleep amount, which is why we also forced larvae to sleep during the day, when sleep pressure is naturally still low (**Figure 4**). We now show that not only do these drug-induced naps drive large amounts of sleep (**Figure 4c**) and reduces sleep pressure (**Extended Data Figure 9**), we also now show that during sleep with low adrenergic tone + high adenosine, a molecule long hypothesized to be a sleep pressure signal, synapses are now lost in the middle of the day (new **Figure 4d**). We believe this provides strong evidence that sleep is necessary but not sufficient to drive synapse loss, with signals of sleep pressure also being required.

We'd like to note that this concept would be consistent with some early predictions of SHY, which suggested that mammalian slow-waves, which are strongest during high sleep pressure, might be a driver of sleep-dependent synaptic weakening¹.

Other points

Abstract:

Line 11: a more recent ref for SHY (Neuron 2014; ref 7) should be added here

Added.

Lines 12-13: the refs here are not the correct ones for "large neuronal populations....etc"

Apologies! We've added the correct de Vivo reference.

Lines 15-17: reading this sentence the reader would likely assume (as I did), that more than one neuron was studied in each animal, which is not the case; please clarify

We have fixed this line.

Lines 17-18: I am not sure the data actually support the claim that it is only sleep with high sleep pressure that is linked to synaptic renormalization (see other comment about figure 3), or in any case this sentence is likely to be misinterpreted by sleep scientists

We believe our new data on adenosine and our new analysis of synapse dynamics in early and late sleepers are in support of this point (**Figure 4**). However, we have removed the word "only" and changed to "[...] synapse loss is greatest during sleep associated with high sleep pressure following prolonged wakefulness."'

Line 17: please avoid the use of "downscaling", unless evidence can be provided that the decrease in synaptic size after sleep is actually scaling (proportional to size)

We agree and have removed this term.

Line 19: it should be clear that forced sleep was pharmacologically induced

We have fixed this to read, "Conversely, sleep induced pharmacologically..." (line 20).

Introduction

Line 40 (Ref 16): It is unclear why this study, which shows that downward homeostatic regulation of firing rate is restricted to sleep, would be inconsistent with SHY.

We agree. We have rewritten this to be more precise. The line now reads, "However, other studies have observed no impact of sleep/wake states on synaptic strength and neuronal firing rates (Cary & Turrigiano, 2021; Torrado Pacheco et al., 2021), and some have observed synaptic strengthening during sleep (Chauvette et al., 2012; Durkin et al., 2016; Ruiz et al., 2013; Ognjanovski et al., 2014)"

Line 40-41: it should be stated that Refs 4,21 refer to studies in young adolescent mice in which there is still spine formation/elimination: in fact, both studies found that spine formation prevails over spine elimination during wake, and the opposite during sleep

Great point. We have changed to this more precise statement: “Long term imaging of small segments of dendrites in young and adult mice have also observed sleep/wake-linked synapse dynamics”.

Results

Line 70-71: SHY is about synaptic strength; the number of synapses may or may not change, especially in a mature brain; in other words, SHY does not require that both strength and number must change

This is fair. We have rewritten this line to simply say, “To test whether behavioural state modulates synapse strength and number at the single-neuron level”, as we have already placed the work in context with SHY in the introduction.

Line 81: Figure 1d and Ext Figure 2 prove developmental stability in the number of synapses, but the point is never actually made in the main text; for this reason the message conveyed by panel 1d is a bit obscure. Legend (line 729): each row is a synapse?

We have now added the clarifying line “After confirming the relative stability of tectal neuron synapse counts in the 6-9 dpf developmental window (**Extended Data Figure 2b-d**)[...]”

Line 84: larvae are freely moving between imaging sessions, but are they video-recorded to quantify their sleep/wake amounts?

We did not measure sleep/wake amounts in these initial studies, due to experimental limitations for obtaining so many neurons across long enough timescales. See reply to main comments.

Line 106-107: the conclusion that SHY does not apply to all neurons assume that the only difference across animals is in the cell, not in the sleep/wake pattern. As mentioned above, it is crucial to provide evidence to support this assumption.

This is a great point. See reply to main comment and **Extended Data Figure 6**.

Line 104-107: this sentence implies that multiple neurons were studied in each animal, which is not the case; please correct; “population level” here refers to different neurons in different animals

Thank you for catching this. We have rewritten this section extensively, and this inaccurate phrasing was removed.

Line 113: 4 neurons subtypes are mentioned, but type 1 is never discussed; please state why.

We have added a note about Type 1 (line 126), “excluding the rarely observed Type 1 neurons”. They are exceedingly rare, either due to biases/limits in our methodology for labelling neurons, or because they make up a small proportion of the neurons.

Figure 2g is difficult to reconcile with the statement that there were minimal synaptic changes in type 4 neurons; this seems to apply mainly to their number

The difference in Figure 2g is not significant for Type 4 neurons ($p=0.057$, directional one sample t-test), but we agree there may be a small increase during waking and loss during sleep. We have made this clearer in the text.

Suppermpool et al

Line 136: see previous comment for the abstract; the sentence that neurons lose synapses in sleep only when sleep pressure is high may be misinterpreted

We have corrected this to be more accurate by removing all instances of “only”, which is too strong.

Line 170: in relation to the permissive (or not) role of sleep, it seems relevant to discuss the study by Miyamoto et al (Nature Comm 2021).

We have added this study in the main text “Long term imaging of small segments of dendrites in young and adult mice have also observed sleep/wake-linked synapse dynamics (Maret et al., 2011; Miyamoto et al., 2021; Yang et al., 2012)”

Discussion

Line 203; most slow waves are actually local, and local plastic changes lead to local changes in sleep slow waves (Huber et al., Nature 2004).

Point taken. We have corrected the phrasing to say (line 236-238), “Sleep pressure, as reflected by the density of slow wave activity in mammalian sleep, have also been linked to changes in synapses associated with learning and memory”. Linking our diverse single neuron dynamics to local changes in sleep pressure or slow waves (or the zebrafish equivalent), would be an exciting future direction.

Methods

Line 539: it is stated that locomotor activity was measured in randomly selected larvae. Are those the ones used for image analysis in figures 1 and 2?

As noted above, we did not simultaneously track locomotor activity in Fig1-2 datasets. For Figure 3 and 4 datasets, we measured both locomotor activity and synapse dynamics for the majority of larvae. We have removed the term “randomly selected”, which could be misleading. Strictly speaking, these animals are not random, in the sense that we pre-screen to behaviourally track only larvae verified to have a labelled neuron.

Reviewer 2

This is a well written and engaging paper, reporting on a stylish and original way of testing the Synaptic Homeostasis Hypothesis (SHY) hypothesis, a major but controversial hypothesis for the reason why we sleep. SHY suggests that synapses are downscaled during sleep, but in the last few years, there have been quite a few papers finding the opposite, at least in mammalian systems. Here, Suppermpool et al., use zebra fish larvae and an elegant genetic way to visualize excitatory synapses to count synapses in the tectum of the fish during wake and natural sleep, following sleep deprivation, and with drug-induced sleep. The authors find that, indeed, many excitatory synapses disappear during sleep, but this depends on the type of neuron, and the subdomain of the neurons. This loss happens mainly during high sleep pressure, and after sleep deprivation, but not with drug-induced sleep. The findings of Suppermpool et al could be important for the sleep field, showing for the first time what actually happens to synapses globally during sleep. It’s a win-win situation: the originators of SHY might be pleased, as this new paper confirms the basic SHY idea; but also the investigators who found the opposite findings will be intrigued, as many tectal neurons show no change or even an increase in synapse number during larval sleep (Type 3 neurons). So, the current manuscript can be a satisfying resolution of the previous conflicting literature, although revealing

quite a complicated and subtle situation.

We thank the reviewer for their kind assessment of our manuscript.

Specific points:

1. Development. Are these findings confounded by synaptogenesis in these larval fish? As far as I could understand, synapses increase during the day (e.g. 14%) but only decrease at night by 1.9% (lines 85, 86). Similarly, the average GFP signal intensity increased 37% in wake and only decreased 11.7% at night (lines 90, 91). Similarly, focusing in on type 2 cells, 14.8% increase during the day, 8.7% reduction at night. So overall, synapses are being added to the brain each day? Thus, a confound to the study is that the brain is still developing? Maybe this type of synapse loss/pruning during sleep is a specific developmental phenomenon and nothing to do with sleep pressure? This confound should be addressed. If it is not a developmental effect, why is there a net increased synapse number?

We agree that developmental changes to synapses in our analysis window could be playing a role by overall increasing the absolute number of synapses over time. We have noted in the line 84-85, "After confirming the relative stability of tectal neuron synapse counts in the 6-9 dpf developmental window (Extended Data Figure 2b-d)." Please also notice in Extended Data Figure 4, where we performed some controls for bleaching by imaging once at 7dpf and again at 9dpf, we did not see any significant increase in synapse counts or intensities.

(a further point, the decrease in synapse number is not "renormalization", otherwise values would go back to what they were before, and this word "renormalization" is misused in the Abstract).

We agree and have removed the use of the term "renormalization", which was inaccurate.

2. Circadian influences. It was a good experiment to do this in free running and "clockbreak" conditions (e.g. green and pink traces in Fig. 1g). Nevertheless, as the authors discuss, there is a baseline circadian effect, but damped, and in "clockbreak conditions" (Pink traces in Fig 1g) the synaptic changes are abolished. (line 98 onwards). So, doesn't this mean the opposite of what the authors concluded, that there is a strong effect of the circadian system? Or have I misunderstood something? This needs to be in the Discussion maybe.

We apologize for the confusion. The clockbreak experiments indeed show that the circadian system has a strong effect on synapse dynamics. However, the clockbreak experiments also flatten the 24hr locomotor and sleep/wake behaviour (e.g. **Figure 1f**), meaning that this experiment alone cannot disentangle a direct influence of the clock versus an indirect effect of the clock on the timing of sleep/wake state. The key experiment to disentangle the clock from sleep/wake state is in **Figure 3b**, when we keep the animals awake into the night phase. Even though the circadian clock continues to tick in both controls and sleep deprived conditions (**Extended Data Figure 8b**), the SD animals gain synapses, while the sleeping control animals lose synapses. This suggests that the major (but not necessarily the only) way in which the clock affects synapse dynamics is via changes in sleep and wake dynamics. We have changed several lines of the text to make this clearer, especially line 110, 111), "[...] changes in synapse counts are independent of lighting conditions but do require an intact circadian clock (to drive rhythmic sleep/wake behaviour, see below) (Figure 1g)."

In Fig 1g and i, in the free running conditions (green lines), the synapse count is lower, but the relative intensity is surprisingly much higher compared to the control (blue lines). As the authors mentioned (line 100-102), it may indicate that there is an increase in the synaptic strength. At ZT10, although the synapse count did not change, the intensity significantly increased. As for the synaptic function, both the number and the strength of synapses matter. Does it mean that under the free running conditions, the homeostasis is displayed slightly different? (increased total synaptic efficacy during the day due to enhanced strength at individual synapse, although the number of synapse does not change, and then renormalized at night).

This is an excellent observation that we cannot fully explain at this time. One challenge in making refined interpretations of our free-running experiments is that we performed these under constant light, which means that not only is continuous exposure to light stimuli a confounding factor, but so is the large reduction in sleep duration that occurs under these conditions. Disentangling these factors will require doing free-running experiments in constant dark conditions, but since we must intervene to mount/image synapses/unmount larvae without introducing phase shifts from stray light pulses, this is technically out of reach.

Due to these limitations, we had restricted our initial conclusions from the free-running experiment to the observation that the day-increase, night-decrease in synapse number is not due only to alternating periods of light and dark. However, per the reviewer suggestion, we have now pointed out this difference explicitly in the text (line 106), "The average synapse intensity was significantly elevated across all timepoints and showed a further significant increase in strength only during the subjective day, with no loss of intensity during the subjective night (Figure 1i,j, green). Collectively, these data show that, while light influences the baseline levels of synaptic strength (e.g. Figure 1i)..."

4. *To test the idea that only the sleep elicited by high sleep pressure causes synapse number to change, the authors want to elicit extra day time sleep by giving the fish either melatonin or clonidine. They call these drug-evoked states "sleep" (in the abstract and elsewhere in the text). The description of this vigilance state as 'sleep' is not strictly accurate. In mammals, clonidine (a dirty alpha2 adrenergic agonist) might give a NREM-like sleep that is a sedative state, as well as numerous other changes in vasculature, heart rate etc. Furthermore, It might decrease cAMP which might have the opposite effect on what is happening in real sleep, and so maybe counteract any synaptic changes found in real sleep? And melatonin resets the circadian clock, but at least in mammals is definitely not a hypnotic. So, when the authors write "daytime sleep" for clonidine (line 209), this is misleading. Depending on dose, this is quite likely to be sedation/light anesthesia. So, what they really can conclude is that synapse numbers do not change during "sedation". Which is fine. But the effect of "daytime sleep", in my opinion, has not been tested, and this should be addressed if the authors want to show that the last line of the abstract is true.*

We agree with the reviewer that drug-induced daytime sleep must be interpreted with some caution, and we have refined our language to call these states "drug-induced sleep". This is precisely why we chose to test this with two drugs that have different mechanisms of action. Clonidine is indeed a dirty drug for inducing daytime sleep, but in zebrafish, the accumulated evidence is that melatonin is a natural hypnotic that serves as the sleep-inducing output signal from the circadian clock (Zhdanova et al., 2001; Ghandi et al., 2015), as opposed to the phase-resetting role in mammals.

To reinforce our interpretation that clonidine and melatonin-induced quiescent states are *bona fide* sleep states, we have performed additional control experiments in **Extended Data Figure 9-11**. First,

we show that both melatonin and clonidine administered for 5 hours during the day reduce the accumulation of sleep pressure, since at the start of the subsequent night period, sleep latency is longer and sleep amount is reduced in both drug-treated conditions compared to controls (**Extended Data Figure 9a-c**). Thus, while perhaps not natural sleep, these drug states perform at least one feature of sleep—the dissipation of sleep pressure. Second, we show that at the dose used in our experiments, clonidine-induced quiescence is acutely reversible—animals show rapid, vigorous responses to a stimulus, suggesting this is not a strongly anaesthetic state, paralysis, or coma (**Extended Data Figure 11**). Third, we now have used a third drug, 2-chloroadenosine, which also induces larval sleep at levels similar to clonidine and melatonin. As in the other drug cases, synapses are still gained during adenosine-induced daytime sleep (**Figure 4 and Extended Data 10**).

Finally, we have now leveraged the different mechanisms of action of these drugs to dissect the mechanism by which sleep states govern synapse dynamics. In new data presented in **Figure 4 and Extended Data Figure 10**, we now show that the combination of clonidine and adenosine, but not other combinations, lead to a reduction in synapses during the day. This is consistent with the hypothesis proposed in Diering et al., 2019 that low noradrenergic tone (as induced in clonidine by alpha2-autoreceptors on the locus coeruleus) coupled with high levels of adenosine signalling (as during high sleep pressure) are needed for the weakening or elimination of synapses during sleep.

5. In Fig. 4b, the conceptual diagram, please label the y axis. Is this sleep drive on the left-hand side and amount of sleep on the right hand side? This diagram is quite confusing, and there is also a lot of wishful thinking in it, in my opinion. I think the purple zone for ZT5 to ZT10 is sedation (at least for clonidine, as noted above), not induced daytime “sleep”, and the dotted curves are presumably the ‘Process S’ for sleep drive. First, why does the dotted purple curve for clonidine have lower values? – this suggest clonidine has reduced the sleep drive and has therefore fulfilled some aspect of sleep’s function independent of synapse number adjustment? This would be a very interesting result if true, because it would suggest that clonidine could replace natural sleep, the opposite of what the authors are suggesting? Second point is, in general, why should the homeostatic sleep drive have these shaped curves as a universal property of sleep drive? This curve shape (Process S) is experimentally based on delta power in the mammalian EEG with time awake, and also assumes delta power is reflective of the sleep drive. I feel it is at best a hopeful hypothesis here. There is no a priori reason why any sleep drive curves should have this shaped curve for zebra fish?

We agree that this figure was unnecessarily confusing and have removed it. Instead, we have decided to experimentally confirm some of the predictions for sleep drive following drug induced daytime sleep. For example, if these drug-induced states are sleep states that can dissipate sleep pressure, we predict that sleep pressure at the end of the day would be lower in drugged vs. control animals. Indeed, we see that readouts of sleep drive, such as the speed at which animals fall asleep, which is slower in drug-treated animals, and the total amount of sleep, which is lower, are altered in a way consistent with the day-time sleep states reducing sleep pressure (**Extended Data Figure 9**). We therefore do believe the reviewers’ statement that “clonidine [and melatonin] has reduced the sleep drive and has therefore fulfilled some aspect of sleep’s function independent of synapse number adjustment” -- and this is a revealing result. In addition, we now provide experimental evidence for a mechanistic explanation of our observations; namely, drug-induced sleep during the day can dissipate sleep pressure but synapses can only be lost when molecular conditions of high sleep pressure (high adenosine levels, low adrenergic tone) are also present (**Figure 4d**).

6. Abstract. A small point, but the phrase “synapses are collectively lost during sleep in most, but not

all neurons” is a rather strong statement for quite a complex and subtle situation found by the authors. Of the 4 types of tectal neurons studied, this loss seems to happen mainly in type 2 neurons, and in fact, only in a subdomain of these neurons (Lines 115 and 117), and Type 3 have the “opposite dynamic”, an effect of synapse gain as strong as synapse loss. So, I feel a rephrasing of the abstract is needed to get the complexity across.

This is fair. We have altered our language in the abstract and throughout the text. The abstract now reads, “[...] synapses are gained during periods of wake (either spontaneous or forced) and lost during sleep in a neuron-subtype dependent manner.” (Line 17).

7. Another small point. Several times the authors describe sleep as being more “intense” (lines 146, lines 169). This word is used too loosely and could be removed. I think “intensity” is assumed to be higher if delta power of NREM sleep (in mammals) is higher. But is there evidence that delta power does correspond to intensity? Other than higher delta power, there is no clear evidence that rebound sleep after sleep deprivation is actually more “intense”. (I have been guilty of the same misuse myself, but I now think we should not use this word unless there is evidence; sleep can be of course longer, but this is not necessarily more “intense”).

We completely agree and have removed the term “intense” and “intensity” when discussing sleep. What we do observe is an increase in sleep consolidation, as measured by an increase in the average length of sleep bouts.

8. This is just a comment. Certainly, adenosine is a neurotransmitter like many others that can regulate sleep, but adenosine as a candidate somnogen is a rather tired (excuse the pun) suggestion (line 200). At least in mammals, there is contradictory evidence about adenosine’s role as a somnogen, and especially about the measurements of and if adenosine does actually increase with wakefulness. It now seems hard to find support for this idea. So, it is interesting here that the authors want to speculatively resurrect it. How do they envisage the adenosine increases being confined to just one part of the cell (for e.g. type 2 neurons)?

We wholeheartedly agree and, honestly, came into this study as strong skeptics of the adenosine hypothesis. However, we now provide new data where the addition of adenosine supports the reduction of synapses, even in the day, which is consistent with adenosine related hypotheses (e.g. Diering et al., 2019). How adenosine might locally alter synapse dynamics, either on neuronal subtypes and even on selective compartments in the same neuron will require future experiments. We provide new data (**Extended Data Figure 12**) that shows at least some single tectal neurons express subsets of both adenosine and adrenergic receptor transcripts. We can only speculate, but one possibility is that receptor protein translation/localization is restricted to neuronal subtypes/compartments. We do see evidence of transcripts localized to dendritic arbors (**Extended Data Figure 12**), an intriguing observation that will require work beyond the scope of this study to evaluate its importance. Another possibility is that differential adenosine/adrenaline accumulation/release within local areas could drive local synapse dynamics. We have added to the discussion to highlight these exciting possibilities.

(9. Minor. Journal name (Neuron?) missing from ref. 25 and ref. 8).

Fixed

Referee #3:

The synaptic homeostasis hypothesis, formulated by Cirelli and Tononi, has shaped much of the thinking about sleep function in the field of neurobiology. The hypothesis has been supported by a large amount of data from a few labs, demonstrating increased measures of synaptic density, strength, and activity following a period of wakefulness when compared with a similar period of sleep. However, in recent years (i.e., the past 10 or so) a number of publications, from many different labs and model systems, have demonstrated that synaptic "downscaling" is not uniform during sleep, may be neuron type-specific, and is not consistently linked to any functional consequences of sleep/sleep deprivation in the brain.

With this in mind, the present study uses a recently-developed genetic tool to image synapse density and "intensity" (which may or may not be related to synaptic strength) in a different neural system, the optic tectum of developing zebrafish. The technique used for this quantification is recently-developed and interesting. However the overall experimental design and outcome are similar to much of what has been presented in the literature recently - some synapses on some neurons (but not all) are reduced after a period of sleep and increased after a period of wake. Hence, this adds another example of a phenomenon to an established body of literature, but does not constitute a major advance.

Our work is different from previous work examining sleep-dependent synaptic homeostasis in a fundamentally important way—we document the entire population of synapse changes on single neurons that are tracked *in vivo* repeatedly during sleep-wake states. To our knowledge, almost all previous work examining SHY were “snapshot” studies of large populations of neurons, where synapse density/strength from one set of animals after a long waking period were compared to a completely different set of animals after a long sleep period^{2–16}. There have been some studies that repeatedly image small patches of dendrites^{17–23}, or groups of presynaptic terminals of a neuronal population²⁴, but as we show in our data, the expected dynamics of small numbers of synapses across brain states will depend wholly on which small set of synapses one examines, and one may get a different result by imaging even a different part of the same neuron.

This is not just a technical distinction, as neither single timepoint snapshots across many neurons in the same animal nor repeated imaging of small subsections of neurons could reveal the sleep-wake dependent dynamics that we reveal occurring at the level of single neurons. One of the major new advances of our study, that the rates of synaptic loss per neuron depend on the level of sleep pressure and that not all periods of sleep fulfill this function, is an important new concept for SHY that could have not been identified by these previous approaches. To strengthen the novelty of our results further, we have now added extensive new data that provide a mechanistic explanation for this relationship between single neuron synapse loss and sleep pressure (new **Figure 4**). In particular, we now show that molecules classically associated with sleep pressure, such as adenosine, coupled with a lower adrenergic tone, are capable of supporting synapse loss, even in the middle of the day, when sleep pressure remains low.

A major confound, however, is that sleep/wake control in the main experiments is primarily a function of ambient light. In the case of a brain region that processes visual information, this raises many questions about how generalizable the findings will be. Even in the case of experiments involving sleep deprivation, the manipulation occurs in red light, and it isn't clear at all that this doesn't constitute a period of heightened visual input to the tectum. This (to my read, more parsimonious) interpretation - that visual input drives synapse formation, and lack thereof drives loss, in the tectum, would reconcile some of the findings in the paper that are incongruous with the hypothesis - i.e., that hypnotic augmented sleep does not cause the same synapse loss.

We do not wholly agree that the use of tectal neurons, which are light responsive, represent a “major confound”. We show under several control conditions that the presence of light alone is not sufficient to support synapse formation. For example, during the constant light “clockbreak” experiments (**Figure 1e-j**), which delivers more ambient visual input than any other condition, synapse numbers do not increase at all—and in fact have a lower, not higher, baseline of synapse numbers (**Figure 1g**). During free-running conditions, which are also continuously exposed to light, synapse numbers only increase during the subjective day, when the animals are also most awake. Moreover, while we use a red light to perform the sleep deprivation studies, the control animals that are allowed to sleep under red light show synapse loss over this same period (**Figure 3b**). Secondly, we now include an additional analysis that shows that natural late sleepers present in the control conditions continue to gain synapses in the first part of the night, while natural early sleepers lose synapses (**Figure 3d,e** and **Extended data figure 8e-g**). Given that these animals experience identical lighting conditions, the most parsimonious explanation is that these synapse changes are associated with sleep/wake state, not light exposure. Thirdly, we have new experiments unpicking the mechanism of sleep-pressure related synapse loss that show a combination of drugs that simultaneously reduce adrenergic tone and increase adenosine levels during the day now support the loss of synapses during sleep (**Figure 4**). Thus, even though these animals are in the presence of light, tectal neurons can be induced to lose synapses under appropriate, sleep-related conditions.

However, we concede the point that tectal neurons do represent a kind of “special case” whose activity may be especially sculpted by visual experience in a way that might not generalize to other neurons of the brain. For this reason, **we now present an analysis of synaptic changes for two additional non-visual neurons**—a population of neurons in the vestibular system and a neuronal type in the hindbrain involved in touch-evoked responses (**Figure 3f-h**). These data show the same sleep-dependent dynamics as tectal neurons—net synapse loss during sleep, net synapse gain during sleep deprivation. We believe this new data provides strong support for the generalizability of our findings in tectal neurons.

Another major concern is with data presentation, in that the authors do not present any image of what precisely they are quantifying. There is a zoomed out view of an entire dendritic arbor, and then a cartoon schematic of what is happening to synapses - no data in the middle. It is hard for any reviewer to evaluate the quality of the data when it isn't even shown in a representative image.

We apologize for this omission in the original manuscript and thank the reviewer for prompting us to make our data analysis and quality clearer for the reader. We now provide in **Extended Data Figure 3** high resolution images of a representative neuron across two time points, including several zoomed-in areas showing examples of synapses that are changing, along with examples of the computer-assisted skeletonized images that assisted in the analysis of the synapse data.

Referee #4 (Remarks to the Author):

In this manuscript, Suppermpool et al. explored the permissive or causal role of sleep on synaptic downscaling using the tectal neurons of larval zebrafish. The authors selected three zeitgeber times spanning one day-night cycle to measure the synaptic number change over daytime and nighttime.

One small note, we also examined 52 neurons across 3 consecutive day-night cycles, encompassing 6 timepoints.

Then they used sleep deprivation (SD) to test the sleep function on synapse number change. Finally, they argued that the sleep's effects on synapse number change is subordinate to sleep pressure for the induced sleep is not sufficient to trigger synapse loss at daytime with low sleep pressure. The authors leveraged the advantage of zebrafish to measure all synapses on individual tectal neurons and found subtype-specific (type 2 tectal neuron) change and regulation by sleep. In total, this study mainly describes a potentially interesting observation that overlaps with a prior study (Appelbaum et al., 2010) while views more from the sleep perspective.

Although Appelbaum et al. 2010 is an important paper that we have cited, there is virtually no overlap at all between our studies. See below in response to major comments.

And the subtype-dependent and even sub-arbor dependent phenomenon is beyond comprehension .

We apologize if our description of the diversity of synapse dynamics was confusing. We have now made several organizational and stylistic changes that we hope have improved the manuscript substantially.

A more detailed mechanistic investigation would strengthen the manuscript.

Thank you for this comment. We have now included several new mechanistic studies and now demonstrate that a combination of low adrenergic tone and high levels of adenosine are capable of supporting synapse loss even during the day, when sleep pressure is low (**Figure 4**). Thus, we give experimental support to our hypothesis that synapse dynamics during sleep are regulated by molecules associated with sleep pressure.

Major Concerns:

1. Appelbaum et al, *Neuron*, (2010) previously showed a homeostatic regulation of synapse number in Hypocretin neurons in larval zebrafish. Although using different cell model, like the present study, they used synaptic marker to assay the change of number of synaptic puncta of individual cells after sleep deprivation . This prior study has striking similarities to the present manuscript in methodology and result: increase of synaptic number after SD. Unfortunately, despite the relevance, this prior study was not cited appropriately or discussed ..

While the Appelbaum et al. (2010) paper was inspirational for our current work, we **strongly disagree** that it is similar to our work in any meaningful way, and we also believe it is appropriately cited in our manuscript. We have modified the reference in our new version to be clearer about what was found in the original Appelbaum paper: (line 41: "and in zebrafish, axon terminals of wake-promoting hypocretin neurons are circadian-clock regulated to peak during the day²⁴"). Below, we show the conceptual, methodological, anatomical, molecular, functional, technical, and interpretive distinctions between our work and this previous work in zebrafish.

First, there are fundamental conceptual differences between the work. Appelbaum et al. examine the axon terminals of a small subset of axons in a whole population of hypocretin neurons of the hypothalamus, while we examine every excitatory post-synapse on single neurons. Thus, Appelbaum et al. examined the synaptic outputs (and only a subset, across a whole population of neurons), while we examine all the excitatory inputs onto single neurons. This is a crucial distinction, because it is the relative number and strength of the total inputs onto neurons, not a subset of outputs, that will affect the overall synaptic "burden" of a neuron. For this reason, the Synaptic Homeostasis Hypothesis (SHY) is relatively silent about pre-synapses and instead aims to explain the dynamics on the post-synaptic side, the site where synaptic strength associated with wakefulness is modulated.

Our paper, which aims to visualize processes associated with SHY, is therefore focused on the post-synaptic side.

Second, Appelbaum et al., use a transgenic line that labels all the hypocretin neurons and images pre-synaptic changes in only some of the target fields of the axons of these neurons. As we show in our manuscript, and elaborate in our response to reviewer 1-3 comments above, population-level synapse dynamics of only some of the synapses masks what is happening at the level of single neurons, and, depending on which subsets of synapses are observed, could in principle yield a variety of day-night synaptic dynamic patterns. While we examine every post-synaptic density on single neurons, and for hundreds of individual neurons, Appelbaum et al., examine just a subset of axons from wake-promoting hypocretin/orexin neurons, and in a neuronal population, not at the level of a single cell. This is critical because, as we show clearly in our manuscript, local synapse dynamics in neuronal sub-compartments can go in any and all directions (Extended Data Figure 7): it is only when you examine the single neurons *in toto* do you see the SHY dynamics at work.

Third, the Appelbaum paper provides good evidence for a circadian influence on hypocretin pre-synapses, but the evidence for homeostatic regulation is not convincing. A few key technical points about this are important to bring up here, to make as clear as possible the very big distinction in findings compared to our work. The sleep deprivation method used in Appelbaum et al. to test the effects of sleep homeostasis was “gentle vibration and tapping”, but no evidence was provided that this actually reduced sleep amount or led to subsequent changes in sleep amount or timing, just effects on locomotion, which is not the same (see their Figure S3C), nor do they show that the method does not phase shift the circadian clock. In contrast, we demonstrate that gentle handling has the predicted effects of sleep deprivation sleep need and subsequent sleep, with shorter sleep latencies, increased total sleep, and more consolidated sleep bouts (our **Extended Data Figure 8**). However, much more importantly, even assuming the sleep deprivation method of Appelbaum et al. worked to alter sleep, they do not actually find an effect of sleep deprivation on the pre-synapses of hypocretin neurons, as the effect is entirely masked by the circadian regulation of synapses. To explain this clearly, we have reproduced the Appelbaum Figure S3 here, which shows all the relevant data for their claim (note that part of the data is repeated in the main Figure, but S3 shows the full dataset).

Figure S3 from Appelbaum et al., Neuron (2010).

On the left are normalized pre-synapse counts after 3 hours of sleep deprivation and on the right is after 6 hours, with three different circadian times before or after the SD. As noted by Appelbaum et al., “these results suggest that 3 hr of SD has no effect on synapse number”. After 6 hours, they claim there is an effect of the SD compared to controls, but close inspection of the CT20 in the left panel compared to the CT20 on the right panel suggests that the **only effect is due to the circadian rhythm** – the data from CT20 looks identically reduced for both 3 hours and 6 hours of sleep

deprivation and for the controls. In other words, *if* there is an effect of SD in Appelbaum et al., it is totally swamped by the effects of the circadian clock, and puncta counts are reduced in the subjective night with or without SD.

In contrast, we see after only 4 hours of sleep deprivation by gentle handling a large increase in post-synapse number for most neurons, and this effect of SD overrides any influence of circadian time (see our **Figure 3**).

For this reason, we believe our citation of Appelbaum et al. as showing circadian influences on axon terminal numbers in a subset of target fields of the population of hypocretin axons is appropriate, and our work is fundamentally different in concept and result.

Finally, although we believe these conceptual and finding differences between the two papers are complete and considerable, there are also critical technical aspects of the work that preclude all but the most superficial similarity between Appelbaum et al. and our study. Most crucially, Appelbaum et al. used overexpression of a GFP-tagged pre-synaptic protein. Not only does this mean that synaptic strength could not be monitored by Applebaum, but overexpression of tagged synaptic proteins is documented to cause synapse artifacts, including the nucleation of ectopic synapses and other issues²⁵⁻²⁷, although these effects are not as substantial on the pre-synaptic side. Instead, we use a recently developed methodology that does not interfere with basic synaptic function and does not involve any overexpression of endogenous proteins^{28,29}. We believe this is an important technical distinction that allows us to make stronger claims about synapse changes across sleep-wake states without fear of over-expression artifacts.

This prior study also examined the circadian influence while the present study only investigated and explained from the sleep perspective

We don't "only investigate and explain from the sleep perspective". We provide a number of experiments showing that circadian influences, including clock break and free-running experiments (**Figure 1**), influence synapse dynamics, but that this influence is mainly via the effect on sleep/wake timing (**Figure 3**).

2. Throughout the manuscript, the author only measured the total number and intensity change of FingR(PSD95) puncta, but articulated this is synapse dynamics. I envision "synapse dynamics" as the concurrent synapse formation and elimination events. What do the authors mean by synapse dynamics and, given their definition, how is synapse dynamics affected?

We are confused about what the reviewer means here. We show for example in **Figure 1d** and **Extended Data Figure 3** "concurrent synapse formation and elimination events", and throughout the paper we show the net gains or loss of synapses per neuron. Ideally, we would have liked to systematically report both the appearance and disappearance of each synapse for each timepoint/neuron. However, because we repeatedly mount and unmount the larva for imaging to enable us to preserve and to track the behavioral state of the animal—a technically challenging protocol—we are reluctant to make overly strongly claims about how every single synapse behaves within the neuron and focus on more modest claims about whole neuron net changes. For example, small changes in the mounting angle of a fish between timepoints can slightly change the interpretation whether a synapse imaged at two points is truly the same synapse versus an event in which one synapse was lost and another was gained closely nearby, especially in some of the denser

arborization fields of these neurons. Future work should explore this further, but that work will be even more technically challenging than the current study and is beyond the scope of this paper.

3. How about increase sleep at nighttime with high sleep pressure as implied by the authors. Can the sleep pressure be experimentally measured?

We have now added experimental measures of sleep pressure in support of our claims. Sleep pressure affects the propensity to fall asleep (i.e. sleep drive), and we show that after SD, three measures of sleep propensity change, including the latency to sleep, the total duration of sleep, and the consolidation of sleep as measured by sleep bout length (**Extended Data Figure 8-9**). We also now include data showing that our daytime drug-induced sleep states are capable of dissipating or slowing the accumulation of sleep pressure during the day, as measured by these parameters at the night following drug treatment.

At single neuron/synapse level, whether the synapses gained during SD correlate with synapse loss or are more sensitive to subsequent sleep recovery.

As noted above, we are reluctant to make strong claims about how specific synapses change during sleep/wake states under experimental conditions where the animal's behavior is also tracked. However, we now include more data that show that neurons that lose synapses during the first part of the night are more likely to gain synapses in the second part of the night, and vice versa (**Figure 3d**), as a function of natural early or late sleepers. Future work will have to disentangle if newly formed synapses during SD are more or less likely to be lost in subsequent sleep periods, but this will require developing a method to observe sleep states in fish that remain mounted under the microscope, a major technical challenge beyond the scope of this study.

Minor Concerns:

1. line 73–‘have a stable window of synapse maturation from 5 to 9 days post fertilization (dpf)’

The synapse should not be mature until about 7 dpf as shown in the previous work (Smith, 2004 , Nat. Neurosci.).

Smith et al., 2004 Nature Neuro. shows a time course that jumps from day 5 then to day 7, so it is difficult to precisely pin down when the synapse numbers are wholly mature. As we show in **Extended Data Figure 2b-d**, we see synapse counts are fairly stable from day5/6. Note, we do our experiments on day 7 and 9, which is consistent with the mature synapse window of Smith et al., 2004 and our own observations. We have fixed the line to clarify this (line 74: “and have a stable window of synapse maturation from 7 to 9 days post fertilization (dpf) (Niell, Meyer, & Smith, 2004)”).

2. line 161–‘Consistent with this, the total time each 160 larva spent asleep was significantly negatively correlated with the rate of synapse loss in both halves of the night.’

This description is controversial. Should not it be positively correlated?

Thank you for catching this. We mean time spent asleep correlates with synapse change. We have corrected the wording to say (line 178), “[...] the total time each larva spent asleep was significantly correlated with the rate of synapse change...”.

References:

1. Tononi, G. & Cirelli, C. Sleep and synaptic homeostasis : a hypothesis. *Brain Res. Bull.* **62**, 143–150 (2003).
2. Abel, T. *et al.* Genomic analysis of sleep deprivation reveals translational regulation in the hippocampus. *Physiol. Genomics* (2012) doi:10.1152/physiolgenomics.00084.2012.
3. Bridi, M. C. D. *et al.* Daily Oscillation of the Excitation-Inhibition Balance in Visual Cortical Circuits. *Neuron* **105**, 621-629.e4 (2020).
4. Mehnert, K. I. *et al.* Circadian changes in Drosophila motor terminals. *Dev. Neurobiol.* (2007) doi:10.1002/dneu.20332.
5. Noya, S. B. *et al.* The forebrain synaptic transcriptome is organized by clocks but its proteome is driven by sleep. *Science (80-.)*. **366**, (2019).
6. Ruiz, S. *et al.* Rhythmic Changes in Synapse Numbers in Drosophila melanogaster Motor Terminals. *PLoS One* **8**, (2013).
7. Vyazovskiy, V. V., Cirelli, C., Pfister-Genskow, M., Faraguna, U. & Tononi, G. Molecular and electrophysiological evidence for net synaptic potentiation in wake and depression in sleep. *Nat. Neurosci.* **11**, 200–208 (2008).
8. Weiss, J. T. & Donlea, J. M. Sleep deprivation results in diverse patterns of synaptic scaling across the Drosophila mushroom bodies. *Curr. Biol.* 1–14 (2021) doi:10.1016/j.cub.2021.05.018.
9. Brodin, A. T. S., Gabulya, S., Wellfelt, K. & Karlsson, T. E. Five Hours Total Sleep Deprivation Does Not Affect CA1 Dendritic Length or Spine Density. *Front. Synaptic Neurosci.* **14**, (2022).
10. Brüning, F. *et al.* Sleep-wake cycles drive daily dynamics of synaptic phosphorylation. *Science (80-.)*. **366**, (2019).
11. Cary, B. A. & Turrigiano, G. G. Stability of neocortical synapses across sleep and wake states during the critical period in rats. *Elife* **10**, 1–28 (2021).
12. de Vivo, L. *et al.* Ultrastructural evidence for synaptic scaling across the wake/sleep cycle. *Science (80-.)*. **355**, 507–510 (2017).
13. Diering, G. H. *et al.* Homer1a drives homeostatic scaling-down of excitatory synapses during sleep. *Science (80-.)*. **515**, 511–515 (2017).
14. Gilestro, G. F., Tononi, G. & Cirelli, C. Widespread Changes in Synaptic Markers as a Function of Sleep and Wakefulness in Drosophila. *Science (80-.)*. **324**, 109–112 (2009).
15. Liu, Z. W., Faraguna, U., Cirelli, C., Tononi, G. & Gao, X. B. Direct evidence for wake-related increases and sleep-related decreases in synaptic strength in rodent cortex. *J. Neurosci.* **30**, 8671–8675 (2010).
16. Mackiewicz, M. *et al.* Macromolecule biosynthesis: A key function of sleep. *Physiol. Genomics* (2007) doi:10.1152/physiolgenomics.00275.2006.
17. Du, X. *et al.* Circadian regulation of developmental synaptogenesis via the hypocretinergic system. (2023) doi:10.1038/s41467-023-38973-w.
18. Li, W. & Gan, W. Sleep promotes branch-specific formation of dendritic spines after learning. *Science* **344**, 1173–1178 (2014).
19. Maret, S., Faraguna, U., Nelson, A. B., Cirelli, C. & Tononi, G. Sleep and waking modulate spine turnover in the adolescent mouse cortex. *Nat. Neurosci.* **14**, 1418–1420 (2011).

20. Miyamoto, D., Marshall, W., Tononi, G. & Cirelli, C. Net decrease in spine-surface GluA1-containing AMPA receptors after post-learning sleep in the adult mouse cortex. *Nat. Commun.* **12**, 1–13 (2021).
21. Norimoto, H. *et al.* Hippocampal ripples down-regulate synapses. *Science (80-.)*. **359**, 1524–1527 (2018).
22. Raven, F., Meerlo, P., Van der Zee, E. A., Abel, T. & Havekes, R. A brief period of sleep deprivation causes spine loss in the dentate gyrus of mice. *Neurobiol. Learn. Mem.* **160**, 83–90 (2019).
23. Yang, G. & Wen-Biao Gan. Sleep contributes to dendritic spine formation and elimination in the developing mouse somatosensory cortex. *Dev Neurobiol.* **23**, 1–7 (2012).
24. Appelbaum, L. *et al.* Circadian and homeostatic regulation of structural synaptic plasticity in hypocretin neurons. *Neuron* **68**, 87–98 (2010).
25. Alder, J., Kanki, H., Valtorta, F., Greengard, P. & Poo, M. M. Overexpression of synaptophysin enhances neurotransmitter secretion at *Xenopus* neuromuscular synapses. *J. Neurosci.* **15**, (1995).
26. Verstraelen, P. *et al.* Image-based profiling of synaptic connectivity in primary neuronal cell culture. *Frontiers in Neuroscience* vol. 12 at <https://doi.org/10.3389/fnins.2018.00389> (2018).
27. Watson, E. T., Pauers, M. M., Seibert, M. J., Vevea, J. D. & Chapman, E. R. Synaptic vesicle proteins are selectively delivered to axons in mammalian neurons. *Elife* **12**, (2023).
28. Gross, G. G. *et al.* Recombinant probes for visualizing endogenous synaptic proteins in living neurons. *Neuron* **78**, 971–985 (2013).
29. Son, J.-H. *et al.* Transgenic FingRs for Live Mapping of Synaptic Dynamics in Genetically-Defined Neurons. *Sci. Rep.* **6**, 18734 (2016).

Reviewer Reports on the First Revision:

Referees' comments:

Referee #1 (Remarks to the Author):

I truly enjoyed reading the scholarly rebuttal of the authors and the revised paper. The authors added a significant amount of new experiments and addressed all my previous concerns, especially related to variability due to single larvae vs single cells. The addition of data on two new cell groups is also very important. I congratulate the authors for an outstanding study.

Chiara Cirelli

Referee #2 (Remarks to the Author):

The authors have responded appropriately and fully to all my concerns. The paper is substantially improved and contains important findings about how sleep pressure is established at the synaptic level. I think it is an elegant study that will interest many researchers in the sleep field and beyond.

Referee #3 (Remarks to the Author):

My major concerns were not addressed by the authors, and remain major concerns about the manuscript.

Referee #4 (Remarks to the Author):

The authors have significantly improved the manuscript and I am satisfied that most major issues have been addressed adequately. One remaining concern is about the definition of “synapse dynamics”. I suggest it may be changed to “synapse number change” or something like. To my knowledge, the synapse dynamics is a concept describing the remodeling of synapses, including formation and elimination events. The net gain and loss of synapses are static results of synapse dynamic changes and thus cannot be viewed as synapse dynamics. A net gain of synapses can derive from either a higher synapse formation rate or a lower synapse elimination rate, and vice versa for a net loss of synapses.

Minor Concerns:

1. How about the synapse intensity change for different tectal neuron subtypes across light:dark cycles (Figure 2)?
2. It would be nice to provide typical image cases for Figure 3b and Figure 4.
3. It had better add statistics in Figure 2d,e, Extended Data Figure 4b,c, Extended Data Figure 5d,e,f, and Extended Data Figure 8d.

4. Extended Data Figure 10, the purple panels should be the melatonin treatment group.

Referee #5 (Remarks to the Author):

The revised manuscript by Suppermpool et. al. is an exemplary investigation of a foundational hypothesis that delivers no less than the first measurements of how all of the synapses on individual neurons change during sleep. Cleverly leveraging the larval zebrafish model, this work sidesteps challenges of monitoring synapses in vivo in other models and convincingly links sleep and synaptic downregulation. In the revision, the authors addressed the bulk of my major concerns, primarily shared with Reviewers 1-2. In particular, their thoughtful analysis of multiple neurons in a single fish absolutely merits inclusion as supplemental material in the paper. Simply put, it sets the bar for future studies. Similarly, to my mind, their pharmacological induction of sleep adequately addresses the familiar “but what about mechanism” critique. I therefore recommend acceptance without hesitation. Below I share major and minor concerns, predominantly linked to reporting of statistics, null hypotheses, and microscopy.

David Schoppik, Ph.D.

1. Statistics. I found presentation of statistical outcomes inconsistent and incomplete across figures. Specifically, Figures 1-3 used an ANOVA and reported outcomes (as best I could tell) without any correction of alpha for multiple comparisons. Crucially, it was not clear to me that the data met the criteria for ANOVA: i.e. were normally distributed with comparable variance across groups. This led to confusion in the text, e.g. line 91 — is a decrease of -1.9% even real? and in the legend for Figure 1 & 2 e.g. line 745 and 746 — the $P < 0.05$ isn't corrected. Fortunately, the authors seem aware of this issue as Figure 4 appropriately uses the non-parametric equivalent and corrects for multiple comparisons. Perhaps the issue is that the authors wish to use the repeated measures design? If so, the manuscript would benefit from explicitly showing that the assumptions of the ANOVA are met, perhaps in a supplemental table? Alternatively, the authors might explore mixed effects models for repeated measures, or estimation statistics as they've used effectively in Benoit et. al. 2023

2. Microscopy (conceptual). If I understood this correctly, the study measured individual puncta across time on a single neuron (e.g. 1d). Excitingly, a large fraction of puncta can be reliably identified from scan to scan. This suggested a way to define a null distribution, defined operationally an estimate of the gain/loss not due to sleep. Have the authors considered repeatedly imaging and measuring puncta over similar timecourses in anesthetized fish? Fixed fish would be better, of course, as those could not change, but perhaps shrinkage / loss of fluorescence would make it too difficult. A report of such a null distribution would be particularly helpful given the statistical challenges above. Apologies if I've missed this somewhere in the supplemental material.

3. Microscopy (reporting). The Airyscan 2 method is appropriately suited for measuring synaptic puncta,

but the manuscript would benefit from directly reporting the actual resolution. Given the ~2Kx2K images at ~60um I'm guessing it's twice Nyquist which is the SR-4's best 140nm x 140nm (X/Y). However, the Leica imaging in 436-443, particularly using the 20x/0.75NA objective, has considerably worse resolution (on order 500nm with 488nm excitation? This made Extended Data Figure 1c confusing, with what appear to be ~60um steps. My recommendation would be to just report the NA and imaging medium of the objectives used, and, when using the Airyscan 2, be clear about the expected resolution. I've listed some of the specific parts in the Methods below. I stress that this is not a concern about the microscopy methods, only a request for clarification.

Minor points:

38 "renormalized"

39 "also increases"

40-41 "imaging...observed" don't agree

50 "at which"

72 "strength" the manuscript rests on the assumption that size/intensity and strength are correlated. I don't believe this has been established in the larval zebrafish outside the spinal motor neurons, and this is a concern given that these are early developmental stages. Authors' call — it's clearly the standard in the field.

91-92 are these changes significant?

189 are you sure these neurons don't receive visual input? are you confident that the more medial cells are actually vestibulospinal?

226-227 this speaks to point 2 — it would be unfortunate if some neurons types were more prone to loss of synapse across imaging sessions because of the particulars of imaging those neurons, not because the numbers truly decrease.

436-443 as above, what was the axial resolution relative to the size of the puncta you're measuring?

449 what's a non-gaussian [sic]?

468 20x what NA? Guessing 1.0.

491 if it's dim red light it's probably not a great choice for urban cycling. A measure of luminance would facilitate replication.

498 here's a great place to report the expected axial resolution

581 no "%" after 100

Author Rebuttals to First Revision:

We thank the reviewers for their overall strong positive support for our manuscript. Referee 4 and 5 brought up some additional minor concerns that we have now fully addressed in a revised manuscript.

Referees' comments in black; our responses in red:

Referee #4 (Remarks to the Author):

The authors have significantly improved the manuscript and I am satisfied that most major issues have been addressed adequately. One remaining concern is about the definition of “synapse dynamics”. I suggest it may be changed to “synapse number change” or something like.

We thank the reviewer for their positive comments. We have made the suggested edit regarding “synapse dynamics” throughout the manuscript.

Minor Concerns:

1. How about the synapse intensity change for different tectal neuron subtypes across light:dark cycles (Figure 2)?

We have now added this data.

2. It would be nice to provide typical image cases for Figure 3b and Figure 4.

We have added typical image cases for Figure 3b and Figure 4 into a new Extended Data Figure 10.

3. It had better add statistics in Figure 2d,e, Extended Data Figure 4b,c, Extended Data Figure 5d,e,f, and Extended Data Figure 8d.

We have now added these statistics. Note that the relevant statistics for Figure 2d, e are handled in the summary Figures 2f,g and Extended Data Figure 5; and the statistics for Extended Figure 4b, c are handled in panels d and e; and Extended Data Figure 5d,e,f are combined into Figure 2f,g.

4. Extended Data Figure 10, the purple panels should be the melatonin treatment group.

Thank you for finding this error, which is now corrected.

Referee #5 (Remarks to the Author):

The revised manuscript by Suppermpool et. al. is an exemplary investigation of a foundational hypothesis that delivers no less than the first measurements of how all of the synapses on individual neurons change during sleep. Cleverly leveraging the larval zebrafish model, this work sidesteps

challenges of monitoring synapses in vivo in other models and convincingly links sleep and synaptic downregulation. In the revision, the authors addressed the bulk of my major concerns, primarily shared with Reviewers 1-2. In particular, their thoughtful analysis of multiple neurons in a single fish absolutely merits inclusion as supplemental material in the paper. Simply put, it sets the bar for future studies. Similarly, to my mind, their pharmacological induction of sleep adequately addresses the familiar “but what about mechanism” critique. I therefore recommend acceptance without hesitation. Below I share major and minor concerns, predominantly linked to reporting of statistics, null hypotheses, and microscopy.

We thank the reviewer for their generous comments about our work and address the comments below.

1. Statistics. I found presentation of statistical outcomes inconsistent and incomplete across figures.

We apologize for the confusion around some of our statistical approaches. We address each point in turn:

Specifically, Figures 1-3 used an ANOVA and reported outcomes (as best I could tell) without any correction of alpha for multiple comparisons.

We now report the ANOVAs with alpha correction for multiple comparisons using Benjamini-Hochberg. This is corrected in all the figure legends and text and stated clearly now in the statistical section of the methods. We also now more clearly state in all cases the interaction statistic p-value for 2-factor mixed ANOVAs.

Crucially, it was not clear to me that the data met the criteria for ANOVA: i.e. were normally distributed with comparable variance across groups.

We have tested all the data for normality using the Shapiro-Wilk test for normality, following up all cases with direct visual inspection of the residual Q-Q plots for each level (timepoint), as suggested by Chambers et al. (1983). Graphical Methods for Data Analysis. Boston. Duxbury Pres. We note that ANOVAs are considered robust against even fairly large violations of normality (discussed in Zar (1999) Biostatistical Analysis, 4th Edition, among other sources), so we believe this approach is appropriate. This is now stated clearly in the Statistics section of the Methods. For repeated and mixed measures designs, we tested for sphericity using Mauchly’s test and corrected with Greenhouse-Geisser when needed; for other comparisons, we tested for equal variances using Levene’s test. In the cases where the equal variance tests failed, we used Welch’s corrected ANOVA for repeated measures, Kruskal-Wallis (for more than two group comparisons) or Mann-Whitney (for two group comparisons).

This led to confusion in the text, e.g. line 91 — is a decrease of -1.9% even real? and in the legend for Figure 1 & 2 e.g. line 745 and 746 — the $P < 0.05$ isn’t corrected.

We agree this line was confusing—we meant here that the changes we see in synapses during the day period are significantly different than seen at night, as measured by a repeated measures ANOVA. The question of whether -1.9% is “real” requires comparison to some ground truth null distribution, which we believe is best captured by the clock-break experiment. See our detailed response to point 2 below. When we compare the L:D data to the clock-break data, we see that the daytime difference is statistically greater in the L:D, but the nighttime decrease is not significantly different. As we demonstrate in Figure 2, this is because the different tectal subtypes have different synapse dynamics across the day:night cycle, only some of which strongly lose synapses at night. For lines 745-746, we now also compare the tectal subtypes to the clock-break null distributions and correct for multiple comparisons (see point 2 below as well).

Fortunately, the authors seem aware of this issue as Figure 4 appropriately uses the non-parametric equivalent and corrects for multiple comparisons. Perhaps the issue is that the authors wish to use the repeated measures design? If so, the manuscript would benefit from explicitly showing that the assumptions of the ANOVA are met, perhaps in a supplemental table? Alternatively, the authors might explore mixed effects models for repeated measures, or estimation statistics as they’ve used effectively in Benoit et. al. 2023.

We are big fans of estimation statistics but feel that a repeated measures/mixed ANOVA design is more easily digestible for the reader for this dataset, given that we follow neurons across multiple timepoints and conditions. However, in line with your suggestion to emphasize effect size estimates over arbitrary p-value cutoffs, we now also report Hedge’s g as an estimation of effect size for key data. Following the reviewer’s suggestion, we have added a supplemental table showing where the assumptions of ANOVA are met and where non-parametric stats are required.

2. Microscopy (conceptual). If I understood this correctly, the study measured individual puncta across time on a single neuron (e.g. 1d). Excitingly, a large fraction of puncta can be reliably identified from scan to scan. This suggested a way to define a null distribution, defined operationally an estimate of the gain/loss not due to sleep. Have the authors considered repeatedly imaging and measuring puncta over similar timecourses in anesthetized fish? Fixed fish would be better, of course, as those could not change, but perhaps shrinkage / loss of fluorescence would make it too difficult. A report of such a null distribution would be particularly helpful given the statistical challenges above. Apologies if I’ve missed this somewhere in the supplemental material.

This is an interesting idea but there are several challenges to the proposed methods. As the reviewer notes, repeated imaging of fixed fish on a similar timescale is not possible due primarily to the loss of FingR-GFP fluorescence, which in our hands must be imaged without antibody amplification (e.g., the synapse becomes hard to distinguish from cellular background fluorescence, possibly due to amplification of a very low-level of FingR-GFP trafficking to and from the nucleus.) The long-term anesthetized fish experiment is interesting, but unpublished single cell data we have generated of fish exposed to anaesthesia even for several hours identified sets of neurons that apparently are activated, as measured by the expression of many immediate early genes. In other contexts/species, sleep

promoting neurons are also known to be activated by various anaesthetics (e.g., see L.E. Nelson, et al. “The sedative component of anesthesia is mediated by GABA(A) receptors in an endogenous sleep pathway”. *Nat. Neurosci.*, 5 (2002), pp. 979-984) or Moore et al. “Direct activation of sleep-promoting VLPO neurons by volatile anesthetics contributes to anesthetic hypnosis”. *Curr. Biol.*, 22 (2012), pp. 2008-2016.) Thus, we are not certain that long-term anaesthesia would represent a true null distribution for synapse changes and would be a fascinating future study by itself.

However, we believe that our “clock break” experiments represent the kind of null distribution the reviewer envisions, allowing us to estimate the “gain/loss not due to sleep”. This is because, when raised in constant light, the larvae fail to develop a coherent circadian clock to drive consolidated bouts of sleep and wake, making sleep/wake equally distributed across time (e.g., Figure 1F and Prober et al., 2006). Unlike the proposed fixed fish test, which would only account for technical variance, the clock-break experiment also accounts for other biological factors such as developmental age, while avoiding the possible complications of anaesthesia. Consistent with the notion that this represents a useful null distribution, under these conditions, neither the synapse number nor the synapse PSD-95 content change across the repeated measurement windows (Figure 1g-j).

We have now made more explicit in our manuscript the notion that the clock break experiment is a null distribution and made the appropriate comparisons to this dataset for claims of significant changes due to sleep/wake and day/night. We have also used the clock break experiment to test the extent to which we get the same synapse measurements across time for the tectal subtypes (see our response to the minor point below).

3. Microscopy (reporting). The Airyscan 2 method is appropriately suited for measuring synaptic puncta, but the manuscript would benefit from directly reporting the actual resolution. Given the ~2Kx2K images at ~60um I’m guessing it’s twice Nyquist which is the SR-4’s best 140nm x 140nm (X/Y). However, the Leica imaging in 436-443, particularly using the 20x/0.75NA objective, has considerably worse resolution (on order 500nm with 488nm excitation? This made Extended Data Figure 1c confusing, with what appear to be ~60um steps.

My recommendation would be to just report the NA and imaging medium of the objectives used, and, when using the Airyscan 2, be clear about the expected resolution. I’ve listed some of the specific parts in the Methods below. I stress that this is not a concern about the microscopy methods, only a request for clarification.

We have now added the requested microscopy details to the methods. Thank you for pointing out these omissions in the original document. See also minor comments below.

Minor points:

38 “renormalized”

fixed

39 “also increases”

fixed

40-41 “imaging...observed” don’t agree

fixed

50 “at which”

fixed

72 “strength” the manuscript rests on the assumption that size/intensity and strength are correlated. I don’t believe this has been established in the larval zebrafish outside the spinal motor neurons, and this is a concern given that these are early developmental stages. Authors’ call — it’s clearly the standard in the field.

We meant here “strength of the MAGUK expression”, and not synaptic strength per se. We have therefore changed this to say “readout of synaptic PSD-95 content”. While the number of PSD-95 molecules has been reported to be proportional to synaptic strength, there are exceptions to this and we do not definitively know this is the case for all synapses of a zebrafish larva. For these reasons we have focused throughout the manuscript on net synapse counts, which do not depend on this assumption, but also report changes in the PSD-95 intensities we observe across the day:night cycle.

91-92 are these changes significant?

Yes, as indicated in the graph. However, also see above our response about the statistical approaches we have taken in this manuscript to address your comments.

189 are you sure these neurons don’t receive visual input? are you confident that the more medial cells are actually vestibulospinal?

Good point. Here we mean “direct retinal input”, as to our knowledge, these neurons do not reside in any of the known arborization fields for retinal ganglion cells in zebrafish (Baier, H., & Wullmann, M. F. (2021). Anatomy and function of retinorecipient arborization fields in zebrafish. *Journal of Comparative Neurology*, 529(15), 3454–3476.). We have clarified this in the text. As for the identity of these cells as vestibulospinal, we cannot be 100% certain and based this on their location and appearance. We have rewritten the text to call these “presumptive vestibulospinal neurons” to clarify this uncertainty. Regardless, this does not change the results or conclusions of the manuscript.

226-227 this speaks to point 2 — it would be unfortunate if some neurons types were more prone to loss of synapse across imaging sessions because of the particulars of imaging those neurons, not because the numbers truly decrease.

In extended data Figure 5d-f, we present long term tracking of multiple single neurons for the tectal subtypes across three day-night cycles. Although the synapse counts change from session to session, there is little evidence for any of the subtypes to show a systematic or progressive loss of synapse counts from one session to the next, which you would expect if a subtype got harder and harder to

image. One exception might be Type 4 neurons getting harder to image after the first imaging session and then becoming stable, but see below why we believe this is unlikely.

To test this more explicitly, we re-analysed each tectal subtype under the constant light “clock break” conditions that represent a null distribution for sleep/wake—if some neuron types are more prone to loss of synapses across imaging sessions, we would expect this to also appear in the clock break data. Under clock break conditions, the rate of synapse changes is similar across all tectal subtypes, suggesting that there is not a technical bias in detecting synapses across the subtypes. We have added these observations to the Extended Data Figure 5g-j and Extended Data Figure 6. One complication is that we also found that the relative numbers of the four subtypes is developmentally altered in clock break conditions, so Type 2 neurons are underrepresented in this dataset and no conclusions can be drawn about it. Nevertheless, it is very unlikely there are any imaging systematic imaging problems with Type 2 neurons, as, given their robust day-night cycling dynamic, one would have to speculate that they become harder to image, then easier to image, then harder to image again, etc. Also note that, under clock break conditions, Type 4 neurons’ synapses remain relatively stable, suggesting there is not a technical difficulty in repeatedly imaging these neurons.

How the circadian clock alters the development of the cellular identities that make up the tectum is unknown and will be investigated in a future study.

436-443 as above, what was the axial resolution relative to the size of the puncta you’re measuring?

For the antibody co-labelling experiment in Extended Figure 1c, we performed the analysis on a maximum projection (in Z) of 5 to 10 micron Z-stacks taken at 1 micron step intervals. Thus, in Z, the resolution cannot be better than 5-10 microns. Then, cross-section lines across single puncta were taken in the X-Y plane of the max projection to identify the co-localization of FingR-PSD95.GFP and MAGUK single puncta signals. Since this was a sense-check validation of the tool, this methodology was modified from Son et al., (2016) “Transgenic FingRs for Live Mapping of Synaptic Dynamics in Genetically-Defined Neurons”. Sci Rep. 2016 Jan 5;6:18734, where the fidelity of FingR-PSD95 for labelling zebrafish synapses was first described.

449 what’s a non-gaussian [sic]?

Thanks for catching this placeholder omission. This now states, “the normalized grey values were interpolated with a cubic polynomial implemented by the SciPy(v1.11.4) function `scipy.interpolate.interp1d`”

468 20x what NA? Guessing 1.0.

Yes, the NA was 1.0. We have added this to the methods along with axial resolution from the Airyscan.

491 if it’s dim red light it’s probably not a great choice for urban cycling. A measure of luminance would facilitate replication.

We measured the light level as experienced by the larvae in the range of 5.2-30.5 lux. This is now reported in the text.

498 here's a great place to report the expected axial resolution

We have now added the axial resolution here.

581 no "%" after 100

Fixed

Reviewer Reports on the Second Revision:

Referees' comments:

Referee #5 (Remarks to the Author):

The authors have adequately addressed all of my concerns and I find the revised manuscript to be considerably stronger for their effort. I congratulate the authors on a fine piece of work.

Author Rebuttals to Second Revision:

Response to Reviewers:

Referee #5 (Remarks to the Author):

The authors have adequately addressed all of my concerns and I find the revised manuscript to be considerably stronger for their effort. I congratulate the authors on a fine piece of work.

David Schoppik, Ph.D.

NYU Grossman School of Medicine

We thank the reviewer for their suggestions and kind words.